# Unified quantitative characterization of epithelial tissue development

**Boris Guirao[1]\*, Stéphane U Rigaud[1†], Floris Bosveld[1†], Anaïs Bailles[1§], Jesús López-Gay[1], Shuji Ishihara[2], Kaoru Sugimura[3,4], François Graner[5]\*‡, Yohanns Bellaïche[1]\*‡**

[1]Polarity, Division and Morphogenesis Team, Genetics and Developmental Biology Unit (CNRS UMR3215/Inserm U934), Institut Curie, Paris, France; [2]Department of Physics, School of Science and Technology, Meiji University, Kanagawa, Japan; [3]Institute for Integrated Cell-Material Sciences, Kyoto University, Kyoto, Japan; [4]Precursory Research for Embryonic Science and Technology (PRESTO), Japan Science and Technology Agency, Tokyo, Japan; [5]Laboratoire Matière et Systèmes Complexes (CNRS UMR7057), Université Paris-Diderot, Paris, France

**\*For correspondence:** boris.guirao@curie.fr (BG); francois.graner@univ-paris-diderot.fr (FG); yohanns.bellaiche@curie.fr (YB)

[†]These authors contributed equally to this work
[‡]These authors also contributed equally to this work

**Present address:** [§]Institut de Biologie du Développement de Marseille, Aix Marseille Université, Marseille, France

**Competing interests:** The authors declare that no competing interests exist.

**Abstract** Understanding the mechanisms regulating development requires a quantitative characterization of cell divisions, rearrangements, cell size and shape changes, and apoptoses. We developed a multiscale formalism that relates the characterizations of each cell process to tissue growth and morphogenesis. Having validated the formalism on computer simulations, we quantified separately all morphogenetic events in the *Drosophila* dorsal thorax and wing pupal epithelia to obtain comprehensive statistical maps linking cell and tissue scale dynamics. While globally cell shape changes, rearrangements and divisions all significantly participate in tissue morphogenesis, locally, their relative participations display major variations in space and time. By blocking division we analyzed the impact of division on rearrangements, cell shape changes and tissue morphogenesis. Finally, by combining the formalism with mechanical stress measurement, we evidenced unexpected interplays between patterns of tissue elongation, cell division and stress. Our formalism provides a novel and rigorous approach to uncover mechanisms governing tissue development.

## Introduction

The advances in live imaging have beautifully illustrated how the growth and shaping of tissues and organs emerge from the collective dynamics of cells (for review see [*Keller, 2013*]). In this context, the development of quantitative methods is essential to determine the role in tissue and organ development of each cell process, in particular cell divisions, cell rearrangements, cell size and shape changes and apoptoses (*Rauzi et al., 2008*; *Blanchard et al., 2009*; *Aigouy et al., 2010*; *Bosveld et al., 2012*; *Tomer et al., 2012*; *Krzic et al., 2012*; *Economou et al., 2013*; *Heller et al., 2014*; *Khan et al., 2014*; *Zulueta-Coarasa et al., 2014*; *Monier et al., 2015*; *Lau et al., 2015*; *Rozbicki et al., 2015*; *Etournay et al., 2015*). However, a general formalism valid in two and three dimensions and allowing to unambiguously decompose tissue growth and morphogenesis into the parts due to each cell process is still needed. Building such a formalism is critical to define the roles of each process and advance our understanding of how gene activities and mechanical forces cooperate in controlling cell dynamics to regulate the growth, shaping, repair or homeostasis of monolayered or three-dimensional cohesive tissues. In particular the lack of a general formalism has impeded

**eLife digest** In animals, the final size and shape of each tissue is determined by the precise control of when, where and how much individual cells grow, divide, move and die. An important challenge in biology is to understand how the behaviors of each individual cell can act together to generate a large and reproducible change at the scale of entire tissues and organs. Here, Guirao et al. have developed a new approach to provide maps that reveal how much each cell process contributes to the development of tissues.

A caterpillar becoming a butterfly is a famous example of insect 'metamorphosis'. The fruit fly offers another example of such tissue development: within five days, a rice grain-like maggot morphs into an adult fly with long antennae, legs and wings. Guirao et al. used a microscope to observe cells over a period of several hours during the metamorphosis of the adult fruit fly wings and thorax (the region between the neck and abdomen).

In both regions, Guirao et al. showed that all the cell processes participate in the formation of the adult tissue. Cell division, cell death, and changes in cell size affect the size of the tissue, while cell division, cell rearrangements, and changes in cell shape alter the shape of the tissue. The relative contributions of these cell processes varied a lot in both space and time. Further experiments then used mutant flies with defects in cell division to analyse the impact of cell division on the other cell processes and the eventual shape of the tissue. Finally, Guirao et al. showed that there are unexpected interactions between the patterns of tissue growth, cell division and the mechanical forces in the tissue.

These findings provide a new approach to uncover how animals from different species can have such a variety of shapes and sizes, even though they each start life as a single cell. Ultimately, this may also aid efforts to understand how certain diseases affect the development of tissues.

a comprehensive characterization of the role of cell divisions and of their orientations during tissue development.

The growth and morphogenesis of tissues typically involves cell divisions leading to the formation of organs of the correct size and shape. So far, two fundamental properties of cell division have been reported during the morphogenesis of proliferative tissues. First, the orientation of cell divisions has been shown to play a key role in tissue elongation, either by being an anisotropic source of force, or by reducing mechanical stress (*Baena-López et al., 2005*; *Saburi et al., 2008*; *Aigouy et al., 2010*; *Quesada-Hernández et al., 2010*; *Ranft et al., 2010*; *Mao et al., 2011*; *Gibson et al., 2011*; *Aliee et al., 2012*; *Mao et al., 2013*; *Legoff et al., 2013*; *Campinho et al., 2013*; *Wyatt et al., 2015*). Second, cell division orientation has been reported to be mainly oriented along the direction of mechanical stress (*Fink et al., 2011*; *Mao et al., 2013*; *Legoff et al., 2013*; *Campinho et al., 2013*; *Wyatt et al., 2015*). These conclusions were mainly drawn from the observation of tissues where cell divisions are the major contributor of tissue elongation or where cell divisions, tissue deformation and mechanical stress display a common preferred orientation. Yet, embryos, as well as many other tissues or organs, are heterogeneous: cell divisions display rates and orientations varying in space and time, and they are concomitant to other morphogenetic processes such as cell rearrangements, cell size and shape changes and apoptoses. One of the broad challenges in developmental biology is therefore to test and generalize these proposed functions of cell divisions in more heterogeneous contexts.

## Results and discussion

### A formalism to quantify tissue development and its validation using simulations

We have previously proposed a statistical method to quantify cell rearrangements and cell size and shape changes during tissue development (*Bosveld et al., 2012*; *Bardet et al., 2013*). The method took advantage of the links joining the centroids of a cell and its neighbors (*Graner et al., 2008*). Measuring the changes of position, size and direction of links, as well as link swapping, yielded a

quantification of cell rearrangements and cell size and shape changes characterized by an amplitude, an anisotropy and an orientation. This enabled to separately measure cell rearrangements and cell size and shape changes during tissue development. Generalizing the method to incorporate the remaining biological cell processes, in particular cell divisions and apoptoses, is not straightforward since these cell processes can substantially modify the cell and link numbers. We therefore developed a novel formalism that takes into account the changes in link number and which disentangles the measurement of each cell process during tissue development (See Appendix for detailed information and comparison with previous approaches).

In this novel formalism, valid in two and three dimensions, the four main cell processes (cell divisions, cell rearrangements, cell shape and size changes and apoptoses) are unambiguously distinguished and independently quantified by four measurements ($\mathbf{D}$, $\mathbf{R}$, $\mathbf{S}$ and $\mathbf{A}$, respectively). These four measurements quantify the changes in link length or orientation as well as link appearances or disappearances due to their respective cell processes; they add up to the local tissue rate of deformation measuring the rate of tissue growth and morphogenesis ($\mathbf{G}$) due to these four cell processes (*Figure 1a*). Whenever needed, the subdivision of these main cell processes can be further refined, for instance in a mono-layered tissue to distinguish apoptoses from live cell extrusions, or to distinguish simple rearrangements through four-fold vertices from those with five-fold vertices or higher. Furthermore, introducing cell divisions and apoptoses naturally enables the addition of the other cell processes changing the number of cells such as: (i) the integration of new cells in epithelium sheets ($\mathbf{N}$); (ii) the fusion (coalescence) of cells ($\mathbf{C}$), and (iii) the in/outward cell flux ($\mathbf{J}$), representing the cells entering and exiting the microscope field of view or the boundaries of the tissue of interest (*Figure 1—figure supplement 1a*). The formalism therefore yields a complete and unified quantitative characterization of tissue deformation and of all cell processes reported to occur during epithelial tissue development and homeostasis.

In a tissue where tissue deformation is solely associated with cell divisions, cell rearrangements, cell size and shape changes and apoptoses, this unified characterization is expressed as a balance equation where the deformation rate of a region in the tissue is decomposed into the sum of the deformation rates associated with each cell process:

$$\mathbf{G} = \mathbf{D} + \mathbf{R} + \mathbf{S} + \mathbf{A} \tag{1}$$

Note that in the absence of divisions, rearrangements and apoptoses (i.e. $\mathbf{D} = \mathbf{R} = \mathbf{A} = 0$), our formalism therefore yields an exact equality between the rates of tissue deformation $\mathbf{G}$ and of cell size and shape changes $\mathbf{S}$.

Here, we explicitly describe its practical implementation and measurements in the context of the general interest in understanding the growth and morphogenesis of epithelial sheets (*Heisenberg and Bellaïche, 2013*; *Guillot and Lecuit, 2013*). We first acquire a time-lapse movie in which cell apical contours have been labeled by E-Cadherin:GFP, images are segmented and cells tracked to determine their positions over time and their lineages. Then, the formalism is applied between successive images to separately measure the growth and morphogenesis associated to each cell process ($\mathbf{D}$, $\mathbf{R}$, $\mathbf{S}$ and $\mathbf{A}$) and of the tissue ($\mathbf{G}$) (*Figure 1b* and *Figure 1—figure supplement 1b*). Each measurement can be represented with a circle and a bar (*Figure 1c* and *Figure 1—figure supplement 1c*). The circle diameter represents the local rate of tissue isotropic dilation or tissue growth: it is positive for an increase in size (white filled circle) and negative for a decrease (grey filled circle). The bar, which has a length and orientation, represents the local rate of tissue anisotropic deformation or tissue morphogenesis: it quantifies the local elongation rate (and respective equal contraction rate in the orthogonal direction), thereafter named the contraction-elongation (CE) rate (*Figure 1d*). Finally, the analysis is multi-scale, in the sense that each statistical measurement can be averaged at any supra-cellular scale over space, over time, and over several animals, thereby linking the length and time scales associated with cells, groups of cells and the whole tissue (*Figure 1e*, *Video 1*).

In order to confirm that each measurement quantifies unambiguously and accurately its associated biological process, we applied the formalism on computer simulations of cell patches undergoing known deformations. In each simulation, we imposed that the growth and morphogenesis of the cell patch was mainly driven by only one of the cell processes: cell divisions, cell rearrangements, cell size and shape changes or apoptoses. We first tested the measurements of $\mathbf{G}$ and $\mathbf{S}$ by imposing an isotropic dilation of the cell patch, followed by its CE along the horizontal axis, both patch

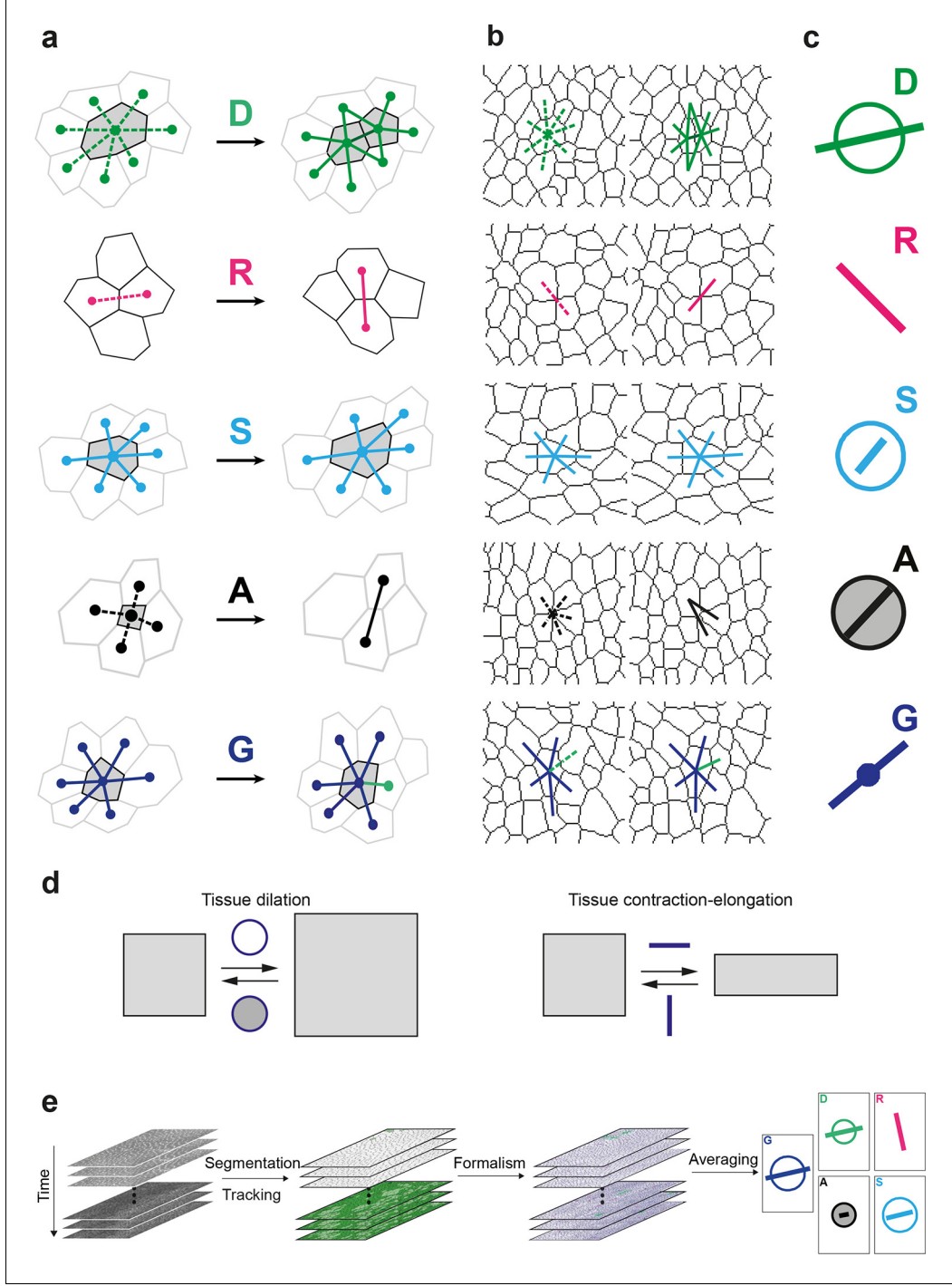

**Figure 1.** Definitions of the main formalism quantities and analysis workflow. (**a**) Characterizations of the four main elementary cell processes and of tissue deformation: **D** divisions (green; and dark green for the link created between the daughter cells); **R**, rearrangements (magenta); **S**, size and shape changes (cyan); **A**, apoptosis/delaminations (black). They are defined and measured from the rates of changes in length, direction and number of cell-cell links, here on two schematized successive images. They make up the tissue deformation rate **G**, the measurement of which is based on geometric changes of conserved links (dark blue links) excluding non-conserved links (green). Dots indicate cell centroids. Lines are links between neighbor cell centroids. Dashes are links on the first image (left) which are no longer present on the second one (right). Some cells are hatched in grey to facilitate the comparison. (**b**) Measurements of the four elementary main cell processes rates and of tissue deformation rate. Same as (a), this time showing cell-cell links on two actual successive segmented images

*Figure 1 continued on next page*

*Figure 1 continued*

extracted from experimental time-lapse movies. (**c**) Representation with circles and bars of the quantitative measurements performed on (**b**) of the deformation rates explained in (**d**). (**d**) Deformation rate: a deformation quantifies a relative change in tissue dimensions: it is expressed without unit, e.g. as percents. A deformation rate is thus expressed as the inverse of a time, e.g. $10^{-2}$ h$^{-1}$ represents a 1% change in dimension within one hour. It can be decomposed into two parts. First (left): an isotropic part that relates to local changes in size. The isotropic part can either be positive or negative, reflecting a local isotropic growth or shrinkage of the tissue. The rate of dilation is represented by a circle, the diameter of which scales with the magnitude of the rate. Positive and negative dilations are represented by circles filled with white and grey, respectively. Second (right): an anisotropic part that relates to local changes in shape. The anisotropic part of the deformation rate quantifies the local contraction-elongation or convergence-extension (CE) without change in size. It can be represented by a bar in the direction of the elongation, the length and direction of which quantify the magnitude and the orientation of the elongation. (**e**) Workflow used to quantify tissue development. Image analysis leads to characterization of cell contours (segmentation), and lineages (tracking) in the case of movies. Our formalism yields an identification of each cell-level process and its description in terms of cell-cell links (see a–b) and a quantitative measurement of their associated deformation rate (see c–d). Averaging over time, space and/or movies of different animals yields a map of each quantity in each region of space at each time with a good signal-to-noise ratio (see *Videos 1*, *4*, *5*).

The following figure supplement is available for figure 1:

**Figure supplement 1.** Characterizations of the additional elementary cell processes **N**, **C**, **J**.

---

deformations solely occurring via cell size and shape changes. We independently measured the imposed deformation rates for **G** and **S** with 0.3% of error, and obtained **G** = **S** as expected (*Figure 2a*, *Video 2a*). Next, we tested the measurements of **D**, **R**, **A** by allowing deformation of the cell patch by oriented cell divisions, oriented rearrangements and apoptoses, respectively. In each simulation, the balance equation shows that the tissue deformation rate **G** was determined by the main process enabling the deformation of the cell patch (*Figure 2b–d*, *Video 2b–d*; see *Figure 2—figure supplement 1* and *Video 2e–i* for the others processes). This confirmed that the formalism unambiguously measures the tissue deformation rate as well as the deformation rates associated with each individual cell process.

## Quantitative characterization of epithelial tissue growth and morphogenesis

Having validated the formalism in silico, we illustrate its relevance to study tissue development by undertaking an analysis of the role of cell division orientation and its relationship to other cell processes and to mechanical stress during the development of a heterogeneous epithelial tissue. During pupal metamorphosis, the *Drosophila* dorsal thorax (notum, yellow dashed box in *Figure 3a,b*) is a monolayered cuboidal epithelial tissue. From 10 h after pupa formation (hAPF), it undergoes several morphogenetic movements associated with cell divisions, cell rearrangements and cell size and shape changes as well as delaminations, which can be due to live cell extrusions or apoptoses (*Bosveld et al., 2012*; *Marinari et al., 2012*). An important feature of this tissue is its heterogeneity, which enables to simultaneously investigate the various mechanisms driving morphogenesis and their interplays. Furthermore, applying our formalism on this tissue will provide a valuable resource since it is a general model to uncover conserved mechanisms that regulate planar cell polarization, tissue morphogenesis, tissue homeostasis and tissue mechanics, and to perform genome-wide RNAi screen (see for example [*Mummery-Widmer et al., 2009*; *Olguín et al., 2011*; *Bosveld et al., 2012*; *Marinari et al., 2012*; *Antunes et al., 2013*]).

We imaged the development of this tissue by labeling cell adherens junctions with E-Cadherin: GFP and followed ~10 $10^3$ cells over several cell cycles with 5 min resolution from at least 14 to 28 hAPF. We segmented and tracked the cells of the whole movie (~3 $10^6$ cell contours with a relative error below $10^{-4}$, *Figure 3c*, *Video 3a*). The display of cell displacements, as well as the tracking of cell patches deforming over time enable to visualize the heterogeneity of tissue growth and morphogenesis between 14 and 28 hAPF (*Figure 3d*, *Video 3b,c*). Directly measuring the rate and orientation of cell divisions, we found that ~17 $10^3$ divisions take place during the development of the tissue, and that both the cell division rates and orientations display major variations in space and

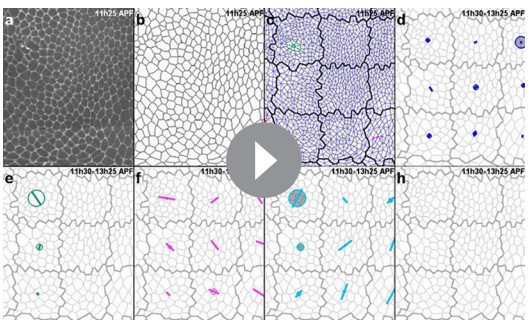

**Video 1.** Workflow of measurements of tissue and cell process CE rates at the patch scale. (a) Detail of a movie in the scutellum region, tissue labeled with E-Cad:GFP and imaged by multi-position confocal microscopy at a 5 min time resolution, 11:25 to 27:25 hAPF. (b) Cell tracking: cells are colored in shades of green according to their number of divisions: light (1), medium (2), dark ($\geq$3); black for the last five frames before a delamination; red for fused cells. (c) Evolution of cell-cell links: links which appear or disappear are represented with thick straight lines and colored as follows: divisions (green), rearrangements (magenta), delaminations (black), integrations (purple), fusions (red), boundary flux (orange). Conserved links are represented with thin dark blue lines. Links corresponding to four-fold vertices are in lighter colors. Cell contours are indicated by thin grey outlines, patch contours by thick black outlines. (d-h) Maps of dilation rates (circles filled in white [positive] or grey [negative]) and of CE rates (orientation: bar direction; anisotropy: bar length), for (d) the tissue $\mathbf{G}$ (compare bar amplitudes and orientations with the *evolution* of patch shapes), (e) cell divisions $\mathbf{D}$, (f) rearrangements $\mathbf{R}$, (g) cell size and shape changes $\mathbf{S}$ and (h) delaminations $\mathbf{A}$. Patch contours are indicated by thick grey outlines. Dilation and CE rates in a given patch are calculated from the evolution of links in this patch between two successive images, then averaged with a sliding window of 2 h. The stillness at the beginning and end of the measurement movies comes from this time averaging.

time (*Figure 3c,e*). Cell division rate is higher in the posterior part of the tissue (*Figure 3c*) and many regions harbor oriented cell divisions (*Figure 3e*). Division orientation is represented by a bar ($\mathbf{D_o}$, Appendix C.3.2), the length and orientation of which represent respectively the cell division orientation anisotropy and main orientation in each region. In particular, in the central posterior part of the tissue the orientation of cell divisions is medial-lateral, while in a more anterior and lateral domain cell divisions are oriented at roughly 45° relative to the anterior-posterior axis (*Figure 3e*, boxes). While these descriptions of tissue development by following patches of cells, cell division rate and orientation are essential, we now explain how the formalism enables to rigorously tackle three major steps to quantitatively study the morphogenesis of an epithelial tissue: (i) measure the local CE rates associated with each process for one animal; (ii) determine the average and the variability of cell dynamics over several animals; and (iii) measure the components of each cell process CE rate along tissue morphogenesis.

## Measurement of CE rates associated with each cell process in one animal

In order to quantify development over the whole tissue from cell to tissue level, we applied our formalism to measure $\mathbf{G}$, $\mathbf{D}$, $\mathbf{R}$, $\mathbf{S}$, $\mathbf{A}$ using 2 h sliding window averages in cell patches of initially $40{\times}40$ μm$^2$, typically encompassing tens of cells that were tracked between 14 and 28 hAPF (*Video 4a-e*). Dilation and CE rate maps vary smoothly in both space and time, while remaining symmetric relative to the midline axis. This indicates that the averaging length scale ($40{\times}40$ μm$^2$) and time-scale (2 h) are appropriate to describe the dynamics of the tissue; moreover, the symmetry indicates that each process is regulated. The results were also plotted as 14 h average maps on the last image analyzed to quantify the growth and morphogenesis of each patch between 14 and 28 hAPF (*Figure 3f–j* and *Figure 3—figure supplements 1*, *2*). We find that tissue dilation mainly occurs through divisions, cell size changes and apoptoses, the dilation rates of the other processes being negligible (*Figure 3—figure supplement 1*). While the dilation rates are critical to study important processes such as apical constriction or local tissue growth, we focus here on the CE rates to illustrate how the formalism enables to better understand the role of cell processes in morphogenesis, with a focus on oriented cell divisions. The tissue CE rate $\mathbf{G}$ map demonstrates that the tissue undergoes various CE movements, in particular in its posterior medial and lateral domains (*Figure 3f*, boxes). As expected, the cell division orientation ($\mathbf{D_o}$) and division CE rate ($\mathbf{D}$) maps (*Figure 3e,g*) show strong similarities (alignment = 0.94). Importantly, the formalism now enables to compare directly the division CE rate $\mathbf{D}$ to the other CE rates. First, the $\mathbf{G}$ and $\mathbf{D}$ CE rates are roughly aligned in the medial posterior region, while they have clearly different orientations in the lateral domains (*Figure 3f,g*). Second, both cell rearrangements ($\mathbf{R}$) and cell shape changes ($\mathbf{S}$) have strong CE rates in many domains

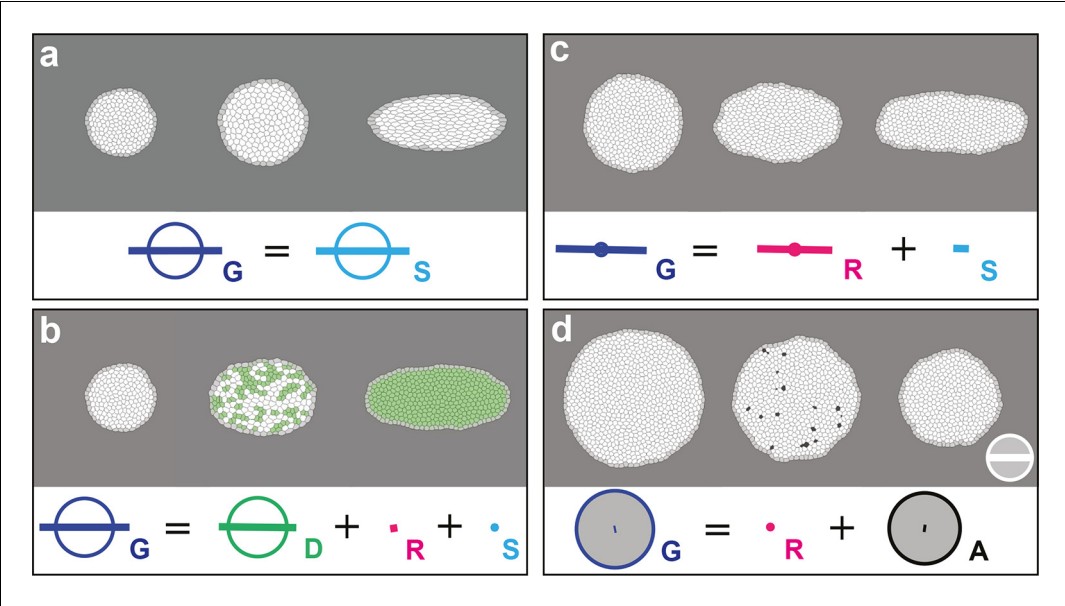

**Figure 2.** Computer simulations validating the quantitative characterizations of the main cell processes and tissue deformation. In (a–d), upper panels: simulated deformation of a cell patch; left: initial state of the simulation; middle: intermediate state; right: final state. Lower panels: *Equation 1* is visually displayed. (a) By direct image manipulation (hence not followed by any cell shape relaxation), the initial pattern (left) is dilated (middle) then stretched (right) with known dilation and CE rates, thereby solely generating the same size and shape changes for the patch and for each individual cell. The patch deformation rate $\mathbf{G}$ and the cell size and shape change rate $\mathbf{S}$ are measured independantly with 0.3% of error, and, as expected when no topological changes occur, we find $\mathbf{G} = \mathbf{S}$. This validates the measurement of $\mathbf{G}$ and $\mathbf{S}$, which in turn validates the other measurements in the next simulations. (b) Potts model simulation of oriented cell divisions. Forces are numerically implemented along the horizontal axis. They drive the elongation of the cell patch while each cell divides once along the same axis. Therefore both $\mathbf{G}$ and $\mathbf{D}$ have their anisotropic parts along the horizontal direction. The residual cell rearrangements and cell shape changes CE rates $\mathbf{R}$ and $\mathbf{S}$ are respectively due to some cell rearrangements actually occurring in the simulation, and to some cells having not completely relaxed to their initial sizes and shapes. This is not due to any entanglement between the cell process measurements in the formalism. Divided cells are in green. (c) Potts model simulation of oriented cell rearrangements. The same forces as in (b) drive the elongation of the cell patch first leading to the elongation of cells that then relax their shape by undergoing oriented rearrangements along the same axis, thereby leading to both $\mathbf{G}$ and $\mathbf{R}$ having their anisotropic parts along the horizontal direction. The cell shape relaxation is not complete as cells remain slightly elongated by the end of the simulation (right), thereby giving a residual $\mathbf{S}$. (d) Potts model simulation of cell delaminations. Delaminations were obtained by gradually decreasing the cell target areas of half the cells of the initial patch to 0, thereby driving the isotropic shrinkage of the patch to half of its initial size. It leads to equal negative growth rates for $\mathbf{G}$ and $\mathbf{A}$, up to residual other processes. Delaminating cells are in black. The white scale bar and circle in (d) both correspond respectively to CE and growth rates of $10^{-2}$ $h^{-1}$ for simulation movies lasting 20 h, in all panels (a–d). Only measurements with norm $> 10^{-3}$ $h^{-1}$ have been plotted.

The following figure supplement is available for figure 2:

**Figure supplement 1.** Computer simulations validating the quantitative characterization of the additional cell processes $\mathbf{N}$, $\mathbf{C}$, $\mathbf{J}$ and testing rotation.

where cell divisions are also oriented (*Figure 3g–i*). Third, while cell delaminations are spatially regulated and numerous ($\sim2.5\ 10^3$), their CE rates ($\mathbf{A}$) are negligible, and their effect is mostly isotropic (*Figure 3j* and *Figure 3—figure supplements 1e*, *2e*). Therefore, tissue CE mostly occurs through oriented divisions, rearrangements and cell shape changes. Together this shows how our formalism enables to express quantitatively the relevant information extracted from millions of cell contours over space and time and to separately measure and represent the CE rates of the tissue and of each cell process at the scale of groups of cells in a whole epithelial tissue.

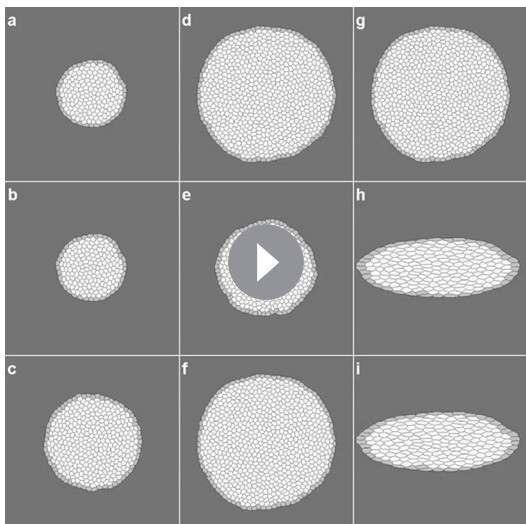

**Video 2.** Movies of computer simulations corresponding to (a-d) *Figure 2a–d* and to (e–i) *Figure 2—figure supplement 1a–e*. (a) Cell size and shape changes **S**, (b) oriented cell divisions **D**, (c) oriented cell rearrangements **R**, (d) delaminations **A**, (e) integrations of new cells **N**, (f) fusions of two or more cells **C**, (g) cell flux **J**, (h) rotation, (i) convergence-extension with initial and final states similar to those in (h).

## Average over multiple animals: archetypal CE rates of tissue and cell processes

In order to determine the average and the variability of cell dynamics of a tissue, we developed a general method to register in time and space, and to rescale movies obtained from different animals. This method uses reliable landmarks (for space registration) and prominent, stereotyped rotational movements (for time registration); and it can be applied to mutant conditions and to other tissues (see Materials and methods and below for mutant conditions). Upon space-time registration, each deformation rate measured in each cell patch in 5 hemi-notum movies can be averaged to yield average maps of an archetype hemi-notum representing the dynamics of ~7 $10^6$ cell contours, i.e. ~23 $10^6$ links, 40 $10^3$ divisions, 5.4 $10^3$ delaminations (*Figure 4a,b* and *Figure 4—figure supplement 1a*). For each process and in each region of the archetype hemi-notum, the corresponding biological variability is measured by quantifying its variations between hemi-nota in each region. This was used to determine whether each CE rate was significant in a given region (plotted in color or grey, respectively). Collectively, these approaches yield average maps of the tissue CE rate **G** and of the main cell process CE rates (**D**, **R**, and **A**) that can be superimposed for comparison (*Figure 4a,b*). These maps confirmed that division CE rates are parallel to the tissue deformation rates in some regions but not all (compare regions 1, 2 and 3), suggesting that oriented divisions may not have a single and simple effect in morphogenesis.

## Components of cell process CE rates and magnitude of morphogenesis

Studying the morphogenesis of epithelial tissues requires the analysis of both the alignment and the magnitude of each cell process CE rate with respect to the local tissue CE rate. This can be achieved by projecting each CE rate (**D**, **R**, **S** and **A**) onto the direction of the local tissue CE rate (noted $\mathbf{u_G}$) to determine their components along the direction of **G**, thereby yielding $\mathbf{D}_\parallel$, $\mathbf{R}_\parallel$, $\mathbf{S}_\parallel$ and $\mathbf{A}_\parallel$ components respectively (*Figure 4c*). The projection of **G** along its own direction yields the local magnitude of tissue morphogenesis ($\mathbf{G}_\parallel$). Each of these components along **G** can therefore be interpreted as the effective participation of each process in tissue morphogenesis of the wild-type tissue; it should not be confused with a functional role of a cell process that can only be studied using loss and gain of function approaches and modeling. A CE rate aligned with the tissue CE rate has a positive component along tissue morphogenesis, whereas a CE rate displaying a bar orthogonal to the tissue CE rate has a negative component along tissue morphogenesis (*Figure 4c*). In a tissue where tissue deformation is solely associated with cell divisions, cell rearrangements, cell size and shape changes and apoptoses, **G** is equal to the sum of the CE rate of divisions (**D**), rearrangements (**R**) and cell shape changes (**S**), as well as apoptoses (**A**). The magnitude of local tissue morphogenesis ($\mathbf{G}_\parallel$) can therefore also be expressed as the sum of the local components of each cell process along **G**, namely:

$$\mathbf{G}_\parallel = \mathbf{D}_\parallel + \mathbf{R}_\parallel + \mathbf{S}_\parallel + \mathbf{A}_\parallel \qquad (2)$$

Maps of the components associated with each cell process can be then plotted using circles that are hollow or filled for positive or negative components, respectively, and the areas of which scale with the magnitude of each components (*Figure 4d–g* and *Figure 4—figure supplement 1b,c*).

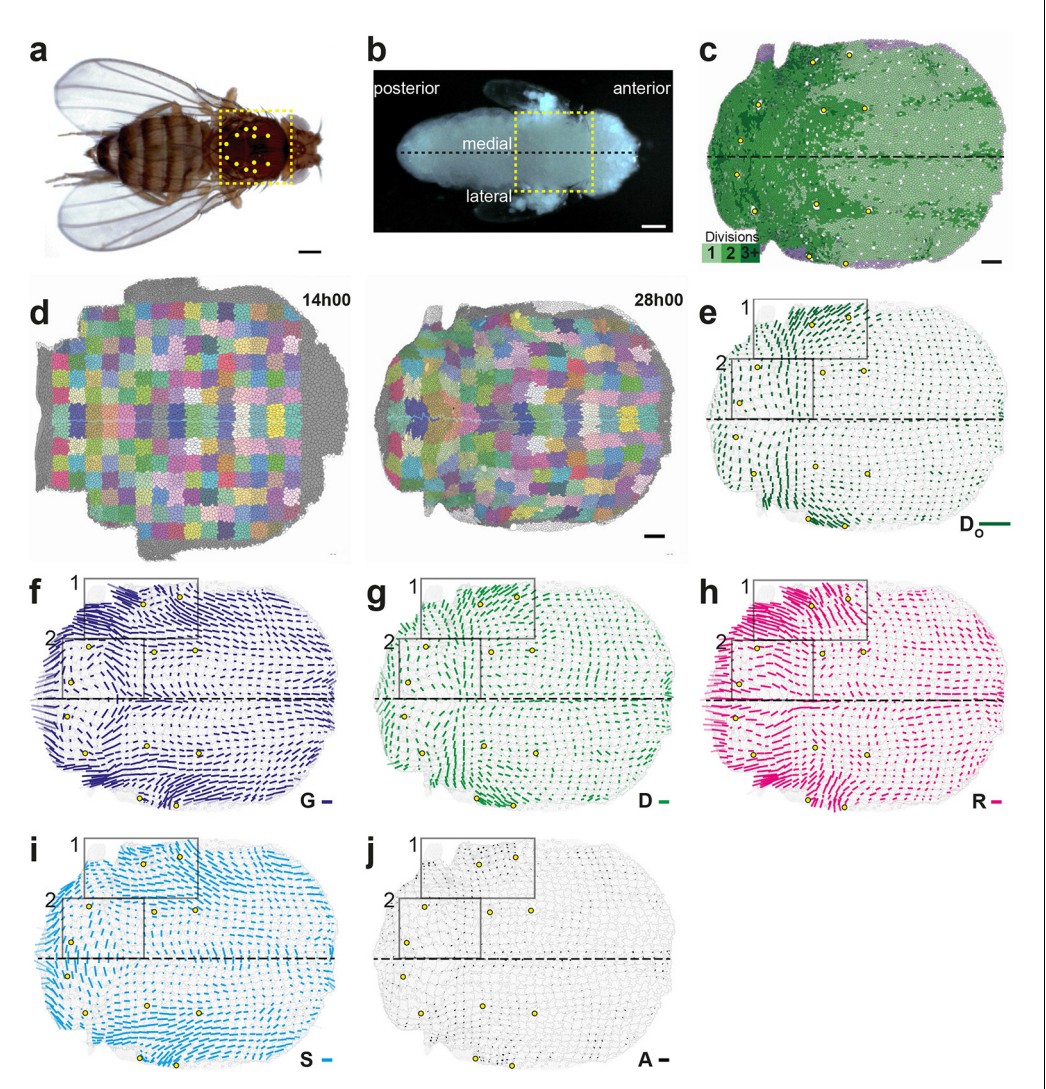

**Figure 3.** Quantitative characterization of tissue morphogenesis of the whole Drosophila notum. (**a**) Drosophila adult fly. Yellow dotted box is the notum, circles filled in yellow are macrochaetae. (**b**) Drosophila pupa. Yellow dotted box is the region that was filmed. Black dotted line is the midline (along the anterio-posterior direction, mirror symmetry line for the medial-lateral axis). (**c**) Rate of cell divisions obtained from cell tracking. Number of cell divisions color-coded on the last image of the movie (28 hAPF), light green cell: one division; medium green cell: two divisions; dark green cells: three divisions and more; purple cells: cells entering the field of view during the movie. Circles filled in yellow indicate macrochaetae. The other white cells are microchaetae. (**d**) Growth and morphogenesis of cell patches during notum development. (Left) Cell contours (thin grey outlines) at 14 hAPF. A grid made of square regions of 40 µm sides was overlayed on the notum to define patches of cells whose centers initially lied withing each region (~30 cells per patch in the initial image). Within each patch, all cells (and their future offspring) were assigned a given color. The assignment of patch colors was arbitrary but nevertheless respected the symmetry with respect to the midline (in cyan) to make easier the pairwise comparison of patches. Each patch was then tracked as it deformed over time to visualize tissue deformations at the patch scale. (Right) Cell contours at 28 hAPF. The variety of patch shapes reveals the heterogeneity of deformations at the tissue scale, as well as their striking symmetry with respect to the midline. (**e**) Map of average cell division orientation (bar direction) and anisotropy (bar length), $\mathbf{D_o}$ (Appendix C.3.2). Its determination is solely based on the links between newly appeared sister cells (link in dark green in *Figure 1a,b*). (**f**–**j**) Maps of orientation (bar direction) and anisotropy (bar length) of CE rates, for (**f**) the tissue $\mathbf{G}$ (compare the bar amplitude and orientation pattern with the pattern of patches in (d) right), (**g**) cell divisions $\mathbf{D}$, (**h**) cell rearrangements $\mathbf{R}$, (**i**) cell shape changes $\mathbf{S}$ and (**j**) delaminations $\mathbf{A}$. In this Figure (and *Figure 3—figure supplements 1* and *2*), measurements over the whole notum have been averaged over 14 h of development (between 14 and 28 hAPF) and plotted on the last image of

*Figure 3 continued on next page*

*Figure 3 continued*

the movie (for their time-evolution see *Video 4*); contours of cells (thin grey outlines) and of initially square patches (thick grey outlines); black boxes outline the posterior regions (medial and lateral) described in the text; patches near the tissue boundary contain less data and are plotted accordingly with higher transparency; circles filled in yellow indicate macrochaetae; dashed black line is the midline. Scale bars: (a,b) 250 µm, (c,d) 50 µm, (e–j) 2 $10^{-2}$ $h^{-1}$.

The following figure supplements are available for figure 3:

**Figure supplement 1.** Complete set of maps of dilation rates (isotropic parts of measurements).

**Figure supplement 2.** Complete set of maps of contraction-elongation (CE) rates (anisotropic parts of measurements).

We briefly illustrate here how the component measurements can further be analyzed to uncover novel interactions between tissue morphogenesis, cell divisions and other cell processes. The measurements show that the cell division component (*Figure 4e*) can either be strongly positive in some regions (regions 1 and 2) or strongly negative in others (region 3). In contrast, cell rearrangements and cell shape changes mainly have positive components along tissue morphogenesis (*Figure 4f,g*). In regions 1 and 2, the positive component of cell divisions corresponds to the one described in the literature, namely that cell divisions and tissue elongation are oriented in the same direction. Conversely, our measurements in region 3 show that cell divisions have a negative component corresponding to divisions taking place mostly orthogonally to the direction of tissue elongation. The measurements of cell rearrangements and cell shape components show that the elongation of the tissue in this region results from the strongly positive components of cell shape changes or of cell rearrangements (*Figure 4f,g*). Finally in region 4, although most cells divide once, the cell division component is small relative to that of cell shape changes that almost completely accounts for tissue deformation in that region (*Figure 4d–g*).

The existence of these distinct relationships between process CE rates and total CE rate can be further analyzed by plotting each relative component versus time to characterize its temporal evolution throughout tissue development (*Figure 4h–k*). In addition to the spatial heterogeneity of morphogenesis, these plots reveal that morphogenesis also strongly varies over time in a given region of the tissue and that it can be further decomposed in different periods (see *Figure 4h–k* legends for details). More generally, such analyses will provide important insight about the respective roles of each cell process in tissue morphogenesis and their time evolution.

## Quantitative characterization of morphogenesis in a pupal wing

To illustrate the generality of our approach and to determine whether cell divisions can be oriented perpendicularly to the tissue CE rate in another tissue, we performed a similar characterization of the morphogenesis in another epithelium, the pupal wing (*Figure 5*, *Figure 5—figure supplement 1*, *Video 5a*). The important deformations of cell patches over time show that wing morphogenesis is heterogeneous as well, displaying strong tissue deformations in the anterior and posterior domains of the wing, and milder deformations in the medial domain (*Figure 5a*, *Video 5b*). To characterize quantitatively these deformations of the tissue and relate them to cellular processes occurring in this wing, we applied our formalism to this pupal wing between 15 and 32 hAPF and determined maps of averaged tissue CE rates $\mathbf{G}$ and cell processes CE rates $\mathbf{D}$, $\mathbf{R}$, $\mathbf{S}$ and $\mathbf{A}$ (*Figure 5b–e*), as well as their projections onto $\mathbf{G}$ direction, $\mathbf{G}_{\parallel}$, $\mathbf{D}_{\parallel}$, $\mathbf{R}_{\parallel}$, $\mathbf{S}_{\parallel}$ and $\mathbf{A}_{\parallel}$ (*Figure 5f–i*). Even on a single wing, all CE rate maps display heterogeneous but smooth patterns over time, showing that this scale of space-time averaging ($40 \times 40$ µm$^2$, 2 h) is also suitable for the wing (*Video 5c–f*). These measurements confirmed the observed heterogeneity in tissue morphogenesis by evidencing major tissue CE rates in anterior and posterior domains of the wing, with a tilt of about $\pm 45°$ with respect to the horizontal axis, respectively, and smaller CE rates in the medial domain (*Figure 5b,f*). The CE rates of cell processes $\mathbf{D}$, $\mathbf{R}$, $\mathbf{S}$ display smooth and heterogeneous maps while displaying different patterns in space (*Figure 5c–e*), while the CE rate of $\mathbf{A}$ is negligible (*Figure 5—figure supplement 1b*). Like in the dorsal thorax, rearrangements and cell shape changes

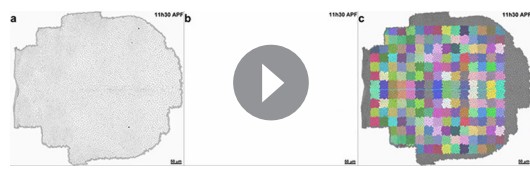

**Video 3.** Movies of cell tracking, cell trajectories and patch deformation in the whole *Drosophila* notum. (**a**) Cell tracking displaying divisions (see *Figure 3c*): cells are color coded in light green for the first division, medium green for two divisions, dark green for three divisions and more; black for the last five frames before a delamination; red for cells that fuse; grey for the first layer of boundary cells; purple for new cells. (**b**) Cell trajectories: as it moves, each cell leaves a trail corresponding to the successive positions of its center in the last 10 images. Same color code as in (a). (**c**) Growth and morphogenesis of cell patches during development (see *Figure 3d*). Cell contours are indicated by thin grey outlines, cells within a patch have same arbitrarily assigned colors that respect the symmetry with respect to the midline. Movie between 11:30 and 30:45 hAPF. Note that the original movie of the E-Cad:GFP tissue is visible in Supplementary Video 1 of (*Bosveld et al., 2012*).

mainly have positive components along tissue CE rate and they make up most of tissue morphogenesis (*Figure 5f,h,i*). Like in the dorsal thorax, oriented cell divisions in the wing display more variety in their component along tissue morphogenesis than **R** and **S**: **D** has positive components along **G** in anterior and posterior wing domains, where tissue morphogenesis is the strongest, while it has negative components in the medial wing domain, where morphogenesis is milder (*Figure 5c,g* box). Together our analyses of morphogenesis in the notum and the wing illustrate how our formalism enables the quantitative characterization of morphogenesis in various epithelial tissues.

## Quantitative comparison of wild-type and mutant conditions

We found that in the wild-type tissue cell divisions display various orientations with respect to the direction of tissue elongation and thus can have negative and positive components along tissue morphogenesis. These observations raise important questions regarding the role of cell divisions per se in tissue development, namely the role of proliferation and division orientation, as well as its interplay with the other cell processes during tissue morphogenesis. We illustrate here how the formalism can help analyze these central questions by allowing for a rigorous quantification of different experimental conditions.

To experimentally study the role of cell divisions in morphogenesis, we overexpressed the tribbles gene (*trbl$^{up}$*), an inhibitor of G2/M transition (*Grosshans and Wieschaus, 2000*; *Mata et al., 2000*; *Seher and Leptin, 2000*) using the Gal4/Gal80$^{ts}$ system to inhibit proliferation specifically at pupal stage (*McGuire et al., 2003*; *Bosveld et al., 2012*). We segmented and tracked five *trbl$^{up}$* hemi-thoraxes over time (corresponding to ~3.7 10$^6$ cell contours). Both visual inspection of the movie and cell tracking revealed that a *trbl$^{up}$* hemi-notum hardly displays any division as compared to wild-type tissue: ~1.7 10$^3$, i.e. only 4.3% of the number of wild-type divisions (*Figure 6a,b*). We then registered and rescaled in space the five hemi-thoraxes, synchronized them in time, and applied the formalism to determine tissue morphogenesis and the respective CE rates of each cell process (*Figure 6d,f* and *Figure 6—figure supplement 1b,d*). As expected, the measured division CE rate **D** nearly vanishes in accordance with the nearly complete disappearance of cell proliferation (*Figure 6c,d*). Furthermore, we find that tissue CE rate **G** is disrupted in *trbl$^{up}$* tissue, both in direction and amplitude, suggesting that the absence of proliferation impacts tissue morphogenesis (*Figure 6e,f*).

Two complementary maps can be used to quantitatively study the effects of *trbl* overexpression: the difference between the tissue CE rates in *trbl$^{up}$* tissue (**G**$_{trbl}$) and in wild-type tissue (**G**$_{wt}$), namely Δ**G** (*Figure 6g*), and its

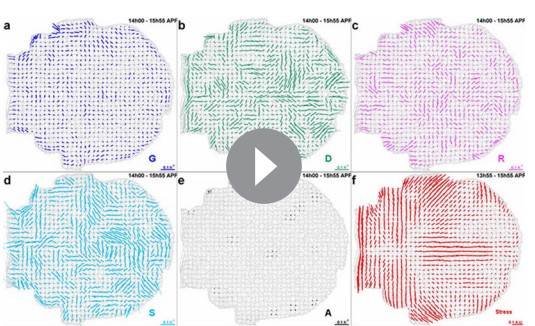

**Video 4.** Time evolution of the cell process CE rates and of the anisotropic junctional stress over the whole *Drosophila* notum. (**a-e**) Time evolution of CE rates of (a) tissue morphogenesis **G**, (b) oriented cell divisions **D**, (c) oriented cell rearrangements **R**, (d) cell shape changes **S**, (e) delaminations **A** (see also *Figure 3f-j*). (f) Time evolution of the anisotropic part of the junctional stress $\sigma$ (see also *Figure 7*). Movie between 14 and 28 hAPF, 2 h sliding window average plotted every 5 min.

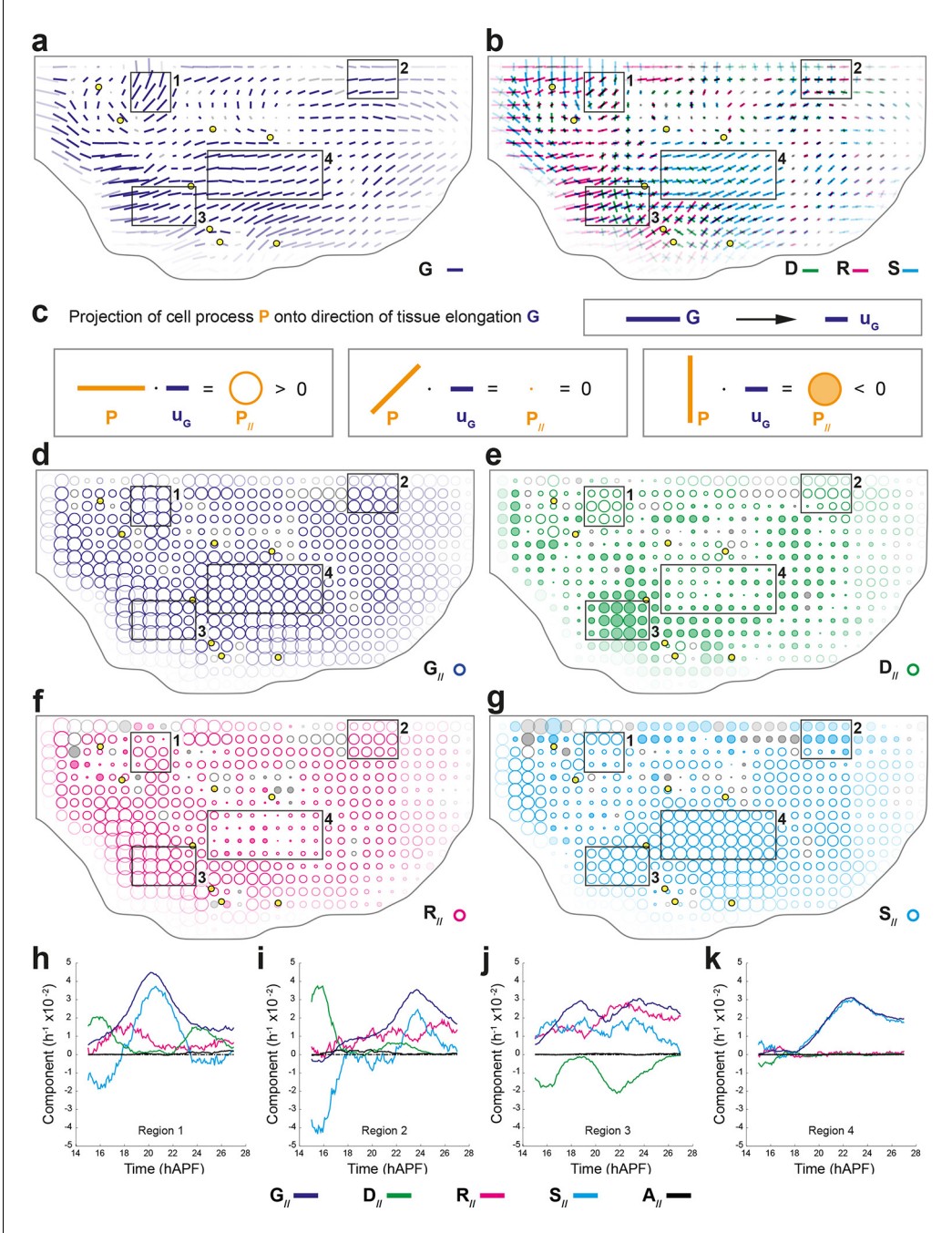

**Figure 4.** Quantifications of tissue and cell process deformations during tissue development, averaged over 5 hemi-nota. (**a,b**) Average maps of CE rates of (**a**) the tissue (**G**, dark blue) and of (**b**) cell divisions (**D**, green), cell rearrangements (**R**, magenta) and cell shape changes (**S**, cyan) overlayed for comparison. Time averages were performed between 14 and 28 hAPF. Scale bars: 2 10$^{-2}$ h$^{-1}$. In this Figure (and *Figure 4—figure supplement 1*), values larger than the local biological variability are plotted in color while smaller ones are shown in grey; a local transparency is applied to weight the CE rate according to the number of cells and hemi-nota in each group of cells; black outline delineates the archetype hemi-notum; the midline is the top boundary; circles filled in yellow are archetype macrochaetae. Black rectangular boxes outline the four regions numbered 1 to 4 described in the text. (**c**) Projection of a cell process along the local axis of tissue elongation: cell process component. Here, $\mathbf{u_G} = \mathbf{G}/\|\mathbf{G}\|$ is unitary and has the direction of the local tissue CE rate **G**: its bar therefore indicates the local direction of tissue elongation (*Equation 16*). For each cell process measurement **P**, its component along **G** can be determined by its projection onto $\mathbf{u_G}$ and is noted $\mathbf{P}_{/\!/}$ (*Equation 17*). It is expressed as a rate of change per unit

*Figure 4 continued on next page*

*Figure 4 continued*

time, i.e. in hour$^{-1}$, and is represented as a circle whose *area* is proportional to the component. (Left) If a cell process CE rate $\mathbf{P}$ (here orange bar) is rather parallel to $\mathbf{G}$ (dark blue bar), it has a positive component $\mathbf{P}_{/\!/}$ on $\mathbf{G}$ (orange empty circle). (Middle) If the $\mathbf{P}$ bar is at 45° angle to $\mathbf{G}$, its component $\mathbf{P}_{/\!/}$ on $\mathbf{G}$ is zero. (Right) If the CE rate $\mathbf{P}$ bar is rather perpendicular to $\mathbf{G}$, its CE rate has a negative component $\mathbf{P}_{/\!/}$ on $\mathbf{G}$ (orange full circle). Note that the component of tissue CE rate along itself, $\mathbf{G}_{/\!/}$, is the amplitude of tissue CE rate $\|\mathbf{G}\|$, and is positive by construction. The additivity of cell process CE rates to $\mathbf{G}$ (*Equation 1*) implies the additivity of these components to $\mathbf{G}_{/\!/}$ (*Equation 2*). Note also that these circles have a different meaning from those used to represent the dilation rates (*Figure 1d*). (d–g) Maps of components along the tissue CE rate $\mathbf{G}$ for (d) the tissue rate itself ($\mathbf{G}_{/\!/}$, dark blue, components are positive by construction) and of (e) cell divisions ($\mathbf{D}_{/\!/}$, green), (f) cell rearrangements ($\mathbf{R}_{/\!/}$, magenta) and (g) cell shape changes ($\mathbf{S}_{/\!/}$, cyan). Same representation as in (a,b). Scale bars: 0.1 h$^{-1}$. (h–k) Time evolution of components along the tissue CE rate $\mathbf{G}$ of the tissue rate itself ($\mathbf{G}_{/\!/}$, dark blue) and of cell divisions ($\mathbf{D}_{/\!/}$, green), cell rearrangements ($\mathbf{R}_{/\!/}$, magenta), cell shape changes ($\mathbf{S}_{/\!/}$, cyan) and delaminations ($\mathbf{A}_{/\!/}$, black) in four regions of interest. Measurements are averaged over sliding time windows of 2 h and spatially over the region (h) 1, (i) 2, (j) 3, (k) 4. In region 1, one can distinguish three phases where tissue morphogenesis mostly occurs via: (14–18 hAPF) cell rearrangements as divisions and cell shape changes nearly cancel out; (18–23 hAPF) cell shape changes, and it reaches its peak; (23–26 hAPF) oriented cell divisions as cell rearrangements and cell shape changes cancel out. In region 2, the same temporal phases can be distinguished: (14–18 hAPF) the effect of divisions is almost cancelled by cell shape changes; (18–22 hAPF) only weak changes occur; (22–26 hAPF) cell rearrangements and cell shape changes add up and morphogenesis becomes significant. In region 3, like in region 1, the two waves of oriented cell divisions can be seen clearly with $\mathbf{D}_{/\!/}$ peaks occurring around 16 and 23 hAPF, but here both division waves have a negative component along tissue CE rate. Cell rearrangements and cell shape changes make up for the negative sign of divisions, thereby mostly accounting for tissue morphogenesis in this region. In region 4, from about 18 hAPF onwards, the tissue CE rate significantly increases and almost perfectly overlaps with cell shape change CE rate, meaning that tissue morphogenesis solely occurs via cell shape changes.

The following figure supplement is available for figure 4:

**Figure supplement 1.** Maps of delamination CE rate and overlay of maps of main cell process components along the tissue CE rate.

---

projection onto the wild-type tissue CE rate, namely $\Delta\mathbf{G}_{/\!/}$ (*Figure 6h*). $\Delta\mathbf{G}$ represents the change brought to wild-type tissue morphogenesis by the *trbl* overexpression, and $\Delta\mathbf{G}_{/\!/}$ measures the effective contribution of this change to wild-type tissue morphogenesis. A region where $\Delta\mathbf{G}_{/\!/}$ is positive means that wild-type tissue morphogenesis has been increased by *trbl* overexpression, and conversely, $\Delta\mathbf{G}_{/\!/}$ is negative means that it has been decreased in this region. Therefore, the $\Delta\mathbf{G}_{/\!/}$ map provides a visual representation of where the tissue morphogenesis is increased or reduced and can be interpreted as the role of cell divisions (proliferation and oriented divisions) during tissue morphogenesis. This map reveals that in almost all regions of the tissue, and regardless of the orientation of cell divisions relative to tissue elongation in wild-type, overexpressing *trbl* reduces wild-type tissue elongation (full circles in *Figure 6h*).

A similar approach can be applied to each cell process to determine how it is impacted by the *trbl* overexpression (*Figure 6h–j* and *Figure 6—figure supplement 1*). Thus, the respective changes in each cell process due to *trbl* overexpression can be measured using $\Delta\mathbf{D}_{/\!/}$, $\Delta\mathbf{R}_{/\!/}$, and $\Delta\mathbf{S}_{/\!/}$ ($\Delta\mathbf{A}_{/\!/}$, not shown). As expected from the nearly complete absence of division in *trbl*$^{up}$ tissue, the $\Delta\mathbf{D}_{/\!/}$ map representing the changes in tissue morphogenesis due to the loss of cell division is almost the opposite of the $\mathbf{D}_{/\!/}$ map (compare *Figure 4e* and *Figure 6—figure supplement 1e*). More interestingly, the $\Delta\mathbf{R}_{/\!/}$, and $\Delta\mathbf{S}_{/\!/}$ maps directly demonstrate that both cell rearrangements and cell shape changes are significantly modified in *trbl*$^{up}$ tissue and contribute to the overall changes in tissue morphogenesis due to *trbl* overexpression (*Figure 6i–j*). This indicates that suppressing proliferation not only makes oriented division CE rate vanish, but also has an indirect impact on both cell rearrangements and cell shape changes. In conjunction with our results in the wild-type tissue, this suggests that both cell proliferation and the orientation of divisions determine the morphogenesis of the tissue, and that a complex interplay exists between cell divisions and the other processes such as cell shape changes and rearrangements.

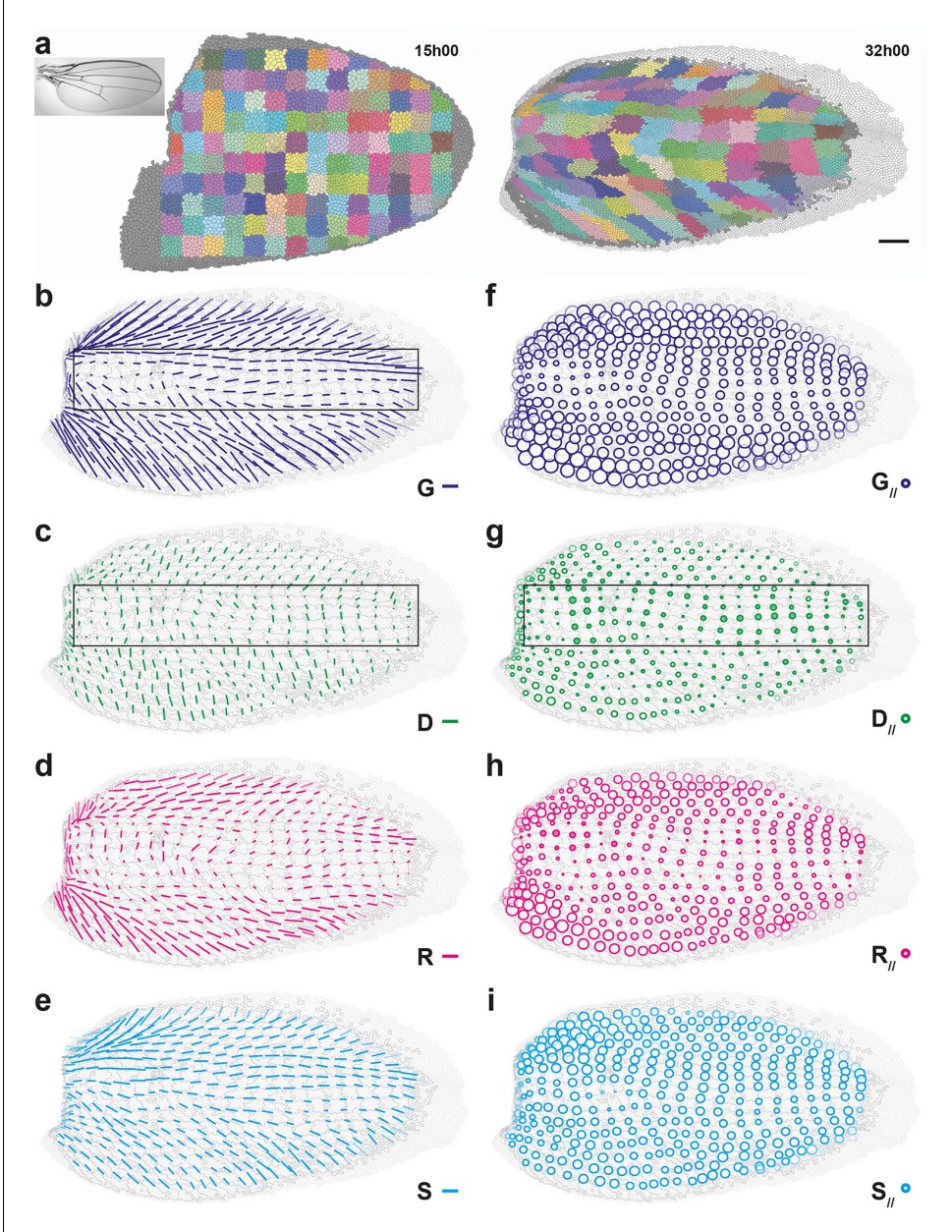

**Figure 5.** Quantitative characterization of pupal wing morphogenesis. (a) Growth and morphogenesis of cell patches during wing development. (Left) Cell contours (thin grey outlines) at 15 hAPF. A grid made of square regions of 34 µm side was overlayed on the wing, to define patches of cells whose centers initially lied withing each region (~30 cells per patch in the initial image). Within each patch, all cells (and their offspring) were assigned the same arbitrary color. Each patch was then tracked as it deformed over time to visualize tissue deformations at the patch scale. Inset: Drosophila adult wing. (Right) Cell contours at 32 hAPF. In this tissue as well, the variety of patch shapes reveals the heterogeneity of deformations at the tissue scale. (b–i) Average maps of main cell process CE rates (b–e) and of their components along tissue CE rate (f–i) for the CE rates of the tissue (**G**, dark blue, b,f), cell divisions (**D**, green, c,g), cell rearrangements (**R**, magenta, d,h) and cell shape changes (**S**, cyan, e,i). Black rectangular box in (b,c,g) outlines the region described in the text. Time averages were performed between 15 and 32 hAPF. A local transparency is applied to weight the CE rate according to the number of cells in each group of cells. Scale bars: (a) 50 µm, (b–e) 0.1 h$^{-1}$; scale circle area (f–i) 0.1 h$^{-1}$.

The following figure supplement is available for figure 5:

**Figure supplement 1.** Maps of proliferation and of additional cell process CE rates during wing development.

To better understand this last point, and more generally the effect of oriented cell divisions, we used computer simulations. We compared our previous simulation of cell divisions oriented along the horizontal axis of tissue elongation (*Figure 2b* and *6k* and *Video 6a*) with simulations where only the pattern of divisions has been modified in two distinct ways: (i) we aligned all cell divisions along the vertical axis, namely orthogonally to tissue elongation, thereby leading to a negative component of $\mathbf{D}$ ($\mathbf{D}_{/\!/} < 0$) (*Figure 6l* and *Video 6b*), thus mimicking our observation in region 3 of the wild-type tissue (*Figure 4e*); (ii) we suppressed divisions ($\mathbf{D}_{/\!/} = 0$, *Figure 6m* and *Video 6c*), mimicking our observation in *trbl^up* tissue (*Figure 6d*). In both cases, we found that modifying the pattern of divisions impacts simultaneously $\mathbf{G}$, $\mathbf{R}$ and $\mathbf{S}$ in addition to $\mathbf{D}$ (*Figure 6l,m*). When divisions are orthogonal to tissue elongation, cell rearrangements, and to a lesser extent cell shape changes, are greatly increased along the direction of deformation, but they only partly compensate the CE rate of horizontally oriented divisions in the initial simulation, thereby resulting in reduced tissue elongation (*Figure 6l*). When divisions are suppressed, cell rearrangements and cell shape changes are moderately increased along the direction of deformation, and compensate even less horizontally oriented divisions, thereby resulting in further reduced tissue elongation (*Figure 6m*). These two simulations therefore recapitulate two aspects of our experimental observations: (i) how divisions orthogonal to the tissue CE rate in wild-type have a negative component along tissue morphogenesis, as found in some regions of the wild-type tissue (*Figure 4e*, region 3); (ii) how divisions, regardless of their orientation, can facilitate tissue elongation by indirectly impacting cells rearrangements and cell shape changes, as observed in *trbl^up* tissue where proliferation is severely reduced and tissue deformations are globally decreased (*Figure 6h*).

Altogether, our formalism reveals the extent of the heterogeneity of division orientation in a tissue, and our analyses of simulations and *trbl^up* experimental condition show that both cell proliferation and oriented divisions can influence tissue morphogenesis. Lastly, our formalism provides a unified approach to independently quantify each cell process, thus revealing a complex interplay between cell divisions, cell rearrangements and cell shape changes and providing a rigorous framework for its future characterization using both mutant conditions and modeling.

## Interplay between tissue elongation, cell division orientation and junctional stress

Epithelial tissue growth and morphogenesis is regulated by mechanical stress (*Heisenberg and Bellaïche, 2013*). To provide a complete set of methods to study tissue development, we therefore aimed to combine our formalism with the measurement of mechanical stress due to tension in adherens junctions. This 'junctional stress' gathers all forces (regardless of their biological origin, including cortical and cytoplasmic forces) transmitted between cells via adherens junctions. The relevance of junctional stress quantification to understand tissue development has been demonstrated by methods such as laser ablation (for review see [*Rauzi and Lenne, 2011*]) or optical trapping of cell junction (*Bambardekar et al., 2015*). However, with these methods, it is difficult to obtain spatial and temporal stress maps at the scale of the whole tissue.

Others and we have previously developed force inference approaches to quantify junction stress from segmented images independently of possible external forces such as a friction of the epithelium on an outer layer (*Brodland et al., 2010*; *2014*; *Ishihara and Sugimura, 2012*; *Chiou et al., 2012*; *Ishihara et al., 2013*; *Sugimura and Ishihara, 2013*). We improved our method to make it numerically more robust and efficient, thereby enabling the determination of cell pressures, junction tension and junctional stress over the whole tissue (see Materials and methods, *Figure 7—figure supplement 1*).

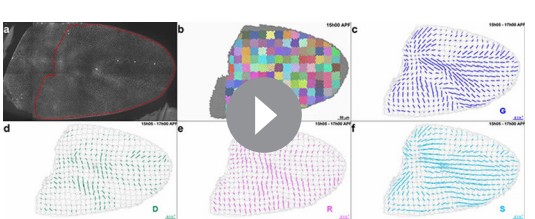

**Video 5.** Growth and morphogenesis during wing development, 15 to 32 hAPF. (**a**) Tissue labeled with E-Cad:GFP and imaged by multi-position confocal microscopy at a 5 min time resolution. (**b**) Growth and morphogenesis of cell patches during development (see *Figure 5a*). Cell contours are indicated by thin grey outlines, patch contours by arbitrarily assigned colors. (**c**–**f**) Time evolution of CE rates of (**c**) the tissue $\mathbf{G}$, (**d**) oriented cell divisions $\mathbf{D}$, (**e**) oriented cell rearrangements $\mathbf{R}$, (**f**) cell shape changes $\mathbf{S}$ (see *Figure 5b–e*).

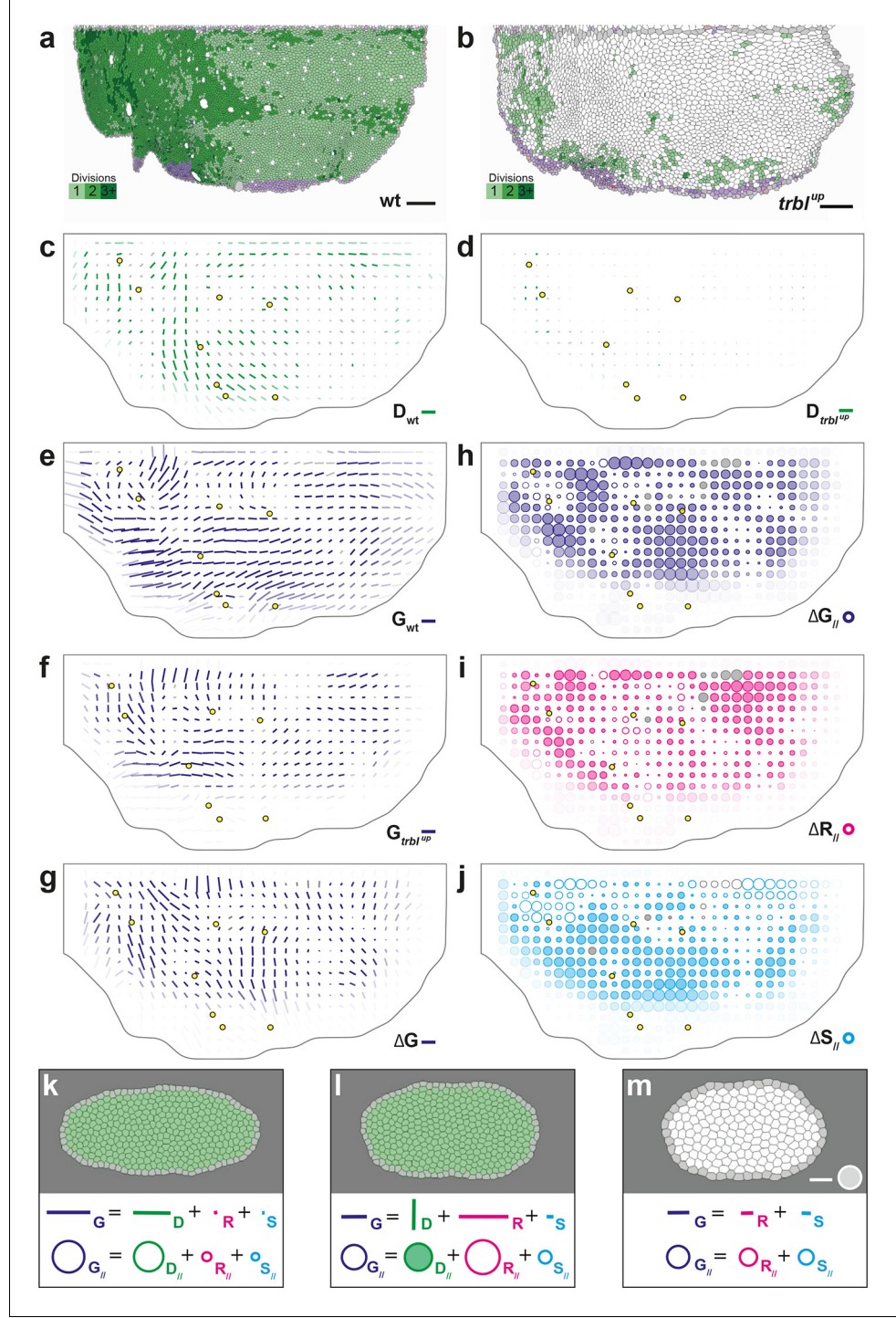

**Figure 6.** Quantitative characterization of tissue development in *trbl^up* mutant notum. (a,b) Comparison between rate of cell divisions in (a) wild-type (extracted from half of *Figure 3c*) and (b) *trbl^up* tissues. Number of cell divisions color-coded on the last image of the movie (28 hAPF), light green cell: one division; medium green cell: two divisions; dark green cells: three divisions and more; purple cells: cells entering the field of view during the movie. (c–j) Comparisons of CE rates in wild-type and *trbl^up* mutant tissues. Time averages were performed between 14 and 28 hAPF. In this Figure (and *Figure 6—figure supplement 1*), values larger than the local biological variability are plotted in color while smaller ones are shown in grey; a local transparency is applied to weight the CE rate according to the number of cells and hemi-nota in each group of cells; black outline delineates the archetype hemi-notum; the midline is the top boundary; circles filled in yellow indicate archetype

*Figure 6 continued on next page*

*Figure 6 continued*

macrochaetae. (c,d) Comparison between cell division CE rate (**D**, green) in (c) wild-type (extracted from *Figure 4b*) and (d) *trbl^{up}* tissues. (e,f) Comparison between tissue CE rate (**G**, dark blue) in (e) wild-type (reproduced from *Figure 4a*) and (f) *trbl^{up}* tissues. (g–j) Subtraction of measurements in *trbl^{up}* tissue minus measurements in wild-type tissue, for (g) tissue CE rate ($\Delta$**G**, dark blue), and for components along wild-type tissue CE rate of (h) tissue CE rate ($\Delta$**G**$_{/\!/}$, dark blue), (i) cell rearrangements CE rate ($\Delta$**R**$_{/\!/}$, magenta) and (j) cell shape changes CE rate ($\Delta$**S**$_{/\!/}$, cyan). (k–m) Simulations illustrating the impact of cell divisions on tissue elongation and on the other processes. Top: last image of Potts model simulations; bottom: measurement of CE rates (bar) and of their components along **G** (circles). (k) As in *Figure 2b*, numerically implemented forces elongate the cell patch along the horizontal axis, and cell divisions are oriented along the direction of patch elongation; (l) same as (k) but with divisions now oriented orthogonally to the direction of tissue elongation, and (m) without any division occurring during tissue elongation. Only non-zero values are plotted. Scale bars: (a,b) 50 μm, (c–g) 2 $10^{-2}$ $h^{-1}$, (k–m) equivalent to $10^{-2}$ $h^{-1}$ for simulation movies lasting 20 h; scale circle areas: (h–m) 0.1 $h^{-1}$.

The following figure supplement is available for figure 6:

**Figure supplement 1.** Additional quantitative characterizations of *trbl^{up}* tissue development.

---

The stress has an isotropic part related to the pressure represented by a circle (*Figure 7—figure supplement 1c*). Its anisotropic part has an amplitude and a direction of traction represented by a bar, and a direction of compression (of equal magnitude and perpendicular, the display of which is redundant). Even on a single animal, the junctional stress maps vary smoothly over time and space, and are symmetric with respect to the midline, revealing the quality of the signal-to-noise ratio (*Figure 7a*, *Figure 7—figure supplement 1c* and *Video 4f*). We then performed their ensemble average over several animals (*Figure 7b*) and compared the anisotropic part of the junctional stress maps and of the CE rate maps of the different processes measured by the formalism. Focusing here on divisions, the analysis confirms that on average cell division orientation aligns well with junctional stress orientation, even in such a heterogeneous tissue (*Figure 7—figure supplement 2*, alignment = 0.87). Moreover, the division CE rate, which is more relevant to tissue morphogenesis, is also well correlated with junctional stress orientation (*Figure 7b*, alignment = 0.73).

Taking further advantage of averaged maps of division CE rate on the one hand, and of tissue and cell process component maps on the other hand, enables to explore more finely the alignment between cell divisions and stress. In particular, we can exclude that a positive or negative component of cell divisions would be due to distinct relationships between division CE rate and stress orientations. Indeed, cell divisions have a positive component in region 1 and 2, while cell division CE rate **D** is either poorly aligned (region 1, alignment = 0.16) or well aligned (region 2, alignment = 0.97) with junctional stress orientation (*Figure 7b*). In addition to regions where stress, division CE rate and tissue elongation are well aligned (region 2, *Figure 7b*, *Figure 4e*), we also find regions where, although cell divisions and junctional stress remain well aligned (region 3, alignment = 0.94, *Figure 7b*), the tissue CE rate (**G**) is almost orthogonal to divisions and stress (alignment = -0.88, *Figure 4e*), mostly occurring through cell rearrangements and cell shape changes (*Figure 4f,g*). Altogether our results illustrate how the combination of the formalism and a stress inference method enables to uncover additional interplays between cell divisions, stress and tissue elongation. This sets the stage for in-depth spatial and temporal investigations of the interactions between each cell process and mechanical stress during tissue development.

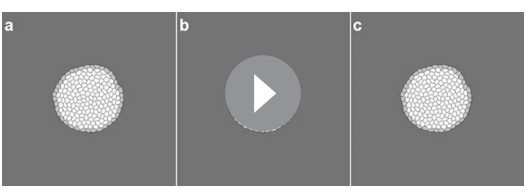

**Video 6.** Potts model simulations illustrating the impact of cell divisions on tissue elongation and on the other cell processes (see *Figure 6k–m*). (a) As in *Figure 2b*, numerically implemented forces elongate the cell patch along the horizontal axis, and cell divisions are oriented along the direction of patch elongation. (b) Same as (a) but with divisions now oriented orthogonally to the direction of tissue elongation. (c) Same as (a) but without any division occurring during tissue elongation.

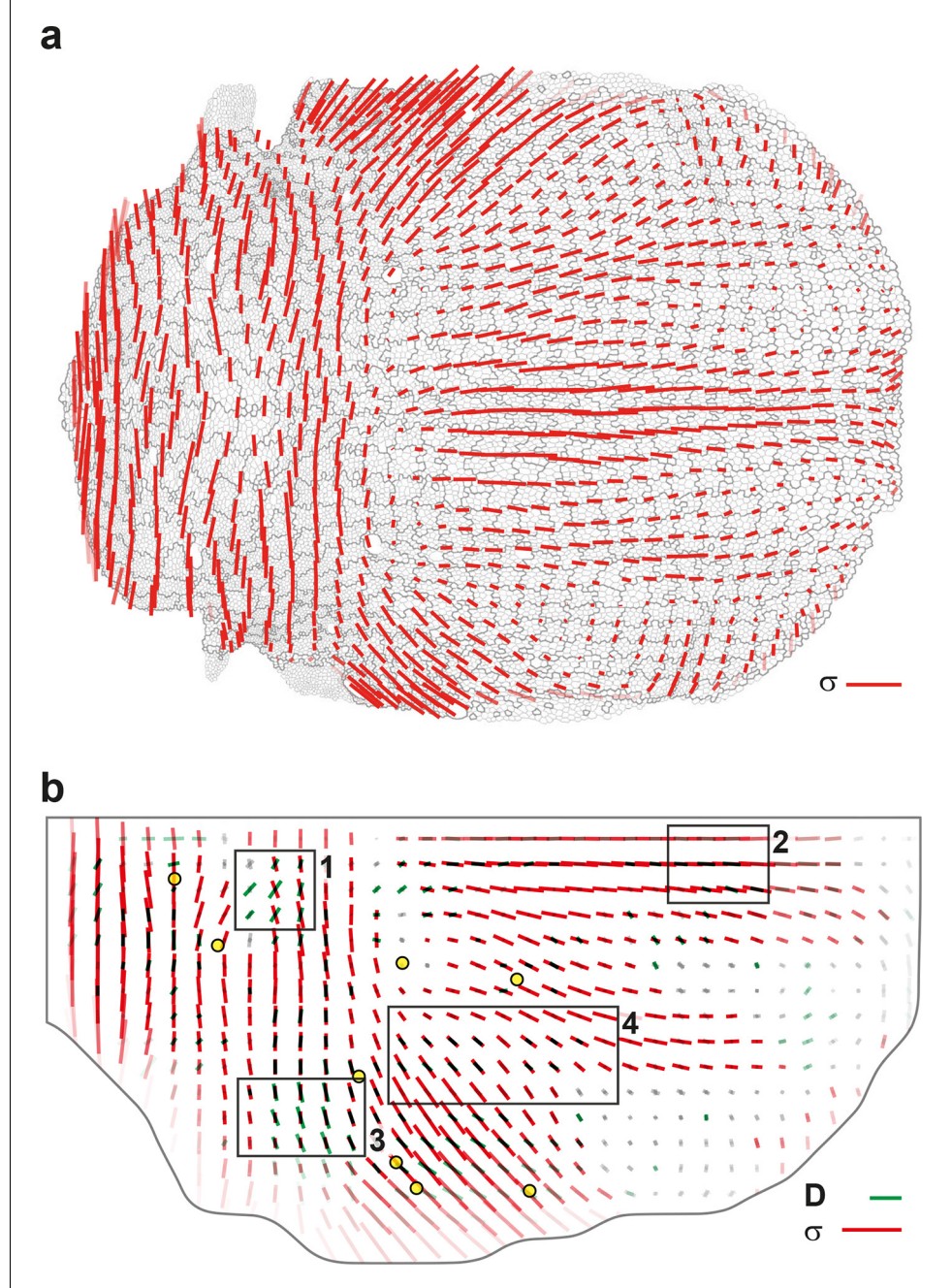

**Figure 7.** Maps of junctional stress $\sigma$ and comparison with division CE rate **D**. (a) Map of the anisotropic part of local junctional stress $\sigma$ covering the whole notum. Average performed between 14 and 28 hAPF, plotted on the last corresponding image: contours of cells (thin grey outlines) and of initially square patches (thick grey outlines). (b) Overlay of division CE rate (**D**, green) and of anisotropic part of junctional stress ($\sigma$, red). Measurements averaged over time between 14 and 28 hAPF and over 5 hemi-nota. In this Figure (and *Figure 7—figure supplement 2*), values larger than the local biological variability are plotted in color while smaller ones are shown in grey; a local transparency is applied to weight the CE rate according to the number of cells and hemi-nota in each group of cells; black outline delineates the archetype hemi-notum; the midline is the top boundary; circles filled in yellow are archetype macrochaetae. Black rectangular boxes outline the four regions numbered 1 to 4 described in the text, same as in *Figure 4*. Stress is expressed in arbitrary unit (A.U.) proportional to the average junction tension (not determined by image analysis). Scale bars: (a,b) 0.1 A.U., (b) 2 $10^{-2}$ h$^{-1}$.

The following figure supplements are available for figure 7:

*Figure 7 continued on next page*

*Figure 7 continued*

**Figure supplement 1.** Maps of cell pressures, junction tensions and junctional stress inferred on single images.
**Figure supplement 2.** Comparison of orientations of junctional stress $\sigma$ and cell division orientation $\mathbf{D_o}$.

## Conclusion

We have developed a unified multiscale formalism that relates cell and tissue behaviors to characterize the growth and morphogenesis of epithelial tissues in two and three dimensions. The formalism is free from assumptions regarding biological mechanisms, modeling or external forces and it has numerous advantages. Its unified and separate measurements of the contributions of each cell process to tissue growth and morphogenesis significantly help describe and quantify the mechanisms governing tissue development. These measurements have been validated with computer simulations. They can be easily represented on spatial and temporal maps or graphs to describe the interplay between divisions, cell rearrangements, cell shape and size changes and apoptoses, as well as the interplays between cell processes and junctional stress, thus facilitating their comparison in wild-type and mutant conditions. In combination with the recent advances in light microscopy, genetics and physical approaches, our unified framework and methods provide a basis for comprehensive analyses of the mechanisms driving tissue development.

## Materials and methods

### Movie acquisition

#### Live imaging

ubi-E-Cad:GFP (*Oda and Tsukita, 2001*) and E-Cad:GFP (*Huang et al., 2009*) were used for live imaging of apical cell contours in notum and wing pupal tissues. In all experiments the white pupa stage was set to 0 h after pupa formation (hAPF), determined with 1 h precision. For notum tissues, pupae were prepared for live imaging as described in (*David et al., 2005*). Pupae were imaged for a period of 15–26 h, starting at 11–12 hAPF, with an inverted confocal spinning disk microscope (Nikon) equipped with a CoolSNAP HQ2 camera (Photometrics) and temperature control chamber, using Metamorph 7.5.6.0 (Molecular Devices) with autofocus. Movies were acquired at 25±1°C. We took Z-stacks with 18 to 28 slices (0.5 µm/slice) to make sure we included the adherens junctions. Maximum projections of Z-stacks were obtained using a custom ImageJ routine (publicly available as the 'Smart Projector' plugin) and have been used for tissue flow analysis. Multiple position movies were stitched using a customized version of the 'StackInserter' ImageJ plugin. Filming 10 to 12 positions with 0.32 µm spatial resolution (pixel size) every 5 min yielded a tiling of a half dorsal thorax (3 movies), 24 positions yielded a tiling of the whole dorsal thorax (1 large movie, available as Supp. Video in [*Bosveld et al., 2012*]), resulting in 5 hemi-notum movies.

For *trbl^up* notum tissues, temporal control of gene function was achieved by using the temperature sensitive Gal4/GAL80^ts (*McGuire et al., 2003*). Tribbles was overexpressed using an UAS-Trbl transgene (*Grosshans and Wieschaus, 2000*) during pupal stage. Embryos and larvae were raised at 18°C. Late third instar larvae were switched to 29°C. After 24 to 30 h, pupae were examined. Those which were timed as 11±1 hAPF were mounted for live imaging at 29°C. Five hemi-notum movies were acquired.

For wing tissue, E-Cad:GFP pupa was prepared for live imaging as described in (*Classen et al., 2008*). Immersol W 2010 (Zeiss) was used instead of voltalef oil. Pupal wing was imaged for a period of 17 h at 5 min interval, starting at 15 hAPF with an inverted confocal spinning disk microscope (Olympus IX83 combined with Yokogawa CSU-W1) equipped with iXon3 888 EMCCD camera (Andor), an Olympus 60X/NA1.2 SPlanApo water-immersion objective, and temperature control chamber (TOKAI HIT), using IQ 2.9.1 (Andor). Other details of treatment and quantification of wing data will be published elsewhere.

## Movie quantification

### Image cross-correlation

The local tissue flow within notum tissues was first estimated by image cross-correlation velocimetry along sequential images using customized Matlab (*Bosveld et al., 2012*) routines based on the particle image velocimetry (*Rael et al., 2007*) toolbox, matpiv (http://folk.uio.no/jks/matpiv). Each image correlation window was a 32×32 pixels square (~10×10 μm$^2$, pixel size 0.32 μm), had a 50% overlap with each of its neighbors. The resulting estimate of the velocity field was used to initiate the tracking procedure, and also for time registration of different movies.

### Segmentation and tracking

Images were denoised with the Safir software (*Boulanger et al., 2010*). Cell contours were determined and individual cells were identified using a standard watershed algorithm. Errors were corrected through several iterations between manual and automatic rounds. Automatic rounds consisted of a custom automatic software (Matlab) which tracked segmented cells through all images and detected each event where two cells fuse (*Bosveld et al., 2012*). The tracking relied on the comparison between cells in one image and the following, based on overlap of cells as well as centroid positions. Divisions were detected and we checked the sizes and elongation of daughter cells. Delaminations were characterized by cell size decrease before cell disappearance, and distinguished from fusions. A fusion between times $t$ and $t + \delta t$ is an artifact which almost always reveals an error of segmentation which is either a false positive cell junction at time $t$, or a cell junction missing at time $t + \delta t$. Both by manual tests on small subsamples, and by automatic tracking of false cell appearances or disappearances, we estimated the final relative error rate to be below $10^{-4}$, which was sufficient for the analyses presented here. The whole notum movie contains ~8.8 $10^3$ cells at the beginning. After one to three divisions per cell, several delaminations, and a flow of cells out of the field of view, the final image contains ~18.4 $10^3$ cells. Altogether, on the five hemi-notum movies, ~7.7 $10^6$ cell contours were segmented and tracked. For *trbl$^{up}$* tissues, there were ~3.7 $10^6$ cell contours. Cells were larger and easier to segment, resulting in errors even smaller than in wild-type tissues (data not shown). Cells in the anteriormost part get out of focus due to tissue flow and elongation, so that we cannot track them throughout the whole movie; similarly, on lateral sides, some cells become visible during the movie (purple cells in *Video 3*). We observed a total of ~40 $10^3$ divisions and ~5 $10^3$ delaminations in the wild-type tissues, ~1.7 $10^3$ divisions and ~7 $10^2$ delaminations in *trbl$^{up}$* tissues. This yielded maps of cell apical surface area and shape, cell centroid displacements (*Videos 1* and *3*), cell lineages, and neighbour changes.

For wing tissue, ~2 $10^6$ cell contours were segmented and tracked. The segmentation was less manually corrected than in the notum. Errors in segmentation or tracking resulted in cell fusions, cell integrations and fluxes; their measured values were small enough that the measurement for other cell processes were not significantly affected. Interestingly, this shows the robustness of our formalism with respect to errors in input data. Altogether, ~13 $10^6$ cell contours (~40 $10^6$ cell-cell junctions) were analyzed.

### Force and stress inference

In epithelial tissues where cells are confluent, assuming mechanical equilibrium, cell shapes are then determined by cell junction tensions $\gamma$ and cell pressures $p$, and information on force balance can be inferred from image observation (*Brodland et al., 2010*; *2014*; *Ishihara and Sugimura, 2012*; *Chiou et al., 2012*; *Ishihara et al., 2013*). For instance, if three cell junctions which have the same tension end at a common meeting point, their respective angles should be equal by symmetry, and thus be 120° each. Reciprocally, any observed deviation from 120° yields a determination of their ratios. The connectivity of the junction network adds redundancy to the system of equations, since the same cell junction tension plays a role at both ends of the junction.

Mathematically speaking, there is a set of linear equations, which involve all cell junction tensions, and cell pressure differences across junctions. We simultaneously estimate tensions and pressures by using Bayesian statistics formulated by maximizing marginal likelihood, or equivalently, by minimizing the Akaike Bayesian information criterion (ABIC) (*Akaike, 1980*; *Kaipio and Somersalo, 2006*). Our expectation that junction tensions are distributed around a positive value is incorporated as a prior (*Ishihara and Sugimura, 2012*; *Ishihara et al., 2013*). This inverse problem requires to invert

large matrices, whose typical size is a few $10^4 \times 10^4$. Our custom program takes advantage that these matrices are sparse (http://faculty.cse.tamu.edu/davis/suitesparse.html), which not only increases the speed of resolution, but also minimizes the ABIC more robustly. It infers all junction tensions up to only one unknown constant which is the tension scale factor, and an additional unknown constant which is the average cell pressure. By integrating tensions and pressures, their contributions to tissue stress can be calculated in any given cell patch $\mathscr{P}$, again up to these two constants, with the Batchelor formula (*Batchelor, 1970*; *Ishihara and Sugimura, 2012*)

$$\sigma = \frac{1}{\mathscr{A}} \left( -\sum_{c \in \mathscr{P}} p_c \mathscr{A}_c \mathbb{1} + \sum_{[ck]} \gamma_{ck} \frac{\vec{\mathscr{L}}_{ck} \otimes \vec{\mathscr{L}}_{ck}}{\|\vec{\mathscr{L}}_{ck}\|} \right) \tag{3}$$

where $\mathbb{1}$ is the two-dimensional identity matrix, $\mathscr{A}$ is the cell patch area, $\mathscr{A}_c$ the area of cell $c$, the sum is over each cell $c$ in the patch and its neighbours $k$, $\vec{\mathscr{L}}_{ck}$ is the vector representing the chord of the junction between cells $c$ and $k$ (it links two vertices, its orientation being unimportant). Only the junction tensions contribute to the deviator of $\sigma$ (hence called 'junctional stress' in the main text), and thus determine the main direction and the difference of two eigenvalues of $\sigma$.

Regarding this junctional stress, we recall three points:

- Junctions contribute to transmit stress between cells even if the biological origin of this stress lies in non-junctional cytoskeletal elements and other structures in the body of the cell. Forces created in the cell body or at the cell boundary are transmitted to its neighbours by contact at the cell-cell interface. These forces transmitted by contact can be decomposed in components parallel and perpendicular to the junction, namely tension and pressure, which we both estimate. Changes of cell volume lead to measurable changes in pressure and thus to detectable changes in our stress estimation.
- We focus on junctional stress because of the growing consensus that it is a important determinant of tissue morphogenesis, as measured by laser ablations of (and outside of) adherens junctions (for review see [*Lecuit et al., 2011*]), or directly probed by experimental manipulation of adherens junctions and compared with simulations (*Bambardekar et al., 2015*). Note that we cannot measure other contributions to the total stress, for instance through cell membrane parts far from adherens junctions.
- The force inference method used to estimate the junctional stress is independent of other contributions to stress or of external force contributions; it thus remains valid even in the presence of cell-substrate interactions that can cause an additional contribution to the total mechanical interactions.

## Data analysis

### Referential

Each of the five wild-type hemi-notum movies was oriented with the midline along the top horizontal side. The whole notum movie was cut into two hemi-notum movies; the left one was flipped to be oriented like the others movies. As spatial landmarks, we use the macrochaetae, which we identify manually at ~16h30 APF $\pm$ 10 min, when the sensory organ precursor (SOP) cells divide (*Gho et al., 1999*), and the following results were independent of the choice of this reference time. For each movie, each macrochaeta is labelled $p$, with $p = 1$ for the closest to the midline, and the largest $p = 7$ or 8 according to the movie. For each of the five movies, labelled $a = 1$ to 5, we call $x$ axis the midline oriented from posterior to anterior, $y$ axis the perpendicular axis oriented from medial to lateral, and as origin $O$ of $(x, y)$ axes the barycenter of the five macrochaetae closest to the midline, $p = 1$ to 5. A box is defined on the first image as the set of cells which barycenter is within a 128×128 pixels square (~40×40 μm²). A box contains several tens of cells, and has a 50% overlap with each of its neighbors. Each box is then labelled by two integer numbers $(m, n)$ which define our two-dimensional space coordinates on a fixed square lattice (Eulerian representation). Averaging over space is implemented by averaging over $(m, n)$. When the whole notum movie is analyzed alone as a whole, the midline is chosen as $x$ axis.

## Weights

Data near the sample boundary are less reliable because of poorer statistics. In order to improve the quality of all quantitative calculations and visual representations, a cell patch at location $(m, n)$ at time $t$ near the boundary of the sample was assigned a weight $0 \leq w^a(m, n, t) \leq 1$, as follows.

We defined the 'bulk' of the tissue in Eulerian description as boxes containing only core cells, and in Lagrangian description as patches placed at least three patches away from the boundary.

In Eulerian description, we defined the 'relative area' $a_r(m, n, t)$ of a box as the sum of the surface area actually occupied by recognizable cells in this box, divided by the surface area of the box. It was close to 1 in all boxes of the bulk, possibly slightly larger than one due to large cells with their barycenter in the box but some of their area outside of the box, and possibly lower than one due to junction pixels (junctions are one pixel wide, they belong to the box but are not assigned to a cell). In Lagrangian description, $a_r(m, n, t)$ was the number of cells in the box, divided by its maximum value in the tissue: in the first images of the movie it was slightly less than 1 over the whole bulk, while it decreased by a factor at least 2 towards the end of the movie.

Its minimum value $a_{\text{bulk}}(t)$ in the bulk at time $t$ was used to normalized all values as

$$a_{\text{norm}}(m, n, t) = \frac{a_r(m, n, t)}{a_{\text{bulk}}(t)} \quad \text{if} \ a_r(m, n, t) < a_{\text{bulk}}(t)$$

which can occur only for boxes out of the bulk, and

$$a_{\text{norm}}(m, n, t) = 1 \quad \text{if} \ a_r(m, n, t) \geq a_{\text{bulk}}(t)$$

which occur for all boxes in the bulk. This way, $a_{\text{norm}}(m, n, t)$ is equal to 1 all over the bulk, and gradually vanishes when approaching the tissue boundary as cell patches contain fewer cells.

Since the results presented below relied on the inversion of the texture matrix (Section C.1.1), we measured the condition number of the texture matrix in this box. The condition number of a function with respect to an argument measures how much the output value of the function can change for a small change in the input argument. For a symmetric 2×2 matrix such as $\mathbf{M}$, the condition number is $\|\mathbf{M}\| \ \|\mathbf{M}^{-1}\|$, always larger than 1. We used its inverse, Matlab's 'reciprocal condition number' $Rc$, which is 1 when $\mathbf{M}$ is isotropic for instance and vanishes when $\mathbf{M}$ is not invertible (i.e. for a box which contains only one half-link, or half-links which are all parallel to each others). Its minimum value $Rc_{\text{bulk}}(t)$ in the bulk at time $t$, typically ~ 0.5, was used to normalized all values as

$$Rc_{\text{norm}}(m, n, t) = \frac{Rc(m, n, t)}{Rc_{\text{bulk}}(t)} \quad \text{if} \ Rc(m, n, t) < Rc_{\text{bulk}}(t),$$

$$Rc_{\text{norm}}(m, n, t) = 1 \ \text{else}$$

This way, $Rc_{\text{norm}}(m, n, t)$ is equal to 1 all over the bulk, and vanishes when approaching the tissue boundary. The weight of the box was the square of the normalized relative area times normalized reciprocal condition number:

$$w^a(m, n, t) = [a_{\text{norm}}(m, n, t) \ Rc_{\text{norm}}(m, n, t)]^2$$

Note that since the stress estimation does not use the inversion of the texture matrix, the weights used for stress measurements are simply $[a_{\text{norm}}(m, n, t)]^2$.

In this work, we mostly consider quantities that have been averaged over some time period ($\Delta t = 2h$ or $14h$) rather than their instantaneous values that can be noisy. For any quantity at $(m, n, t)$, calling $q(m, n, t)$ its instantaneous value, its weighted average over time is $Q(m, n, t)$ and is defined as follows:

$$Q(m, n, t) = \frac{\sum_\tau w^a(m, n, \tau) q(m, n, \tau)}{\sum_\tau w^a(m, n, \tau)}$$

the sum being made between $\tau = t - \Delta t/2$ and $\tau = t + \Delta t/2$. The corresponding mean weights averaged over the same period are:

$$W^a(m, n, t) = \frac{1}{N_{\Delta t}} \sum_\tau w^a(m, n, \tau) \tag{4}$$

with $N_{\Delta t} = \Delta t / \delta t$ is the number of frames during $\Delta t$, the time between two frames being $\delta t$. Weights $W^a(m, n, t)$ decrease according to the number of data in the box, from 1 for a box fully in the bulk of the tissue between $t \pm \Delta t/2$, to 0 for a box outside of the tissue boundary between $t \pm \Delta t/2$. These weights were used in all calculations, averages and graphs. It was even used in maps as an opacity, so that data become more transparent near the boundary where they are less reliable (*Figure 3*, *Figure 3—figure supplements 1*, *2*, *Figure 5*, *Figure 7* and *Videos 4*, *5*).

## Alignment

We define the scalar product of two second-order tensors $\mathbf{Q}, \mathbf{Q}'$ in dimension $d$ as follows:

$$\mathbf{Q}.\mathbf{Q}' = \frac{1}{d}\mathbf{Q}:\mathbf{Q}' = \frac{1}{d}\mathrm{Tr}(\mathbf{Q}\mathbf{Q}'^T) = \frac{1}{d}\sum_{i,j}Q_{ij}Q'_{ji} \tag{5}$$

The scalar product of $\mathbf{Q}$ with itself is the square of its norm

$$\|\mathbf{Q}\|^2 = \mathbf{Q}.\mathbf{Q} = \frac{1}{d}\sum_{i,j}Q_{ij}Q_{ji} \tag{6}$$

Note that with this definition, the identity $\mathbb{1}$ is unitary:

$$\mathbb{1}.\mathbb{1} = \frac{1}{d}d = 1 \tag{7}$$

In dimension 2, for two deviatoric tensors represented as bars, $\mathbf{Q}.\mathbf{Q}' = \|\mathbf{Q}\|\,\|\mathbf{Q}\|'\cos(2\Delta\theta)$ where $\Delta\theta$ is the difference in their bar directions. $\mathbf{Q}.\mathbf{Q}'$ is positive if the bar eigendirections are aligned (close to 0°), negative if these directions are perpendicular (close to 90°), and vanishes in between (close to 45°). The alignement coefficient between two tensors is:

$$\mathscr{A} = \frac{\sum_{m,n}w^a(m,n)\;\mathbf{Q}(m,n).\mathbf{Q}'(m,n)}{\sum_{m,n}w^a(m,n)\;\|\mathbf{Q}(m,n)\|\;\|\mathbf{Q}'(m,n)\|} \tag{8}$$

It is close to 1 if both tensors are aligned everywhere, close to $-1$ if if both tensors are perpendicular everywhere, and vanishes if the tensor directions are independent. *Equation 8* is an average of $\cos(2\Delta\theta)$ with weights combining $w^a(m,n)$ and the bar lengths.

## Comparing and averaging different movies

When the data were averaged over larger time or length scales, the left-right symmetry was visually better (we have checked it quantitatively, data not shown), the reproducibility from one animal to another was increased, and the data maps appeared smoother. Unless stated otherwise, all results presented are computed in boxes of size 128×128 pixels$^2$ (at the onset of the movie), namely 40×40 μm$^2$, with 50% overlap. Time averages are over 2 h for movies or 14 h for still images. Whole notum images are measurements over one animal; archetype refers to average over 5 hemi-notum movies (1 whole animal and 3 half animals). This yielded good statistics while preserving the fine spatial variations of the data maps. Averaging different movies was made possible by defining their common space and time coordinates. We developed a general method to rescale and register movies from different animals and genotypes in time and space, as follows.

## Space registration

We translated the five movies in order to match their origins $O$. The position of macrochaetae is reproducible from one animal to the other: their standard deviation is 5.5 μm along $x$ and 6.3 μm along $y$. We define as an archetype of species $s$ (e.g. the wild-type) a virtual animal which macrochaeta $p$ would be the barycenter of the actual macrochaetae $p$ in the five movies:

$$\left(\overline{x}_p^s, \overline{y}_p^s\right) = \left(\frac{1}{5}\sum_{a=1}^{5}x_p, \frac{1}{5}\sum_{a=1}^{5}y_p\right) \tag{9}$$

We then rescale each actual movie $a$, separately along $x$ and $y$ axes. This is necessary at least for movies taken at different resolutions, or for mutants. The procedure is as follows. Along $x$, the multiplicative factor which minimizes the dispersion of macrochaetae of $a$ from the archetype is:

$$\alpha^a = \frac{\sum_p \left( x_p^a \overline{x}_p^s \right)}{\sum_p \left( x_p^a \right)^2} \qquad (10)$$

In the wild-type tissues, we find $\alpha^a$ ranging from 0.96 to 1.07, with a standard deviation of 0.03. Along $y$ axis, we perform the same analysis, and also include the position of the midline as an independent information; we find rescaling factors of $1 \pm 0.04$. We thus observe that for these five hemi-notum movies the rescaling is not crucial. The variability in tissue scale for these wild-type tissue movies is lower than the variability in macrochaetae positions (not shown). After rescaling, the residual dispersion is slightly lower than the initial one: the standard deviation of macrochaetae position becomes 4.5 µm along $x$ and 5.5 µm along $y$. The referential $(O, x, y)$ in the archetype defines the grid, whereby (0,0) is centered around one box. In the *trbl*$^{up}$ tissues, standard deviations of macrochaetae positions were 6.2 µm along $x$ and 6.3 µm along $y$. With respect to the archetype, the standard deviations were 10.2 and 7.9 µm, respectively. With the same procedure as the wild-type tissue, they were rescaled; after rescaling, the standard deviations were 6.2 and 5.7 µm, respectively. Note that the change in temperature from 18°C to 29°C has apparently no effect on the tissue shape, according to tests performed on the posterior part of the wild-type tissue (*Bosveld et al., 2012*).

## Time registration

While the hAPF was determined with 1 h absolute precision, the tissue rotation rate analysis provided a better relative precision that we used to synchronize the different movies, as described in detail in (*Bosveld et al., 2012*). In the region of the tissue located near the origin of axes, we observed that the rotation rate systematically passed through a maximum during the development: this rotation peak could be used as a biological reference time. For that purpose, the rotation rate was measured and spatially integrated over a rectangular reference window. Plotting this average versus frame number yielded a bell-shaped curve (shown in Figure S4 of [*Bosveld et al., 2012*]). We applied a time translation to each movie so that these curves overlapped. We matched the portion of the curve which had the steepest slope: we thus used the time corresponding to 3/4 of the peak value, in the ascent (rather than the maximum itself, which by definition had a vanishing slope). Its average value was 18:40 hAPF. Hereafter, 'hAPF' indicates the time after this temporal translation has been applied. For instance, after synchronization, the maximum of the contraction-elongation and rotation rates were consistently found at 19:20 and 19:40 hAPF, respectively. This determination reached a $\pm$ 1 interframe (i.e. $\pm 5$ min) relative precision. After this synchronization, we determined that the period of time included in all movies was 13:55-27:55 hAPF (hereafter and in the main text, rounded to '14-28 hAPF'), corresponding to 169 frames, 68 interframes analyzed in what follows. Global time averages were performed over all these frames. Sliding window averages were performed over 2 h every 5 min. The macrochaetae were again manually determined at the time 16h30 APF determined with this synchronization. To improve the precision, the time and space registration was iterated: it changed only slightly, evidencing the robustness of this double registration. The *trbl*$^{up}$ tissues were synchronized with the same procedure as the wild-type tissue, up to a ~10 min precision. For experiments performed at 29°C, the development is accelerated. This was taken into account by dividing time intervals by 0.9 (as determined with $\pm 0.05$ precision by the widening of the wild-type tissue rotation peak [*Bosveld et al., 2012*]).

## Ensemble average, variability and significance

These spatial and temporal adjustments allowed us to assign a system of space-time coordinates $(m, n, t)$ common to the 5 hemi-notum movies, from pupae with a given species $s$ (wild-type or mutant). We again checked, now with spatial details, the reproducibility from one pupa to another. To estimate the averages and standard deviation among animals at each space-time point $(m, n, t)$, for each movie labelled $a = 1$ to 5, we used the time-average weights $W^a(m, n, t)$ defined in *Equation 4*. For any given time-average quantity $Q^a(m, n, t)$, we defined its ensemble average $\overline{Q}^s(m, n, t)$ over the $N_s$ movies of the species (here, the wild-type genotype, $N_s = 5$), local in space and time, as

$$\overline{Q}^s(m,n,t) = \frac{\sum_{a=1}^{N_s} W^a\,(m,n,t)\,Q^a\,(m,n,t)}{\sum_{a=1}^{N_s} W^a\,(m,n,t)} \tag{11}$$

The corresponding mean weights averaged over the same set of animals are:

$$\overline{W}^{\,s}(m,n,t) = \frac{1}{N_s}\sum_{a}^{N_s} W^a(m,n,t)$$

These weights were used in maps as an opacity so that measurements become more transparent near the boundary where data come from fewer animals and are less reliable (see *Figure 4*, *Figure 6*, *Figure 6—figure supplement 1*, *Figure 7*).

The biological variability of the quantity $Q(m,n,t)$ was determined by measuring the weighted standard deviation $\delta Q(m,n,t)$ for the species

$$\delta Q(m,n,t) \;=\; \left( \frac{\sum_{a=1}^{N_s} W^a}{\left(\sum_{a=1}^{N_s} W^a\right)^2 - \sum_{a=1}^{N_s} (W^a)^2} \sum_{a=1}^{N_s} W^a \left[Q^a - \overline{Q}^s\right]^2 \right)^{\frac{1}{2}} \tag{12}$$

The fraction in *Equation 12* generalizes the usual term for unbiased standard deviation, $(N_s - 1)^{-1}$, which is recovered when all weights $W^a$ are equal to 1. This unbiased weighted standard deviation can be calculated for each of the different independent components of a symmetric tensor $\mathbf{Q}$. In general, these components are $\delta(Q_{xx}^s + Q_{yy}^s)$, $\delta(Q_{xx}^s - Q_{yy}^s)$ and $\delta Q_{xy}^s$. When we consider only the deviator of $\mathbf{Q}$, the components are $\delta(Q_{xx}^s - Q_{yy}^s)$ and $\delta Q_{xy}^s$. We compare them to the biological variability $\delta Q$, and define that a measurement at a given position and time $(m,n,t)$ is significant if it satisfies:

$$\|\overline{Q}\| \geq \|\delta Q\| \tag{13}$$

namely if:

$$\left[\left(\overline{Q}_{xx}^s - \overline{Q}_{yy}^s\right)^2 + 4(\overline{Q}_{xy}^s)^2\right]^{\frac{1}{2}} \geq \left[\delta\left(Q_{xx}^s - Q_{yy}^s\right)^2 + 4(\delta Q_{xy}^s)^2\right]^{\frac{1}{2}} \tag{14}$$

Using the significance criterion of *Equation 14*, measurements larger than the biological variability are plotted in color while smaller ones are shown in grey. Note that if the direction of the anisotropic part of the tensor varies a lot between animals, this variation decreases the amplitude of the anisotropic part of the mean tensor $\overline{Q}$, or equivalently the amplitude of $\overline{Q}$, and therefore the measurement is considered as non significant based on *Equation 14*.

## Simulations

We numerically simulated the different processes using the cellular Potts model (*Graner and Glazier, 1992*; *Glazier and Graner, 1993*), an algorithm relevant in biology to describe variable cell shape, size, packing and irregular fluctuating interfaces of cells in 2 or 3 dimensions (*Marée et al., 2007*; *Bardet et al., 2013*). Each cell is defined as a set of pixels, here on a 2D square lattice; their number defines cell area. The pixelisation of the calculation lattice can be chosen to match the resolution of experimental images. A cell shape changes when one of its pixels is attributed to another cell instead. We used periodic boundary conditions, with external medium surrounding cells (a state without adhesion or area and perimeter constraints). We use here a simple version where cells minimise their surface energy. Energy minimisation uses Monte Carlo sampling and the Metropolis algorithm, as follows. We randomly draw (without replacement) a lattice pixel and one of its eight neighboring pixels. If both pixels belong to different cells, we try to copy the state of the neighboring pixel to the first one. If the copying decreases the total energy, we accept it, and if it increases the total energy, we accept it with a probability <1 known as 'fluctuation allowance'.

Divisions were implemented as follows. Cells were growing with an initial asynchrony in their cycle ranging from 0 to 40% (to avoid divisions all occurring at the same time). As they grew, the group of cells underwent a force gradient along the $x$ axis which tended to stretch each individual cell shape as well as the cell patch shape along the $x$ axis. Cells divided only once when their target area had doubled, along their long axis; they did not regrow after division, thereby recovering their initial

size. Cells divided along their long axis, which resulted in divisions oriented along $x$. Cells tended to relax their elongated shapes before or after divisions, resulting in some rearrangements. Each image was $800{\times}800$ pixels$^2$ and was cropped to yield the final movie. There were 50 Monte-Carlo steps between two successive images.

Other processes were implemented in a similar way (except when stated otherwise), as follows. Stretching the pattern and allowing for cell shape relaxation led to oriented rearrangements. Affinely stretching the pattern by direct image manipulation using an image treatment software instead of Potts simulations (hence not followed by any cell shape relaxation) led to strong cell shape changes. Delaminations were obtained by gradually decreasing the cell target area. Cell integration movie was produced by reversing the order of images of delamination simulation movie. Fusions were forced on cells by random removal of cell-cell junctions (the same image was gradually Gaussian-blurred then thresholded), and cells were then let to relax their shapes. Boundary flux was implemented by gradual removal of successive external cell layers. Rotation of the cell patch was achieved by direct image manipulation. A simulation movie is made with 241 frames, 240 interframes. It would be analogous to a experimental movie of 240 $\times$ 5 min, or 20 h. Hence an experimental scale of $10^{-2}$ h$^{-1}$ would be equivalent to 8.3 $10^{-4}$ interframe$^{-1}$ in simulations.

## Practical implementation of the formalism

The approach which leads to the formalism and to the decomposition into separate cellular processes is described in details in the Appendix. We describe here its practical implementation.

### Link assignment

In practice, when studying a monolayer of cells, we are mainly interested in morphogenetic movements within the plane of the monolayer: we focus on in-plane components and actually implement the formalism in two dimensions, as follows. We use two successive images of the segmented and tracked movie. In both images, for each cell $c$ the list of neighbour cells $k$ is identified. A half-link $ck$ is the link between a cell $c$ center and its neighbour $k$ center, and is listed independently from the half-link $kc$ oriented from $k$ to $c$. In both images, we list all half-links. The rare cases where four cells meet (along a vertex, in 2D, or along a line, in 3D) are listed separately: in what follows, it is possible to treat them separately if needed. This is what we do, and at the end we lump their contribution with that of other cells.

The tracking enables to classify each half-link into one and only one category (which ensures the completeness of the formalism): appeared or disappeared through one of the processes, or conserved between both images. For instance, when a cell divides, all its half-link disappearances and the appearances of half-links for its daughters are included in the division process (conversely, the 'division orientation' includes only the link between the newly created daughter cells) (*Figure 1a,b*). Similarly, when cell $c$ undergoes a delamination, some of its former neighbors enter in contact: their new half-links are also counted in the delamination contribution (*Figure 1a,b*). The half-link $kc$ can be in a different category from half-link $ck$, for instance if in this interframe cell $c$ undergoes a division while cell $k$ undergoes a delamination, rearranges with other neighbours, or exits at the sample boundary. The result is a classification of all half-links in each image.

### Tracking of cell patches

All individual movies were analyzed using Lagrangian measurements, thanks to the tracking of cells and patches illustrated in *Figure 3d*, *Figure 5a*, *Videos 1* and *3*. To coarse-grain the measurements, the whole image is subdivided into cell patches $\mathscr{P}$. In the Lagrangian description chosen here, each patch is tracked from one frame to the next. After a division, daughter cells are assigned to the patch of their mother. All cells and links belonging to each patch are used to calculate tissue deformation and the deformations associated with each cell processes occurring between two images, and we then turn them into rates and average them over time (see Appendix, sections B and C). Those deformation rates are related by the balance equation (see *Equation 77*):

$$\mathbf{G} = \mathbf{S} + \mathbf{D} + \mathbf{R} + \mathbf{A} + \mathbf{N} + \mathbf{C} + \mathbf{J} \tag{15}$$

We measure separately each term of both sides of *Equation 15*, by assigning each link geometric change to $\mathbf{G}$, and each link topological change to a specific cell process $\mathbf{P}$, taking advantage of their completeness. Finally, we systematically check that *Equation 15* is satisfied.

## Projection of cellular process onto tissue morphogenesis

We consider any of the cell process (e.g. divisions, rearrangements, cell shape and size changes, delaminations, ...), which we note $P$, measured by the tensor $\mathbf{P}$ (e.g. $\mathbf{D}$, $\mathbf{R}$, $\mathbf{S}$, $\mathbf{A}$, ...). While the isotropic part of $\mathbf{P}$ is a scalar, which directly quantifies the growth rate, the CE part of $\mathbf{P}$ is a tensor characterized by an amplitude and a direction that differ from the amplitude and direction of tissue CE rate $\mathbf{G}$. In order to determine the effective contribution of process $P$ to local tissue deformation, we project it onto $\mathbf{G}$. To do so, in each patch from the tissue, we first define the unitary tensor $\mathbf{u_G}$ that is aligned with $\mathbf{G}$ (*Figure 4c*):

$$\mathbf{u_G} = \frac{\mathbf{G}}{\|\mathbf{G}\|} \qquad (16)$$

which trivially ensures $\|\mathbf{u_G}\| = 1$. Then we project process $P$ contribution $\mathbf{P}$ along $\mathbf{u_G}$ using the scalar product defined in *Equation 5* (*Figure 4c*):

$$\mathbf{P}_\| = \mathbf{P} . \ \mathbf{u_G} \qquad (17)$$

We call this projection $\mathbf{P}_\|$ the component of $\mathbf{P}$ parallel to tissue morphogenesis (or 'along $\mathbf{G}$', for short). It is expressed as a rate of change per unit time, i.e. hour$^{-1}$. It represents the effective contribution of $\mathbf{P}$ to tissue morphogenesis. The additivity (*Equation 15*) also applies separately to these components:

$$\mathbf{G}_\| = \mathbf{S}_\| + \mathbf{D}_\| + \mathbf{R}_\| + \mathbf{A}_\| + \mathbf{N}_\| + \mathbf{C}_\| + \mathbf{J}_\| \qquad (18)$$

In 2D, $\mathbf{P}_\| = \|\mathbf{P}\| \cos(2\Delta\theta)$, where $\Delta\theta$ is the difference in directions of $\mathbf{G}$ and $\mathbf{P}$ bars. Thus, if a process CE rate $\mathbf{P}$ is exactly parallel to the tissue CE rate $\mathbf{G}$, it has a positive component on $\mathbf{G}$, which is exactly its own amplitude: $\mathbf{P}_\| = \|\mathbf{P}\|$. If the $\mathbf{P}$ bar is rather parallel to $\mathbf{G}$, it has a positive component $\mathbf{P}_\|$ on $\mathbf{G}$. If the $\mathbf{P}$ bar is at 45° angle to $\mathbf{G}$, its component $\mathbf{P}_\|$ on $\mathbf{G}$ is zero. If the $\mathbf{P}$ bar is rather perpendicular to $\mathbf{G}$, its CE rate has a negative component $\mathbf{P}_\|$ on $\mathbf{G}$. If the $\mathbf{P}$ bar is exactly perpendicular to $\mathbf{G}$, its CE rate has a negative component on $\mathbf{G}$, which is exactly minus its own amplitude: $\mathbf{P}_\| = -\|\mathbf{P}\|$. As a consequence, the component of tissue CE rate along itself, $\mathbf{G}_\|$, is the amplitude of $\mathbf{G}$, and is positive by construction. *Figure 4c* shows the sign of $\mathbf{P}_\|$.

## Uncertainties

In this section, we list the various sources of uncertainties that we encounter, present our methods to decrease them as much as possible, and discuss why their impact on our analyses remains limited. They fall into two main categories: those related to the acquisition of data which will be used as input for the formalism; and those related to the formalism itself.

- Errors due to image analysis affect the input of our formalism. Segmentation errors can result in false identification of cells, while tracking errors can result in false identification of cells and their lineages. It is thus necessary to choose an image acquisition time sufficient to ensure an image contrast and quality which enable a segmentation with a low error rate. This sets a minimal value to the acceptable time difference $\delta t$ between two successive images. In the notum and wing tissues we observe here, fusion and integration events are used as markers of segmentation and/or tracking errors. In the notum, we use them as feedback to correct the segmentation until the contributions of fusions and integrations are negligible with respect to the contributions we want to measure.

- The tissue is a three-dimensional ('cuboïdal') monolayer and we do not assume it to be 2D. However, we perform its study in 2D. More precisely, we image it using the E-Cadherin:GFP marker, which labels the apical adherens junctions. Moreover, several quantites we measure are actually 2D, such as the velocity field, or its gradient which represents the morphogenesis. Other quantities like cell division orientation or stress field are 3D but can be studied in 2D independently from what occurs in the third dimension. Finally, some 3D processes like delamination in which apical area shrinks have an effect on tissue morphogenesis which is similar to

that of an apoptosis where a cell entirely shrinks, and is thus not necessary to distinguish within the scope of the present study.

- The 3D structure of the notum also plays a role because the tissue is curved. It is imaged in 3D using a confocal spinning disc microscope; the height of its surface is recorded and we can entirely reconstruct its curvature. We can then obtain the local angle $\theta$ of the tissue surface with the horizontal plane: in the notum, this is mainly along the direction $y$ (medio-lateral), since along the axis $x$ the tissue curvature is much smaller. We obtain the projection factor $\cos\theta$ which should in principle be applied as a correction in the corresponding direction either to the raw image before any treatment, or a posteriori to results of the formalism. We measure $\theta$ is at most 0.01 radian over the tissue and reaches at most 0.02 radian in the most lateral part of the whole thorax image. The absolute correction factor is the same for all tensors, and it is negligible with respect to 1 in the results presented here ($1-\cos\theta \sim 10^{-4}$), so that it does not affect significantly the results presented here. For simplicity, we choose not to perform any correction to our results, knowing that we can perform the correction a posteriori if it appears necessary in the future. All these effects are completely negligible in the wing, which is much flatter than the notum.

- For a given set of data used as input to our formalism, the formalism itself does have uncertainties which affect its output. For instance, the time difference $\delta t$ between two successive images should be small enough so that the fraction of conserved links remains larger or comparable to the fraction of non-conserved links, and that our measurement of tissue growth and morphogenesis (which is based on conserved links) accurately quantifies the actual morphogenesis. Since $\delta t$ should be sufficiently large to keep a good image quality, the question is how to choose an optimal $\delta t$. We find that in our case, $\delta t = 5$ min is optimal, and for these value both constraints are satisfied: the image is good enough to be automatically segmented for a large part, and the morphogenesis is well quantified by our measurement. Finally, in order to validate our workflow, we have checked that with the data of the preceding paper (*Bosveld et al., 2012*) our new formalism recovers consistent results. In conclusion, our pipeline which links the image acquisition to the data representation (*Figure 1e*) is valid and its accumulated uncertainties are lower than the biological variability within an animal or between different animals.

## Comparing wild-type and mutant tissue

In mutant tissues, the space and time registration are performed as in the wild-type tissues. Mutant tissues can exhibit a different variability than wild-type tissues; in practice, we observe a larger variability in *trbl$^{up}$* mutant tissues than in wild-type tissues. Moreover, wild-type and mutant tissues can be registered together. The total morphogenesis as well as the measurement of each cell-level process is computed similarly in a mutant tissue. We compare them term by term with the corresponding values in wild-type tissues. We assay at which time and position the mutation has a significant effect with an inter-genotype variability larger than the intra-genotype one. It is also possible to subtract term by term each measurements performed in the wild-type tissues $\mathbf{P}_{wt}$ and the measurements performed in the mutant tissues $\mathbf{P}_{mutant}$. We can plot the measurement difference $\Delta\mathbf{P}$ defined as $\Delta\mathbf{P} = \mathbf{P}_{mutant} - \mathbf{P}_{wt}$ this difference represents the part of $\mathbf{P}_{mutant}$ that has been added by the mutant condition to $\mathbf{P}_{wt}$. It represents the effect of the overexpressed gene in the process $P$ of wt morphogenesis, and its projection $\Delta\mathbf{P}_{\parallel}$ along the wild-type tissue CE rate represents its effective contribution to wild-type morphogenesis.

## Acknowledgements

We thank M Gho, Y Hong and H Oda for reagents; the members of the Developmental Biology Curie imaging facility (PICT-IBiSA@BDD, UMR3215/U934) for their help and advice with confocal microscopy; P Marcq and D Lubensky for comments on the manuscript and numerous discussions; Y Goya, C Genet, M Alexandre, I Aboulker Sitbon and T Abdulmanova for help with segmentation; M Merkel, M Popović, R Etournay, S Eaton and F Jülicher for their comments on the manuscript and in particular on equation 48. This work was supported by Labex DEEP, ARC-4830, ANR-MorphoDro, ERC Starting (CePoDro), ERC Advanced (TiMorp) and Programme Labellisé Fondation ARC (SL220130607097) grants and ARC (individual aid to JL-G).

## Additional information

### Funding

| Funder | Grant reference number | Author |
| --- | --- | --- |
| Centre National de la Recherche Scientifique | | Boris Guirao<br>Floris Bosveld<br>François Graner<br>Yohanns Bellaïche |
| Institut Curie | | Boris Guirao<br>Stéphane U. Rigaud<br>Floris Bosveld<br>Anaïs Bailles<br>Jesus Lopez-Gay<br>Yohanns Bellaiche |
| European Research Council | | Boris Guirao<br>Stéphane U. Rigaud<br>Floris Bosveld<br>Anaïs Bailles<br>Jesus Lopez-Gay<br>Yohanns Bellaiche |
| Labex DEEP | | Boris Guirao<br>Stéphane U. Rigaud<br>Floris Bosveld<br>Anaïs Bailles<br>Jesus Lopez-Gay<br>Yohanns Bellaiche |
| Japan Society for the Promotion of Science | Sakura | Boris Guirao<br>Shuji Ishihara<br>Kaoru Sugimura<br>François Graner<br>Yohanns Bellaiche |
| Institut National de la Santé et de la Recherche Médicale | | Yohanns Bellaiche<br>Boris Guirao<br>Floris Bosveld<br>Stéphane U. Rigaud<br>Jesus Lopez-Gay |
| Fondation ARC pour la Recherche sur le Cancer | | Stéphane U. Rigaud<br>Jesus Lopez-Gay<br>Yohanns Bellaiche |
| Japan Science and Technology Agency | Presto | Kaoru Sugimura |
| Centre National de la Recherche Scientifique | MecaBio | François Graner |
| Centre National de la Recherche Scientifique | CellTiss | François Graner |

The funders had no role in study design, data collection and interpretation, or the decision to submit the work for publication.

### Author contributions

BG, Planned the project; Developed the cell tracking; Developed and implemented the formalism; Improved the force inference method; Designed and performed the numerical simulations; Developed or implemented methods used in the manuscript; Performed the analysis; Assembled the figures and movies; Wrote the manuscript; SUR, Implemented the space registration, rescaling and averaging over animals; Developed or implemented methods used in the manuscript; Assembled the figures and movies; FB, Planned the project; Performed live imaging and segmentation; AB, Implemented the space registration, rescaling and averaging over animals; Developed or implemented methods used in the manuscript; JL-G, Performed live imaging; Developed or implemented methods used in the manuscript; SI, KS, Acquired and segmented the movies of wing development; Initially developed the force inference method; Developed or implemented methods used in the

manuscript; FG, Planned the project; Developed the formalism; Developed or implemented methods used in the manuscript; Performed the analysis; Wrote the manuscript; YB, Planned the project; Performed the analysis; Wrote the manuscript

**Author ORCIDs**
Jesús López-Gay, http://orcid.org/0000-0002-9388-4065
Shuji Ishihara, http://orcid.org/0000-0002-3302-3925

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

## Appendix: Deformation gradient, material strains and cell process strains in a deformable cellular material

# A Introduction

### A.1 Scalar vs tensor measurements

Once the cells have been segmented and tracked throughout the movie (see 'Materials and methods' for details), our goal is to extract useful quantitative information on divisions, rearrangements, cell size and shape changes and apoptoses, and quantify the relative importance of these cell-scale events in complex morphogenetic processes. Several descriptive terms, such as cell or tissue polarity, oriented cell divisions, convergence-extension, or intercalation, have an underlying common point: their quantitative description refers to an amplitude, an anisotropy and a direction. The mathematically robust objects that encompass simultaneously these informations are the tensors, which also offer the advantage to be valid in spaces of two, three or any dimensions. Using tensors is particularly important in a heterogeneous tissue like the notum, where morphogenetic flows of various amplitude, direction and anisotropy occur at different locations.

For instance, counting the divisions results in integer numbers, and statistics on cell division directions results in angle distributions. While these classical descriptions of the cell division rate and orientation are essential, we need to quantitatively determine to which extent they contribute to tissue morphogenesis. Characterizing divisions using tensors can result in a better description of oriented cell divisions and a better understanding of their effects. Similarly, if a cell rearrangement is followed immediately by its inverse (which often occurs either in reality, or as an artifact when segmenting a short junction), this leads back to the initial state, and their opposite effects cancel out; while if two rearrangements occur in the same direction, their effects cumulate. Counting cell rearrangements as integer numbers leads to $1 + 1 = 2$ rearrangements in both examples, while a tensor analysis better captures in each case its actual impact to morphogenesis.

There is a need for a formalism with the following requirements : it should include a self-consistent, rigorous definition of the tensors used for measurements; enable a decomposition of tissue deformation into the sum of contributions coming from elementary cell processes; the tensors should be significantly measurable with a good signal-to-noise ratio, and visually representable; the formalism should be purely descriptive, in the sense that it should quantify observed morphogenetic changes independently of the biological or physical origin of the forces that drive these changes, whether they are internal forces such as active cytoskeletal forces, viscous dissipation or cytoplasm pressure, or external forces such as interaction with another cell layer or solid substrate.

### A.2 Previous approaches

We aim here to generalize and take further the approaches used in four articles which have begun to apply this type of methods to developmental biology, enabling quantitative comparison between experiments and simulations:

- *Rauzi et al. (2008)* used a matrix called 'texture' as defined in Equation 6 of (*Graner et al., 2008*) to quantify cell shape changes (see their Figures 5a and S4). It is based on the links connecting each cell center with the centers of its neighbors: it expresses, in $\mu m^2$, the variance of link length in each direction. Separately, rearrangements were counted and quantified as integer numbers.

- *Aigouy et al. (2010)* used a former version of the texture as defined in the Equation 1 of (*Aubouy et al., 2003*), which expressed the variance of the cell junctions in each direction. More precisely, they called 'cell elongation' (defined in their Eq. S25) the deviatoric part of

the texture. Rearrangements were quantified separately, and as integer numbers (see their Figure 4).

- *Butler et al. (2009)* used their own method (introduced in [*Blanchard et al., 2009*]). They computed the contribution of cell shape changes to flow. They then subtracted the cell shape change rate from the tissue shape change rate. They thus indirectly inferred the contribution of all types of topological changes to morphogenesis as tensors, i.e. including their amplitude and their direction.

- In a preceding paper (*Bosveld et al., 2012*), we adapted the formalism of *Graner et al., 2008*, already validated on non-biological cellular networks (*Cheddadi et al., 2011*), to quantify tensorially the contributions of cell shape changes and rearrangements to tissue shape changes in the Drosophila scutellum, a subpart of the dorsal thorax. We took advantage of averages over several animals to improve the signal-to-noise ratio, and to subtract a mutant from a wild-type tissue, evidencing the effect of the mutation and determining at which place the difference was significant. In (*Bardet et al., 2013*) we used a similar approach in an inter-vein of the Drosophila wing.

Two additional formalisms have also been proposed recently. *Economou et al. (2013)* use scalars (rather than tensors) to analyze the one-dimensional elongation of a developing mouse tissue. *Etournay et al. (2015)* is based on the estimation of the local tissue deformation at the sub-cellular scale of triangles formed by three contiguous links surrounding each tricellular junction (three-fold vertex) and analyses the pupal wing development in details, using a tensorial approach.

## A.3 Present approach

The formalism presented here is tensorial, to take into account the variable orientations of cell processes. It accounts for processes like cell divisions, apoptoses or delaminations which can drastically modify cell number in a proliferative tissue. We made it complete by including the remaining processes that can occur in other tissues, or because of segmentation mistakes, or simply because of the limited field of acquisition of the microscope: coalescences (fusions) and new cell integrations, as well as fluxes of cells through the boundary of the tissue of interest or of the microscope field of view.

In a cellular material such as a living tissue, the material states can be characterized by the links joining the centers of mass of neighboring cells, thereby tiling the tissue with an irregular triangular networks of links (*Figure 1*, *Figure 1—figure supplement 1*, *Video 1*). Then, by keeping track of those links over time, the geometrical and topological changes in the tissue can also be characterized, and related to the elementary cell processes (cell kinematics) (*Video 1*) (*Graner et al., 2008*; *Etournay et al., 2015*; *Tlili et al., 2015*).

Therefore, as a starting point, we use a statistical definition similar to (*Graner et al., 2008*) to characterize the state and the deformation of the material network of links. Each change regarding a link (appearance, disappearance, change of orientation or length) is assigned to one and only one process, guaranteeing that the morphogenetic events are decomposed into individual processes and that these processes add up to tissue growth and morphogenesis. One of our main goals is to obtain a balance equation relating the local tissue deformation to the elementary cell processes making up this deformation, thereby connecting the cell and tissue scales.

To quantify tissue deformation and all cell processes, we aim at defining dimensionless tensors in order unify their characterizations, but also to relate their statistical descriptions to quantities used in continuum mechanics. Indeed, the continuum mechanics description facilitates comparison between different experiments, or between experiments, simulations and theory. It requires to treat the tissue as continuous, that is, describe cell patches without any explicit signature of each individual cell size. Cell processes are described statistically rather than with details. For each cell process, it is possible to construct a biologically relevant continuous counterpart of the statistical quantity, namely a quantity which is dimensionless (or a rate), with no size.

The procedure presented here improves and generalizes the ones presented in (*Graner et al., 2008*; *Bosveld et al., 2012*; *Bardet et al., 2013*). One of the main difficulties to overcome is that biological processes such as divisions or delaminations can change drastically the number of cells and links on the image in proliferative tissues. If we had directly adapted the formalism presented in the above papers (Section A.2), the tensor describing cell shape and size changes would have been strongly affected by the changes in cell and link numbers, and would thus have lost a clear interpretation. In addition, divisions and delaminations would have been mixed with rearrangements.

The formalism presented here provides a unified quantitative characterization of tissue deformation and of all cell processes making up this deformation. All these quantities are characterized with dimensionless symmetric tensors that quantify the tissue strain and the elementary strains associated with each cell process, independantly of rigid body movements. It provides a simple balance equation where the tissue strain is equal to the sum of the cell process strains. Each tensor remains biologically interpretable as a separate contribution to the tissue morphogenesis. Importantly, the formalism is valid at arbitrarily large and heterogeneous deformations, whether the tissue is highly proliferative or not, and it is defined in a space of arbitrary dimension, and is therefore directly implementable in two or three dimensions.

After having briefly recalled some basic definitions of deformation in continuum mechanics and of the statistical description of a cellular material that constitute our starting points, we define a coarse-grained deformation gradient tensor $\mathbf{F}$ for a general deformable cellular material (section B). Then we define a coarse-grained strain tensor to quantify the total strain in the cellular material $\mathbf{G}^*$, and we relate it to the strains associated with elementary cell processes ($\mathbf{S}^*$, $\mathbf{P}^*$) with a simple and general balance equation (section C). Incidentally, this enables us to define the material geometrical ($\mathbf{S}^*$) and topological ($\mathbf{T}^*$) strains. Finally, we compare our approach with the ones previously published (section D).

# B Deformation gradient in a cellular material

Before introducing our statistical description of a general deformable cellular material, we recall some basic notions of finite deformation in continuum mechanics (*Malvern, 1969*). They are used for comparison only, and not used for measurements (section B.1). Then we show how we can define a coarse-grained deformation gradient tensor for such material, and how it is related to our statistical description (section B.2).

## B.1 Continuum mechanics versus statistical description

### B.1.1 Continuum mechanics of deformable materials

A point of the material in its initial or reference configuration is characterized by the vector $\vec{X}$, while $\vec{x}$ defines the position of this point in the final or current configuration of the material (at time $t$). The displacement of this point is $\vec{u} = \vec{x} - \vec{X}$. Rigid body displacements such as translations or rotations of the material moving as a whole obviously generate displacements $\vec{u} \neq 0$, but do not however generate any stress in the material. It is therefore crucial to define quantities that characterize the material deformation or strain independently of rigid body movements, and especially rotations that can wrongfully contribute to the material strain if they are not treated properly. For this purpose, one must therefore define quantities characterizing the material deformation rather than its displacement, which is done by defining tensors based on the gradient of the displacement, whether adopting a Lagrangian or an Eulerian description, as explained in the following.

Lagrangian quantities refer to $\vec{X}$, as does the Lagrangian gradient $\nabla = \partial/\partial\vec{X}$. The displacement gradient $\nabla\vec{u}$ vanishes for an arbitrary rigid body translation, but not for an

arbitrary rigid body rotation. When it is infinitesimal, its symmetrical part determines the infinitesimal deformation $\varepsilon$ (also called 'infinitesimal strain'):

$$\varepsilon = \frac{\nabla\vec{u} + \nabla\vec{u}^T}{2} \tag{19}$$

More generally, an arbitrary finite deformation is classically characterized by the 'deformation gradient' tensor:

$$\mathbf{F} = \frac{\partial\vec{x}}{\partial\vec{X}} = \mathbb{1} + \nabla\vec{u} \tag{20}$$

$\mathbf{F}$ relates any infinitesimal vector $d\vec{X}$ of the undeformed material to its deformed state $d\vec{x}$:

$$d\vec{x} = \mathbf{F}d\vec{X} \tag{21}$$

The polar decomposition theorem enables to decompose $\mathbf{F}$ into the product of a rotation tensor $\Theta$ and of a symmetric tensor $\mathbf{U}$ called the 'right stretch tensor': $\mathbf{F} = \Theta\mathbf{U}$. This means that any finite deformation $\mathbf{F}$ can be decomposed as a stretch of a material ($\mathbf{U}$) followed by its rotation ($\Theta$). To quantify the material deformation independently of rigid body movements (translation and rotation), several symmetric deformation tensors (also called 'strain tensors') based on $\mathbf{F}$ can be defined. A proper strain tensor must be dimensionless, vanish for rigid body movements, and coincide with $\varepsilon$ for small deformations. The Green-Lagrange strain tensor $\mathbf{E}$ involves the right Cauchy-Green strain $\mathbf{F}^{\mathrm{T}}\mathbf{F} = \mathbf{U}^2$, and is commonly used:

$$\mathbf{E} = \frac{1}{2}(\mathbf{F}^{\mathrm{T}}\mathbf{F} - \mathbb{1}) = \varepsilon + \frac{1}{2}\nabla\vec{u}^{\mathrm{T}}\nabla\vec{u} \tag{22}$$

Equivalently, one can alternatively decompose $\mathbf{F}$ in the symmetric tensor $\mathbf{V}$ called the 'left stretch tensor' and the rotation tensor $\Theta$, as $\mathbf{F} = \mathbf{V}\Theta$. In this equivalent formulation, the material is this time first rotated with same rotation ($\Theta$), then stretched ($\mathbf{V}$). It defines another acceptable strain tensor, this time involving the left Cauchy-Green strain $\mathbf{F}\mathbf{F}^{\mathrm{T}} = \mathbf{V}^2$, as follows:

$$\mathbf{E}^* = \frac{1}{2}\left(\mathbf{F}\mathbf{F}^{\mathrm{T}} - \mathbb{1}\right) = \varepsilon + \frac{1}{2}\nabla\vec{u}\nabla\vec{u}^T \tag{23}$$

Note that here, as well as throughout this appendix, we use a $*$ to indicate the dimensionless symmetric tensors we use to characterize the strain of the cellular material, and the strains associated to the contributions of each cell process making up the material strain (**Equation 75**).

Eulerian quantities refer to $\vec{x}$. This includes kinematic quantities such as the velocity gradient $\mathrm{grad}\vec{v}$, where $\mathrm{grad} = \partial/\partial\vec{x}$. The symmetrized velocity gradient is the total deformation rate, which combines the rate of geometrical deformation (related to size and shape changes) and the rate of topological deformation (related to divisions, delaminations and rearrangements).

$$\mathrm{grad}\vec{v}^{\mathrm{S}} = \frac{1}{2}(\mathrm{grad}\vec{v} + \mathrm{grad}\vec{v}^{\mathrm{T}}) \tag{24}$$

## B.1.2 Statistical description of deformable cellular materials
We now introduce statistical methods actually used for measurements. Their role is to extract, from cell-level details, the relevant information for a continuum mechanics description. These measurement definitions are valid in both two and three dimensions.

## Texture

The first step consists in defining a texture tensor which statistically characterizes the pattern itself (a 'state function'), whose changes correspond to cell processes. Here $k$ labels a neighbor of cell $i$, the vector $\vec{\ell}_{ik}$ is the link between the centers of cells $i$ and $k$. For column vectors, the tensor (or outer) product of $\vec{\ell}_{ik}$ by itself reads:

$$\vec{\ell}_{ik} \otimes \vec{\ell}_{ik} = \vec{\ell}_{ik} \vec{\ell}_{ik}^T \tag{25}$$

while for row vectors $\vec{\ell}_{ik} \otimes \vec{\ell}_{ik} = \vec{\ell}_{ik}^T \vec{\ell}_{ik}$. The total texture $\mathbf{M}$ of a cell patch $\mathscr{P}$ is defined as follows:

$$\mathbf{M} = \sum_{i \in \mathscr{P}} \frac{1}{2} \sum_{\langle k \rangle} w_{ik} \vec{\ell}_{ik} \otimes \vec{\ell}_{ik} = \sum_N \frac{1}{2} w \vec{\ell} \otimes \vec{\ell} \tag{26}$$

The texture thus expresses the variance of link length in all directions. Here $i \in \mathscr{P}$ means cells belonging to the cell patch $\mathscr{P}$, $\langle k \rangle$ indicates a sum over cells $k$ neighboring cell $i$, and $N$ is the number of half-links in the patch, in the sense that $\vec{\ell}_{ik}$ and $\vec{\ell}_{ki}$ are two 'halves' of the same link connecting the centers of cells $i$ and $k$. The factor $1/2$ therefore avoids counting twice each link, and naturally assigns a weight $1/2$ to a link $\vec{\ell}_{ik}$ involving a cell $i$ at the patch boundary and having its neighbor $k$ in the neighboring patch. Similarly, if a link belongs to a four-fold vertex at time $t$, it is counted with a weight $w_{ik} = 1/2$, otherwise $w_{ik} = 1$. By doing so, when a link is in the process of appearing or disappearing around time $t$, the associated topological change is attributed half to time interval before $t$ and half to time interval after $t$. In order to use lighter notations, we drop the $ik$ subscripts and rather use sum over half-links like in the last part of **Equation 26**.

The cell texture defined in Equation 3 of (**Graner et al., 2008**) was divided by the number of links in the cell patch, to be an intensive quantity; this had no drawback in foams or in non-proliferative tissues, as the number of links was constant in good approximation. Conversely, most developing tissues proliferates as they deform, and their number of links can change dramatically. We therefore first define the texture as an extensive quantity (**Equation 26**).

## Decomposition of texture changes

Throughout the appendix, any quantity 'Q' in its initial and final states is labeled with upper and lower cases, respectively namely $Q$ and $q$, thereby keeping the convention used in continuum mechanics. We write $\Delta$ the difference between these two states, final minus initial, and write $\delta$ the difference between two successive images of the movie:

$$\Delta Q = q - Q = \sum_i^f \delta Q \tag{27}$$

Between the initial and final states, due to processes of many different types, the texture of the cell patch $\mathscr{P}$ varies:

$$\Delta \mathbf{M} = \mathbf{m} - \mathbf{M} = \sum_n \frac{1}{2} w \vec{\ell} \otimes \vec{\ell} - \sum_N \frac{1}{2} W \vec{L} \otimes \vec{L} \tag{28}$$

Some links (labeled with subscript 'c') are conserved between initial and final states, but their geometry (length and/or direction) may have changed. Conversely, some links in the cell patch have appeared (subscript 'a') or disappeared (subscript 'd'), thereby changing the topology of the link network, due to processes such as divisions, apoptoses/delaminations, rearrangements, integrations/nucleation, fusions/coalescences and inward/outward fluxes through the image boundaries. We now split the sums in **Equation 28** according to these categories and use the fact that for conserved links, $n_c = N_c$ and $w_c = W_c$:

$$\Delta \mathbf{M} = \underbrace{\sum_{n_c} \frac{1}{2} w_c \vec{l} \otimes \vec{l} - \sum_{n_c} \frac{1}{2} w_c \vec{L} \otimes \vec{L}}_{\mathbf{G}} + \underbrace{\sum_{n_a} \frac{1}{2} w_a \vec{l} \otimes \vec{l} - \sum_{N_d} \frac{1}{2} W_d \; \vec{L} \otimes \vec{L}}_{\mathbf{T}} \qquad (29)$$

The first term in **Equation 29** quantifies the total change of texture due to geometrical changes ($\mathbf{G}$) and is measured by the difference of texture calculated over conserved half-links only (number $n_c$):

$$\mathbf{G} = \mathbf{m}_c - \mathbf{M}_c \qquad (30)$$

The second term in **Equation 29** quantifies the total change of texture due to topological changes in the link network ($\mathbf{T}$) and is measured by the difference of texture calculated over appeared and disappeared half-links only (numbers $n_a$ and $N_d$, respectively):

$$\mathbf{T} = \mathbf{m}_a - \mathbf{M}_d \qquad (31)$$

$\mathbf{T}$ can be further broken down by gathering the terms corresponding to the topological changes associated with each specific elementary cell process $'P'$ quantified with tensor $\mathbf{P}$:

$$\mathbf{T} = \sum_P \mathbf{P} \qquad (32)$$

$$\mathbf{P} = \sum_{n_P} \frac{1}{2} w_P \vec{l} \otimes \vec{l} - \sum_{N_P} \frac{1}{2} W_P \vec{L} \otimes \vec{L} \qquad (33)$$

where $n_P$ and $N_P$ represent the numbers of half-links that respectively appeared and disappeared through cell process $P$ between initial and final states. Note that $P$ is a mute variable here that either designate $D$ (divisions), $R$ (rearrangements), $A$ (apoptoses/delaminations), $N$ (integrations/nucleation), $C$ (fusions /coalescences), and $J$ (inward/outward fluxes), see **Equation 15**.

Both $\mathbf{G}$ and $\mathbf{T}$ are discussed below. Together, they encompass all changes in texture, so that the balance equation reads:

$$\Delta \mathbf{M} = \mathbf{G} + \mathbf{T} \qquad (34)$$

The change of texture of the patch of cells between two states therefore contains all the information about geometrical and topological changes at scales ranging for the individual cell to the entire cell patch. The extraction of this information relies on our ability to track the time evolution of every link between the two states and to categorize them.

## B.2 Coarse-grained deformation gradient tensor: $\mathbf{F}$

In a cellular material, the material state can be characterized by the discrete links joining the centers of mass of neighbor cells. Together, these links constitute a network of irregular triangles (**Video 1c**). By following those links over time, the material deformation can be characterized as well, and related with elementary cell processes (**Video 1c–h**) (**Graner et al., 2008**; **Etournay et al., 2015**; **Tlili et al., 2015**).

During growth and morphogenesis, the tissue patches change in shape and size, and their deformation can be characterized in situ by the changes in length and direction of the links that are conserved between successive images.

We now show how we can relate continuum mechanics quantities and the statistical description of a deformable cellular material by building a deformation gradient tensor $\mathbf{F}$, not at the cell scale, but rather directly coarse-grained at the cell patch scale (based on the

changes of the material links between two states) to quantify the deformation of the material, and by relating it to $\mathbf{G}$.

## B.2.1 A simple example: homogeneous geometric deformation

We now want to detail how we determine the average deformation gradient tensor of a deforming cell patch $\mathbf{F}$ in the general case, *Equation 52*. For that purpose, we first illustrate our approach on the simple, hypothetical case of a homogeneous patch deformation, *Equation 40*. We emphasize that our approach is general and does not rely on such simplification.

In the present paragraph, for the sake of pedagogy, we temporarily assume that the deformation gradient, noted $\mathbf{F}_h$, is constant over the whole patch, and that all links are conserved between the initial and final states, implying the conservation of their numbers ($N = n$) and weights ($W = w$ for each link). In this very specific case, the final and initial configurations of each link of the patch are simply related as follows:

$$\vec{l} = \mathbf{F}_h \vec{L} \tag{35}$$

which is the discrete equivalent of *Equation 21*. Therefore, multiplying both sides to the right by $w\vec{l}^T$:

$$w\vec{l} \otimes \vec{l} = w\mathbf{F}_h \vec{L} \otimes \vec{l} \Rightarrow \quad \sum_n w\vec{l} \otimes \vec{l} = \sum_n w\,\mathbf{F}_h \vec{L} \otimes \vec{l} \tag{36}$$

after having summed over all half-links of the patch. Then, $\mathbf{F}_h$ being here constant over the patch, it can be factored out of the sum:

$$\sum_n w\vec{l} \otimes \vec{l} = \mathbf{F}_h \sum_n w\vec{L} \otimes \vec{l} \tag{37}$$

Using *Equation 35* in the right hand side, multiplying by $\frac{1}{2}$, factoring again $\mathbf{F}_h$ out of the sum and using the conservation of link numbers and weights yields:

$$\sum_n \frac{1}{2} w\vec{l} \otimes \vec{l} = \mathbf{F}_h \left( \sum_N \frac{1}{2} W\ \vec{L} \otimes \vec{L} \right) \mathbf{F}_h^T \tag{38}$$

On the left and right hand sides, we now have $\mathbf{m}$ and $\mathbf{M}$, respectively:

$$\mathbf{m} = \mathbf{F}_h \mathbf{M} \mathbf{F}_h{}^T \tag{39}$$

Therefore, $\mathbf{F}_h$ fully determines the final texture $\mathbf{m}$, like it determines the tensor of geometric changes $\mathbf{G}$ (*Equation 30*)

$$\mathbf{G} = \mathbf{m}_c - \mathbf{M}_c = \mathbf{m} - \mathbf{M} = \mathbf{F}_h \mathbf{M} \mathbf{F}_h^T - \mathbf{M} \tag{40}$$

Note that in this very simple example, the total textures $\mathbf{M}, \mathbf{m}$ and the conserved textures $\mathbf{M}_c, \mathbf{m}_c$ coincide since we assumed no topological changes. It is important to note that $\mathbf{F}_h$ can be obtained from the knowledge of the initial and final configurations of the link network in the patch (*Equation 37*):

$$\mathbf{F}_h = \left( \sum_n w\vec{l} \otimes \vec{l} \right) \left( \sum_n w\vec{L} \otimes \vec{l} \right)^{-1} \tag{41}$$

## B.2.2 General deformation of a cellular material

In a general deformation of a patch of cells, the deformation is not homogeneous over the patch and some links appear and disappear during the process. In this realistic case, we show here that it is still possible to define a deformation gradient tensor $\mathbf{F}$ characterizing the patch deformation between the initial and final states, provided that a significant fraction of the links are conserved.

As mentioned earlier, we do not try to estimate the deformation at the triangle level but we rather directly define a coarse-grained deformation tensor at the patch scale. To do so, in a first step, we generalize *Equation 41* by defining a tensor $\mathbf{F}_1$ through the following relationship:

$$\mathbf{F}_1 = \left( \sum_{n_c} w_c \vec{l} \otimes \vec{l} \right) \left( \sum_{n_c} w_c \vec{L} \otimes \vec{l} \right)^{-1} \tag{42}$$

Importantly, the sum here is only carried out over conserved half-links for which we have $n_c = N_c$ and $w_c = W_c$ for each link. Note that $\mathbf{F}_1$ definition both involves final and initial links $\vec{l}$ and $\vec{L}$, but it breaks the symmetry between them, $\vec{l}$ appearing three times, $\vec{L}$ only once. In a second step, we define a similar tensor $\mathbf{F}_0$, that mirrors the symmetry breaking in $\mathbf{F}_1$ as follows:

$$\mathbf{F}_0 = \left( \sum_{n_c} w_c \vec{l} \otimes \vec{L} \right) \left( \sum_{n_c} w_c \vec{L} \otimes \vec{L} \right)^{-1} \tag{43}$$

In $\mathbf{F}_0$ expression, $\vec{L}$ appears three times, $\vec{l}$ only once. $\mathbf{F}_0$ and $\mathbf{F}_1$ enable the derivation of the final conserved texture $\mathbf{m}_c$ from the initial one $\mathbf{M}_c$, whatever $\mathbf{M}_c$. Using successively *Equations 42,43*:

$$\mathbf{m}_c = \frac{1}{2} \sum_{n_c} w_c \vec{l} \otimes \vec{l} = \frac{1}{2} \mathbf{F}_1 \left( \sum_{n_c} w_c \vec{L} \otimes \vec{l} \right) = \frac{1}{2} \mathbf{F}_1 \left( \sum_{n_c} W_c \vec{L} \otimes \vec{L} \right) \mathbf{F}_0{}^{\mathrm{T}} = \mathbf{F}_1 \mathbf{M}_c \mathbf{F}_0{}^{\mathrm{T}} \tag{44}$$

$\mathbf{m}_c$ and $\mathbf{M}_c$ being symmetric tensors, one has:

$$\mathbf{m}_c = (\mathbf{m}_c)^{\mathrm{T}} = (\mathbf{F}_1 \mathbf{M}_c \mathbf{F}_0{}^{\mathrm{T}})^{\mathrm{T}} = \mathbf{F}_0 \mathbf{M}_c \mathbf{F}_1{}^{\mathrm{T}} \tag{45}$$

Therefore

$$\mathbf{F}_1 \mathbf{M}_c \mathbf{F}_0{}^{T} = \mathbf{F}_0 \mathbf{M}_c \mathbf{F}_1{}^{T} \tag{46}$$

Both expressions in *Equation 46* are quite similar to *Equation 39*, but not as symmetric. We now aim at defining a deformation gradient tensor $\mathbf{F}$ restoring this symmetry. We introduce $\mathbf{F}$ as the intermediate between $\mathbf{F}_0$ and $\mathbf{F}_1$ such that it satisfies:

$$\mathbf{F}^2 = \mathbf{F}_0 \mathbf{F}_1 \tag{47}$$

*Equation 46* being true for any symmetric matrix $\mathbf{M}_c$, let us temporarily assume that $\mathbf{M}_c$ can vary independently of $\mathbf{F}_0$ and $\mathbf{F}_1$. In that specific case, one can show that *Equation 46* implies that $\mathbf{F}_0$ and $\mathbf{F}_1$ are proportional, namely that:

$$\frac{\mathbf{F}_1}{\|\mathbf{F}_1\|} = \frac{\mathbf{F}_0}{\|\mathbf{F}_0\|} \tag{48}$$

Using *Equation 48* in *Equation 47* yields:

$$\mathbf{F} = \left(\frac{\|\mathbf{F}_1\|}{\|\mathbf{F}_0\|}\right)^{\frac{1}{2}} \mathbf{F}_0 = \left(\frac{\|\mathbf{F}_0\|}{\|\mathbf{F}_1\|}\right)^{\frac{1}{2}} \mathbf{F}_1 \tag{49}$$

Then, using *Equation 49* in *Equation 44*:

$$\mathbf{m}_c = \mathbf{F}_0 \mathbf{M}_c \mathbf{F}_1^T = \mathbf{F}_0 \mathbf{M}_c \left(\frac{\|\mathbf{F}_1\|}{\|\mathbf{F}_0\|} \mathbf{F}_0\right)^T = \left[\left(\frac{\|\mathbf{F}_1\|}{\|\mathbf{F}_0\|}\right)^{\frac{1}{2}} \mathbf{F}_0\right] \mathbf{M}_c \left[\left(\frac{\|\mathbf{F}_1\|}{\|\mathbf{F}_0\|}\right)^{\frac{1}{2}} \mathbf{F}_0\right]^T \tag{50}$$

Using *Equation 49*, we finally get:

$$\mathbf{m}_c = \mathbf{F} \mathbf{M}_c \mathbf{F}^T \tag{51}$$

and the symmetry in $\mathbf{F}$ of *Equation 39* has been restored. Therefore, knowing the initial conserved texture $\mathbf{M}_c$, $\mathbf{F}$ fully determines the final conserved texture $\mathbf{m}_c$, like it determines the tensor of geometric changes $\mathbf{G}$ (*Equation 30*)

$$\mathbf{G} = \mathbf{m}_c - \mathbf{M}_c = \mathbf{F} \mathbf{M}_c \mathbf{F}^T - \mathbf{M}_c \tag{52}$$

Although the structure of the cellular network (characterized by $\mathbf{M}_c$), and its deformation (characterized by $\mathbf{F}_0$ and $\mathbf{F}_1$), are independent (imagine a medium undergoing the same deformation with different tilings of it), here $\mathbf{F}_0$ and $\mathbf{F}_1$ are not independent of $\mathbf{M}_c$ because their definitions involve the links of the cellular patch in its initial configuration, $\vec{L}_{ik}$. Therefore, for a general deformation, $\mathbf{F}_0$ and $\mathbf{F}_1$ do not necessarily satisfy *Equation 48*, and the last equality of *Equation 52* is only approximate. Interestingly, in the particular case of small deformations ($\|\mathbf{F} - \mathbb{1}\| \ll 1$), *Equations 51 and 52* become exact to linear order, as $\mathbf{F} = \mathbf{F}_{\text{small}}$ with $\mathbf{F}_{\text{small}} = (\mathbf{F_0} + \mathbf{F_1})/2$.

To implement *Equation 52* which is later used in *Equation 59*, and in the following, we define $\mathbf{F}$ as the geometric average of $\mathbf{F}_0$ and $\mathbf{F}_1$, namely $\mathbf{F} = (\mathbf{F_0}\mathbf{F_1})^{1/2}$, and calculate it numerically with Matlab. We checked that $\mathbf{F}$ is always real and yields results that are comparable to its approximated version $\mathbf{F}_{\text{small}}$, but more accurate as deformations are finite rather than infinitesimal. Indeed, we found that *Equation 52* is satisfied in very good approximation: defining the error in a given patch at a given time as $\delta = \|\mathbf{G} - (\mathbf{F}\mathbf{M}_c\mathbf{F}^T - \mathbf{M}_c)\|/\|\mathbf{G}\|$, we found $\langle\delta\rangle = 2.2 \ 10^{-4}$, while we found $\langle\delta\rangle = 1.1 \ 10^{-3}$ when using $\mathbf{F}_{\text{small}}$ (averages performed over space and time on the dorsal thorax of *Videos 3,4*).

*Equation 48* therefore defines a coarse-grained deformation gradient tensor of the cell patch between the initial and final states, $\mathbf{F}$. It is defined for a general deformation that can be large, heterogeneous over the patch and involving topological changes. Being solely based on the conserved links, it only requires that enough links are conserved between the initial and final states. Note that $\mathbf{F}$ is not symmetric in general; it characterizes both the stretch and the rotation of the deforming cell patch.

Although similar to *Equation 40* obtained in the simple case of homogeneous geometric deformations, *Equation 52* has a much broader range of application since it characterizes a general deformation of the cell patch. *Equation 52* relates the $\mathbf{G}$ tensor to the coarse-grained deformation gradient tensor $\mathbf{F}$ that characterizes the deformation of a cellular material and that was derived from our statistical description. In the following, starting from $\mathbf{G}$ which has units in squared length, we show how we can build a dimensionless quantity analogous to a strain tensor to quantify the local deformation of a patch of cellular material. Importantly, the balance between $\mathbf{G}$ and the tensors quantifying the contribution of cell processes (*Equation 15*) is independent of the approximation in *Equation 52* and remains exact at arbitrary finite deformations and regardless of the size of the cell patch.

# C Defining and linking material strains to cell process strains

In the previous section, we have introduced a statistical description based on the links connecting cell elements to define a coarsed-grained deformation gradient tensor $\mathbf{F}$ quantifying the deformation of a patch of cellular material (*Equation 47*). However, there is no simple relationship between $\mathbf{F}$ and the elementary cell processes making up the patch deformation.

In the present section, thanks to the relationship linking the change in texture $\Delta \mathbf{M}$ to geometrical ($\mathbf{G}$) and topological ($\mathbf{T}$) changes in the cellular material (*Equation 34*) and the relationship between $\mathbf{G}$ and $\mathbf{F}$ (*Equation 52*), we define a proper strain tensor $\mathbf{G}^*$ to characterize the total material strain (section C.1) and to relate it to the elementary cell processes making up material deformation, namely cell size and shape changes, divisions, rearrangements, apoptoses/delaminations, integrations, fusions, and inward/outward fluxes (section C.2). To do so, we define dimensionless symmetric tensors ($\mathbf{P}^*$) that quantify the strains associated with each cell process $P$. We obtain a balance equation where the cell process strains $\mathbf{P}^*$ add up to the total material strain $\mathbf{G}^*$, which incidentally enables to identify the material geometrical ($\mathbf{S}^*$) and topological ($\mathbf{T}^*$) strains (section C.3).

## C.1 Strain tensor in a deformable cellular material: $\mathbf{G}^*$

### C.1.1 Building dimensionless symmetric tensors

For any tensor $\mathbf{Q}$ expressed in length squared, such as the terms in *Equations 32,34* which typically scale with the number of cells and their squared size, we can define its dimensionless counterpart rescaled by the initial conserved texture of the cell patch $\frac{1}{2}\mathbf{Q}\mathbf{M}_c^{-1}$. Note that the factor $\frac{1}{2}$ is due to the identification with continuum mechanics quantities such as deformation (see *Equations 23,52* ). Note also that even when both $\mathbf{Q}$ and $\mathbf{M}_c$ are symmetric, in general $\frac{1}{2}\mathbf{Q}\mathbf{M}_c^{-1}$ is not, and that two non-parallel conserved links are sufficient to make $\mathbf{M}_c$ invertible.

In this work, we focus on relating the changes in tissue size and shape to cell processes. The tissue rotation, which is not studied nor required in the present study, could be addressed as well if needed. Based on our statistical description (34), we therefore aim at defining a symmetric strain tensor for the tissue, and relate it to symmetric tensors characterizing the contribution of each elementary cell process to tissue strain. We therefore directly build a dimensionless symmetric tensor corresponding to quantity $\mathbf{Q}$ as follows:

$$\tilde{\mathbf{Q}} = \frac{1}{4}[\mathbf{Q}\mathbf{M}_c^{-1} + \mathbf{M}_c^{-1}\mathbf{Q}^T] \tag{53}$$

Like a deformation, $\tilde{\mathbf{Q}}$ is now expressed without units, e.g. as percents.

### C.1.2 Building $\mathbf{F}$ based tensors

We define here three additional tensors based on the coarse-grained deformation gradient $\mathbf{F}$ that will naturally appear in what follows: $\overline{\mathbf{F}}$, $[\mathbf{F}]$ and $\Psi$. Although $\mathbf{F}$ is already dimensionless, in the following it will be convenient to have defined the following quantity:

$$\overline{\mathbf{F}} = \mathbf{M}_c^{-1}\,\mathbf{F}\mathbf{M}_c \tag{54}$$

By isolating $\mathbf{F}$ and using the commutator of two tensors $[\mathbf{A},\ \mathbf{B}] = \mathbf{A}\mathbf{B} - \mathbf{B}\mathbf{A}$, we get:

$$\begin{aligned}
\overline{\mathbf{F}} &= \mathbf{M}_c^{-1}\left(\mathbf{FM}_c + \mathbf{M}_c\mathbf{F} - \mathbf{M}_c\mathbf{F}\right) \\
&= \mathbf{F} + \mathbf{M}_c^{-1}\left(\mathbf{FM}_c - \mathbf{M}_c\mathbf{F}\right) \\
&= \mathbf{F} + \mathbf{M}_c^{-1}\left[\mathbf{F}, \mathbf{M}_c\right] \\
&= \mathbf{F} + [\mathbf{F}]
\end{aligned} \tag{55}$$

where we have introduced the new quantity:

$$[\mathbf{F}] = \mathbf{M}_c^{-1}\left[\mathbf{F}, \mathbf{M}_c\right] \tag{56}$$

$[\mathbf{F}]$ is therefore a dimensionless tensor which represents the difference between $\overline{\mathbf{F}}$ and $\mathbf{F}$. It only vanishes non trivially when $\mathbf{F}$ and $\mathbf{M}_c$ commute, and it will be used in **Equations 59, 62**. Note that:

$$[\mathbf{F}]^T = [\mathbf{F}, \mathbf{M}_c]^T\mathbf{M}_c^{-1} = [\mathbf{M}_c, \mathbf{F}^T]\,\mathbf{M}_c^{-1} \tag{57}$$

Finally, we define the tensor $\Psi$, a quantity with the same unit as $\mathbf{G}$ which expresses this commutator and that will be used in **Equations 60–63**:

$$\Psi = \mathbf{F}[\mathbf{M}_c, \mathbf{F}^T] \tag{58}$$

Like $[\mathbf{F}]$, $\Psi$ is linked to the non-commutation between $\mathbf{M}_c$ and $\mathbf{F}$, which occurs when $\mathbf{F}$ contains some rotation or when its stretch is not aligned with the initial direction of cell elongation, namely with $\mathbf{M}_c$ eigenvectors. One can show that it also contains a part of a 'corotational derivative' which makes the rotational transport of deformation objective (**Malvern, 1969**). It is not symmetric and not dimensionless, having the same units as $\mathbf{M}_c$, namely square length.

## C.1.3 Total material strain $\mathbf{G}^*$

We now derive our final expression of the total tissue strain tensor $\mathbf{G}^*$ that will be used throughout this study to quantify local deformation of the tissue. Note that 'total' here means that $\mathbf{G}^*$ will quantify the tissue strain regardless of its origin, namely gathering geometrical and topological strains together. Using **Equations 52, 57**:

$$\begin{aligned}
\frac{1}{2}\mathbf{GM}_c^{-1} &\approx \frac{1}{2}\left(\mathbf{FM}_c\mathbf{F}^T\mathbf{M}_c^{-1} - \mathbb{1}\right) \\
&= \frac{1}{2}\left(\mathbf{F}\overline{\mathbf{F}}^T - \mathbb{1}\right) \\
&= \frac{1}{2}\left(\mathbf{FF}^T - \mathbb{1}\right) + \frac{1}{2}\mathbf{F}[\mathbf{F}]^T
\end{aligned} \tag{59}$$

Using **Equations 23, 57, 58**:

$$\frac{1}{2}\mathbf{GM}_c^{-1} \approx \mathbf{E}^* + \frac{1}{2}\left(\mathbf{F}[\mathbf{M}_c, \mathbf{F}^T]\right)\mathbf{M}_c^{-1} = \mathbf{E}^* + \frac{1}{2}\Psi\mathbf{M}_c^{-1} \tag{60}$$

Finally, using definition (53), we get:

$$\tilde{\mathbf{G}} \approx \mathbf{E}^* + \tilde{\Psi} \tag{61}$$

We find that $\tilde{\mathbf{G}}$ is the sum of an actual strain tensor $\mathbf{E}^*$ (**Equation 23**) and of the dimensionless symmetric version of $\Psi$, namely $\tilde{\Psi}$ which reads:

$$\tilde{\Psi} = \frac{1}{4}(\mathbf{F}[\mathbf{F}]^T + [\mathbf{F}]\mathbf{F}^T) \tag{62}$$

$\tilde{\Psi}$ vanishes when $\mathbf{F}$ and $\mathbf{M}_c$ commute, which occurs when $\mathbf{F}$ both has its rotation tensor equal to $\mathbb{1}$, and its stretch tensor commutes, i.e. shares the same eigenvectors, with $\mathbf{M}_c$. In other words, $\tilde{\Psi}$ vanishes when $\mathbf{F}$ both contains no rotation and has its stretch along the same axis as the average cell elongation axis within the patch.

Therefore, in order to use an actual strain tensor to properly quantify local changes in size and shape of the cellular material, namely independently of rigid body movements including rotation, we define the $\mathbf{G}^*$ tensor as follows:

$$\mathbf{G}^* = \tilde{\mathbf{G}} - \tilde{\Psi} \tag{63}$$

and from *Equation 61* we have:

$$\mathbf{G}^* \approx \mathbf{E}^* = \frac{1}{2}(\mathbf{F}\mathbf{F}^T - \mathbb{1}) \tag{64}$$

$\mathbf{G}^*$ is therefore a proper strain tensor, formally very similar to the widely used Green-Lagrange strain tensor (*Equations 22,23*). When deformations are small ($\|\mathbf{G}^*\| \ll 1$), *Equation 64* becomes exact to linear order, and $\mathbf{G}^*$ isotropic part (its trace) quantifies the dilation of the patch of cellular material between its initial and its final states, namely tissue change in size. Its anisotropic part (its deviator) quantifies the local contraction-elongation (CE) (or shear) of the cellular material, and we call it tissue morphogenesis in our case.

## C.2 Linking material strain to cell processes

In this section, we relate the newly defined tissue strain tensor $\mathbf{G}^*$ (*Equation 63*) to all elementary cell processes, namely cell size and shape changes, divisions, rearrangements, apoptoses/delaminations, integrations, fusions, and inward/outward fluxes. This will enable the definition of meaningful tensors to characterize quantitatively the strains associated with each cell process.

First, now that we have shown that one can build a proper strain tensor to characterize tissue deformation based on $\mathbf{G}$, we rewrite *Equation 34* using *Equation 32*, and we use *Equation 53* to make all symmetric tensors dimensionless:

$$\tilde{\mathbf{G}} = \Delta\tilde{\mathbf{M}} - \sum_P \tilde{\mathbf{P}} \tag{65}$$

Using *Equation 63* to make $\mathbf{G}^*$ appear, we get:

$$\mathbf{G}^* = \Delta\tilde{\mathbf{M}} - \tilde{\Psi} - \sum_P \tilde{\mathbf{P}} \tag{66}$$

Now we have a first balance equation linking the tissue strain $\mathbf{G}^*$ to tensors quantifying cell processes. Indeed, $\Delta\tilde{\mathbf{M}}$ is related to the cell size and shape changes in the patch between the initial and final states (*Graner et al., 2008*), while the tensors $\tilde{\mathbf{P}}$ quantify the changes in the link network topology due to each cell process $P$. However, to have tensors that meaningfully and unambiguously characterize the contribution of each elementary cell process to tissue strain requires some additional steps detailed below.

When the number of half-links in the patch varies from $N$ to $n$ between the initial and final states, the variation in texture has two entangled contributions, one due to the cell size and shape changes, and one due to the changes in number of half-links. They can be disentangled by rewriting $\Delta\tilde{\mathbf{M}}$ as follows:

$$\Delta\tilde{\mathbf{M}} = \tilde{\mathbf{m}} - \tilde{\mathbf{M}} = \left(\frac{N}{n}\tilde{\mathbf{m}} - \tilde{\mathbf{M}}\right) + \left(\frac{n-N}{n}\right)\tilde{\mathbf{m}} \tag{67}$$

The first term in **Equation 67**, by renormalizing the final texture $\tilde{\mathbf{m}}$ by the ratio $\frac{N}{n}$, quantifies the cell size and shape changes independently of the changes in half-link number in the patch. The second term in **Equation 67** involves the total difference in half-link numbers in the cell patch between the initial and final states, $\Delta N = n - N$. Importantly, this difference can be decomposed into the sum of the changes in half-link number due to each elementary cell process $P$ that impacts the network topology ($\Delta N_P = n_P - N_P$):

$$\Delta N = n - N = \sum_P (n_P - N_P) = \sum_P \Delta N_P \tag{68}$$

$\Delta N_P$ therefore quantifies the contribution of each topological process to the total change in half-link number within the cell patch. $\Delta N_P > 0$ for divisions, integrations or inward fluxes since those cell processes correspond to a net creation of half-links; $\Delta N_P < 0$ for apoptoses/delaminations, fusions or outward fluxes since those cell processes correspond to a net loss of half-links; and $\Delta N_P \approx 0$ for rearrangements since they mostly conserve the number of links, the only exception involving cells at the image boundaries.

Combining **Equations 67,68** one can therefore rewrite **Equation 66** as:

$$\mathbf{G}^* = \underbrace{\left(\frac{N}{n}\tilde{\mathbf{m}} - \tilde{\mathbf{M}} - \tilde{\mathbf{\Psi}}\right)}_{\mathbf{S}^*} + \sum_P \underbrace{\left(\frac{\Delta N_P}{n}\tilde{\mathbf{m}} - \hat{\mathbf{P}}\right)}_{\mathbf{P}^*} \tag{69}$$

where each $\frac{\Delta N_P}{n}\tilde{\mathbf{m}}$ term related to the change in half-link number due to each topological process $P$ has been gathered with its corresponding tensor $\tilde{\mathbf{P}}$ in the sum, thereby gathering all terms impacted by topological changes.

We can now define our final expressions for the tensors representing the strains of elementary cell processes contributing to tissue strain:

- the first term of **Equation 69** defines the tensor quantifying the contribution of changes of cell size and shape to the tissue strain $\mathbf{G}^*$. It represents the geometrical strain in the tissue and is called $\mathbf{S}^*$ (see section C.3.1).

- the second term of **Equation 69** represents the total topological strain in the tissue and is called $\mathbf{T}^*$. Each term of the sum over topological processes defines the tensor $\mathbf{P}^*$ quantifying the contribution of the cell process $P$ to total tissue topological strain $\mathbf{T}^*$ (see section C.3.2).

With those new notations **Equations 32,69** therefore read:

$$\mathbf{G}^* = \mathbf{S}^* + \mathbf{T}^* \tag{70}$$

$$\mathbf{T}^* = \sum_P \mathbf{P}^* \tag{71}$$

The total tissue strain tensor $\mathbf{G}^*$ can therefore be expressed as the sum of the geometrical strain tensor $\mathbf{S}^*$ and of the total topological strain tensor $\mathbf{T}^*$. The latter can be further broken down into elementary topological strains tensors $\mathbf{P}^*$ associated with each cell process changing the link network topology. We detail and comment these new definitions in the next sections.

## C.3 Material geometrical and topological strains, cell process strains

### C.3.1 Geometrical strain $\mathbf{S}^*$

The first term in **Equation 69** defines the $\mathbf{S}^*$ tensor as follows:

$$\mathbf{S}^* = \frac{N}{n}\tilde{\mathbf{m}} - \tilde{\mathbf{M}} - \tilde{\Psi} \tag{72}$$

This tensor quantifies the average cell size and shape changes in the patch (via its trace and deviator, respectively) and their contribution to the total patch deformation (or strain), between the initial and final states. Being solely impacted by the changes of cell size and shape in the patch, $\mathbf{S}^*$ therefore represents the geometrical strain of the tissue, in opposition to the topological strain ($\mathbf{T}^*$) that is solely impacted by changes of cell topology in the patch (see section C.3.2). Thus, in cases where tissue deformation solely arises from cell shape and size changes (or geometrical deformations), one has the exact equality $\mathbf{G}^* = \mathbf{S}^*$, as illustrated in simulations of **Figure 2a**, **Figure 2—figure supplement 1e** and **Video 2a,j**.

As explained in section C.2, $\mathbf{S}^*$ is independent of changes in the numbers of links or cells. This point is well illustrated by our simulations of cell patches deforming through divisions, delaminations, integrations, significantly changing the number of cells in the patch, but where cell sizes and shapes almost totally recovered to their initial states, one finds $\mathbf{S}^* \approx 0$ (**Figure 2b, d** and **Figure 2—figure supplement 1a**, **Video 2b-e**).

Importantly, $\mathbf{S}^*$ is independent of rigid body movements including rotations, as a strain tensor should. This means that in the case of a pure rotation, $\mathbf{S}^*$ vanishes; in the case of a stretch plus a rotation, $\mathbf{S}^*$ is the same as for a pure stretch. A direct consequence of this property is that $\mathbf{S}^*$ depends on the path followed between the initial and final states. These two properties are illustrated by our last two simulations where: (i) a patch elongated along the $x$ axis made of cells elongated in the same direction is rotated anti-clockwise to result in the exact same patch oriented along the $y$ axis (**Figure 2—figure supplement 1d**, **Video 2i**); (ii) The same initial patch of case (i) now undergoes a contraction-elongation along the $y$ axis, resulting in a patch with same aspect ratio but now oriented along the $y$ axis, very similar to the rotated one in (ii) (**Figure 2—figure supplement 1e**, **Video 2j**). Although the initial and final states of those two simulations are almost identical, in case (i) both tissue strain $\mathbf{G}^*$ and geometrical strain $\mathbf{S}^*$ are null (within segmentation errors), in sharp contrast to case (ii) where significant tissue and geometrical strains are measured leading to $\mathbf{G}^* = \mathbf{S}^*$ (within segmentation errors). $\mathbf{S}^*$ definition (**Equation 72**) is therefore one of the main innovations and improvements of this formalism. This new definition also has important consequences on the definitions of all other processes, as described in Section C.3.2.

### C.3.2 Topological strain $\mathbf{T}^*$ and cell process strains $\mathbf{P}^*$

The second term in **Equation 69** defines the $\mathbf{P}^*$ tensors as follows:

$$\mathbf{P}^* = \frac{\Delta N_P}{n}\tilde{\mathbf{m}} - \tilde{\mathbf{P}} \tag{73}$$

where $\mathbf{P}^*$ designate the tensor quantifying the strains associated to the topological changes in the link network associated with either divisions ($\mathbf{D}^*$), rearrangements ($\mathbf{R}^*$), apoptoses/delaminations ($\mathbf{A}^*$), integrations ($\mathbf{N}^*$), fusions ($\mathbf{C}^*$) and inward/outward fluxes ($\mathbf{J}^*$) between initial and final states. **Equation 69** shows that each of these tensors represents the contribution of each cell process to the total tissue strain $\mathbf{G}^*$.

Importantly, in order to fully disentangle cell size and shape changes from topology changes in the patch, our definition of $\mathbf{S}^*$ correctly characterizes changes in cell shapes and sizes, regardless of changes in cell numbers (**Equation 67**). For each tensor quantifying topological cell process $P$, this resulted in the extra term $\frac{\Delta N_P}{n}\tilde{\mathbf{m}}$ related to the change in half-link number

due to this cell processes $\Delta N_P$ (**Equation 68**). Each tensor $\mathbf{P}^*$ thereby gathers all 'topological' terms due to process $P$ (**Equation 73**).

All these contributions add up to the tensor $\mathbf{T}^*$ that therefore gathers all the topological changes in the link network between initial and final states, regardless of their origin (**Equation 71**):

$$\mathbf{T}^* = \sum_P \mathbf{P}^* = \mathbf{D}^* + \mathbf{R}^* + \mathbf{A}^* + \mathbf{N}^* + \mathbf{C}^* + \mathbf{J}^* \tag{74}$$

where we have made explicit every tensor in the sum. Being solely impacted by the changes of cell topology in the patch, $\mathbf{T}^*$ therefore represents the topological strain of the tissue, in opposition to the geometrical strain ($\mathbf{S}^*$) that is solely impacted by changes of cell size and shape in the patch (see section C.3.1). Together with the geometrical strain, it amounts to the total strain of the patch of cellular material (**Equation 70**):

$$\mathbf{G}^* = \mathbf{S}^* + \mathbf{T}^* = \mathbf{S}^* + \mathbf{D}^* + \mathbf{R}^* + \mathbf{A}^* + \mathbf{N}^* + \mathbf{C}^* + \mathbf{J}^* \tag{75}$$

This balance equation is the key equation of our formalism as it simply expresses the tissue strain between two arbitrary states as the sum of tensors quantifying the respective strains due to all cell processes: cell size and shape changes, cell divisions, cell rearrangements, cell apoptoses/delaminations, cell integrations/nucleations, cell coalescences/fusions and cell inward/outward fluxes. Note that all tensors, including $\mathbf{G}^*$, are determined separately, and that we always check that the balance of **Equation 75** is satisfied.

Finally, since by construction each change in links is attributed to one and only one process, each cell process tensor is unambiguously disentangled from the others; this constitutes the last major improvement of this formalism. Indeed, directly resulting from the additivity of each contribution $\mathbf{P}^*$ to the total topological strain, when the cell patch deforms solely through one topological cell process, this process equals $\mathbf{G}^*$. Taking the apoptoses/delaminatione $\mathbf{A}^*$ as example, one has $\mathbf{G}^* = \mathbf{T}^* = \mathbf{A}^*$ and $\mathbf{S}^* = 0$. In this example, the tissue strain is all accounted for by the topological strain due to apoptoses/delaminations, the geometrical strain being null. This point is illustrated for each process in our simulations by testing their measurement (**Figure 2** and **Figure 2—figure supplement 1**, **Video 2**). Note that most of our simulations being disordered to be realistic, we cannot prevent some occurrences of rearrangements and cell size and shape changes, hence their residual measured contributions in our balances.

Below, we comment each cell process strain more specifically:

- the contribution of divisions. It includes the newly appeared link between the two 'sister' cells (link in dark green in **Figure 1a,b**), as well as the changes in link network topology that involve the neighboring cells (links in green in **Figure 1a,b**). Note that, in parallel to $\mathbf{D}^*$, we build the tensor $\mathbf{D}_0^*$ which is solely based on the links between sister cells. $\mathbf{D}_0^*$ is not a contribution of divisions to morphogenesis but rather a way to quantify tensorially (ie, with amplitude, anisotropy and direction) the average orientation of cell divisions. When many cells divide in the same direction, $\mathbf{D}_0^*$ is represented with a large bar in this direction. Conversely, a small bar for $\mathbf{D}_0^*$ reflects either a low proliferation, or a disorder in division orientation.

- the contribution of links which rearrange. If needed, this can be subdivided into a sub-process associated with simple rearrangements with 4 cells, a sub-process for rearrangements involving rosettes with 5 cells, and so on for 6 or more cells. Their contributions naturally add up, by construction of the formalism. This is one of the main reasons to have chosen links, which are the very objects whose creation and disappearance do define and characterize a cell rearrangement.

- the contribution of delaminations and apoptoses. If needed, this can be subdivided into sub-processes, each associated with a different biological origin, for instance to distinguish simple extrusions from extrusions associated to apoptoses; or apoptoses which occur for cells with 3, 4 or more sides. Their contributions naturally add up, by construction of our formalism.

- the contribution of new cell integrations or nucleations, which occur here only due to segmentation and tracking mistakes. We check that this term is much smaller than the main contributions, which shows that the segmentation and tracking are accurate enough for the present purpose.

- the contribution of cell fusions, which occur here only due to segmentation and tracking mistakes. Again, we check that this term is much smaller than the main contributions, which shows that the segmentation and tracking are accurate enough for the present purpose.

- the changes in links due to cells exiting or entering the region where they are visible. However, since this term exists only at the sample boundary, where all other processes are barely studied anyway (since their weights vanish, see *Equation 4*), $\mathbf{J}^*$ is not important for the results presented here. Note that our formalism leaves the choice between Eulerian and Lagrangian descriptions. An Eulerian description would add at each grid compartment a flux contribution across the compartment boundaries, and our formalism enables to measure it separately ($\mathbf{J}_c$ in *Equation 78*). A Lagrangian description based on patch tracking, which we choose here, does not have such flux. Note also that the formalism is meant to extract relevant information from cell-cell links, to apply to a confluent cell assembly. Still, $\mathbf{J}^*$ systematically takes into account all links between cells at the tissue boundary, and all their possible changes. The formalism is thus designed to rigorously take into account all links between cells, and all their possible changes, whether these cells are at the tissue boundary or not. It applies to tissues with boundaries, even with internal ones like in the case of actual holes in the tissue, or in the case of holes in the skeletonized images due to segmentation problems.

### C.3.3 Defining strain rate tensors

So far, we have defined dimensionless strain tensors to quantify the deformation of the cellular material between two arbitrary initial and final states, and we have related it to elementary cell processes. Because the time during which these deformations occur matters, we now define rate tensors that will be expressed in h$^{-1}$.

Although what precedes is valid for an arbitrary tissue deformation or proliferation, the accuracy in $\mathbf{G}^*$ determination is improved by maximizing the number of links conserved between the two images compared. We thus first determine it between two successive images (separated by a time interval $\delta t$, here 5 min). This is an acceptable choice since here $\delta t$ is a small enough sampling time that most links are conserved, $\mathbf{M}_c$ is always well defined and invertible (except at boundaries), $\mathbf{G}^*$ can be accurately determined, the tracking is reliable, and each process is correctly described.

For any quantity $Q$, we determine $\delta Q$, as the difference between two successive images, then define the rate $\dot{Q}$ as follows:

$$\dot{Q} = \frac{\delta Q}{\delta t} \tag{76}$$

We then average those rates over a morphogenetic timescale. Here we average them over typically 2 h (*Figure 4h–k*, *Videos 4* and *5*), or otherwise 14 h to visualize the average rates of tissue strain and of contributions of elementary cell processes over the whole studied period. Calling $\langle \dot{Q} \rangle$ the time average of $\dot{Q}$ over a given period, and using *Equation 75*, we get:

$$\left\langle \dot{\mathbf{G}}^* \right\rangle = \left\langle \dot{\mathbf{S}}^* \right\rangle + \left\langle \dot{\mathbf{D}}^* \right\rangle + \left\langle \dot{\mathbf{R}}^* \right\rangle + \left\langle \dot{\mathbf{A}}^* \right\rangle + \left\langle \dot{\mathbf{N}}^* \right\rangle + \left\langle \dot{\mathbf{C}}^* \right\rangle + \left\langle \dot{\mathbf{J}}^* \right\rangle \tag{77}$$

For legibility, in the main text and in the Figure captions we only keep the tensor letter, and write *Equation 77* under a simplified form (*Equation 15*):

$$\mathbf{G} = \mathbf{S} + \mathbf{D} + \mathbf{R} + \mathbf{A} + \mathbf{N} + \mathbf{C} + \mathbf{J}$$

Note that in practice, to analyze our movies, we can choose between two descriptions: Eulerian or Lagrangian. We choose a Lagrangian description, with each cell patch $\mathscr{P}$ being

tracked during its deformation (*Figure 3d* and *5a*, *Video 3c* and *5b*), and calculations of instantaneous deformations rates being made between two successive images. When using an Eulerian grid to perform our analysis, we find an extra contribution from the inward/outward flux of links from each grid compartments ($\mathbf{J}_c$):

$$\mathbf{G} = \mathbf{S} + \mathbf{D} + \mathbf{R} + \mathbf{A} + \mathbf{N} + \mathbf{C} + \mathbf{J} + \mathbf{J}_c \qquad (78)$$

This additional term having no biological meaning, in this work we rather use Lagrangian grids for which $\mathbf{J}_c$ vanishes, thereby making the balance equation (*Equation 15*) easier to interpret.

## D Advantages and comparison with literature

To summarize, our formalism quantifies the observed size and shape changes of a deformable cellular material independently of the biological or physical origin of the forces that drive these changes. These forces can be internal ones such as active cytoskeletal forces, viscous dissipation or cytoplasm pressure, or external ones such as interactions with another cell layer or solid substrate. It is written as part of classical mechanics of deformable materials; the latter describes the deformation of patches (usually called 'Representative volume elements' or RVE), large enough that the material appears continuous (i.e. without signatures of cell-scale correlations at patch scale), and small enough so that each measurement is rather homogeneous over the patch, and that it represents a small fraction of the whole material. In this spirit, we determine the deformation self-consistently at the patch scale, i.e. coarse-grained over several cells, through statistics over links between neighbor cell centers. Our formalism is directly written for arbitrary finite deformation, in both 2D and 3D, in analogy with classical continuum mechanics; it is not restricted to homogeneous or small deformations, or to small or large numbers of cells. As they should, deformation measurements vanish for rigid body movements, namely if the tissue translates or rotates as a whole.

As a consequence of our choice, each link change is distributed into one and only one biological process. Each measurement meaningfully and unambiguously quantifies its associated biological process (*Figures 1* and *2*). Our measurements of tissue and cell process strains are obtained by averaging over the relevant biological space and time scales. The quantities we define, expressed in the same relevant unit (dimensionless strain or rate of deformation per unit time), can be separately measured. These quantities add up to yield the growth and morphogenesis at the patch scale (enabling to reconstruct the development at the tissue scale), as reflected in the balance equation (*Equation 15*) where the total material deformation rate is expressed as sum of the respective deformation rates associated with each individual cell process. In addition, since links are the very objects whose creation and disappearance define a change in neighbor relationships (such as cell divisions, rearrangements, or apoptoses/delaminations), our description of these processes is versatile, enabling to deal with four-fold or higher vertices, distinguish cell rearrangements which involve four or more cells, and similarly distinguish apoptoses/delaminations which involve four or more cells.

Our choice has been triply validated. First, by detailed comparison with theoretical predictions for cell shape changes and rearrangements on non-biological cellular networks (*Cheddadi et al., 2011*). Second, by reanalyzing processes (rearrangements, cell shape changes) which conserve cell numbers, in data already published in the preceding paper (*Bosveld et al., 2012*); the present formalism which is more general since it supports changes in the number of cells, recovers the same results. Third, by numerical simulations of cell patches purely deforming via each individual cell process tested one by one (*Figure 2*, *Figure 2—figure supplement 1*).

After averaging over space, time and movies, in the present study each patch represents a large number of links (in the initial image, about 30 cells per patch × 3 links per cell on

average $\times$ 24 images $\times$ 5 movies $\sim10^4$ links, and even more in the following images), so that statistics are good and deformation measurements are accurate. Our measurements vary smoothly in time and space, and in the notum the symmetry of the maps with respect to the midline shows the quality of the signal-to-noise ratio in each measurement (*Figure 3*, *Figure 3—figure supplements 1*, *2* and *Video 4*).

While each of the published approaches, including our preceding articles, has its own advantages, we argue that the present one offers the following advantages. With respect to (*Economou et al., 2013*), which is a complete and rigorous formalism in 1D, we take into account the variable orientations of cell processes by using tensor measurements. With respect to (*Graner et al., 2008*; *Rauzi et al., 2008*; *Blanchard et al., 2009*; *Aigouy et al., 2010*; *Bosveld et al., 2012*), we now measure separately and take into account all processes, including those which vary cell numbers like cell divisions or apoptoses/delaminations.

While the present manuscript was under review, a formalism by (*Etournay et al., 2015*) has been published. Like ours, it estimates the deformations associated with several cell processes, including cell divisions and apoptoses/delaminations which change the cell number. However, there are some differences with our approach, some of which are detailed below.

In (*Etournay et al., 2015*), to estimate tissue deformation and the contributions of cell processes the authors choose to use the sub-cellular unit made by the triangle connecting the three centers of cells sharing a three-fold (or tricellular) vertex. Their motivation is that the deformation of such a triangle can be unambiguously defined and determined. Currently, their formalism has been developed in 2D with triangles, and it could in principle be generalized to 3D using tetrahedrons. Our formalism, although we used it here to study 2D epithelial tissues, has been directly written in a way which is independent of the space dimension, and is therefore valid in both 2D and 3D.

An important difference with our formalism is the existence of an additional term in their balance equation relating local tissue CE (or shear) to cell processes. This term is related to correlations between cell elongation and tissue rotation, or between cell elongation and cell area change. It captures local inhomogeneities of shear in the patch and arises for instance when neighboring cell rows slide with respect to each other. This correlation term appears when averaging over the triangles tiling a patch of tissue. Indeed, in the simple case of tissue patch deforming in the absence of topological changes, the shear of a given triangle is exactly equal to its (corotational) change of elongation. When averaging the shears and elongations of triangles over the patch, the tissue shear (defined as the average triangle shear over the patch) is now equal to the (corotational) change of average triangle elongation, plus this extra correlation term. Therefore, this term already exists without topological changes occurring and remains even when averaging at large scale. When this correlation term contributes significantly to tissue shear, their formalism and ours should yield different measurements of the other terms quantifying cell processes.

Another difference with our approach is that in (*Etournay et al., 2015*) the authors, to simplify the equations, write them in the case of small deformations. Since the time interval between two images is finite rather than infinitesimal, deformations are finite rather than infinitesimal; treating deformations as small might result in errors which accumulate when cumulating deformations over several time intervals. This is why we built our formalism with tensor measurements directly defined for finite deformations.

Finally, when measurements are performed in fixed compartments of a square grid (Eulerian treatment), the inward and outward flux of cells across each compartment boundary adds an additional term in the balance equation ($\mathbf{J}_c$ in *Equation 78* in our case) that has no biological meaning. The authors of *Etournay et al., 2015* used such measurements in their Figure 5 and Videos 6 and 7, and found the convective contribution to be negligible. On the other hand, we found that it contributes significantly to the balance. Like us, they also perform their measurements over moving and deforming tracked patches (Lagrangian treatment) which conserve the same set of cells (and their offspring) over time (in Figures 3, 6 and 8 for

instance). By doing so, there is therefore no inward and outward flux of cells across each compartment boundary, the only fluxes being at the animal or image boundaries (our *Figure 3—figure supplements 1h,2h*), and the balance equation is easier to interpret without this term (*Equation 15*).

In the future, it should be possible to compare our approach with that of (*Etournay et al., 2015*). For instance, it would be relevant to quantify the difference between our respective results for each of our simulations, for a same tissue, or alternatively for tissues with a large number of four-fold or higher vertices, of cell rearrangements which involve four or more cells, and of delaminations which involve four or more cells. It would also be useful to compare the impact on the balance equation of cell fluxes either occurring at the boundaries of the tissue, of the field of view, or of each Eulerian grid compartment (for different compartment sizes). Lastly, by performing on their Video 6 an averaging over time, space and animals similar to ours, it would be interesting to compare their analysis of wing morphogenesis and their signal-to-noise ratio with ours (*Video 5*).

## E Glossary for segmentation and tracking

- Apoptosis: see 'delamination'.

- Box: fixed region used as a particular case of cell patch specific to Eulerian measurement. It is always smaller or equal to the field of view. It is usually (but not necessarily) much larger than a cell size, and rectangular.

- Coalescence: see 'fusion'.

- Core cell: cell completely within the field of view's boundary, which does not touch an incomplete cell nor the sample's boundary (or the box boundary, in case of Eulerian measurement).

- Delamination: disappearance of a cell apical surface. In our movies, it corresponds either to actual apoptosis or to exit of apical surface through 3D cell shape change. Technically, it is detected as the end of one cell by size decrease to zero, without start at the same time, same place, and without exit through the field of view's boundary.

- Division: end of one cell and at the same time, same place the starts of two neighboring cells.

- End: disappearance of a cell between images $t$ and $t + \delta t$. In our movies, this is due to either division, delamination or exit through the field of view's boundary.

- Eulerian: point of view of the experimentalist, who is fixed, and looks at cells passing in front of the camera's field of view (or a subset of it, see 'box'). Refers to quantities which are a function of time and space.

- Field of view: image defined by the camera. If it is a simple field of view, it is rectangular. It can also result from pasting several camera images together.

- Field of view's boundary: extreme limits of the image defined by the camera. If the field of view is simple, its boundary consists of four straight lines.

- Fusion: ends of two neighboring cells and at the same time, same place the start of one cell. Also called 'coalescence'. In our movies, cells do not fuse, and such event corresponds to a segmentation error.

- Gain: creation of a link between images $t$ and $t + \delta t$. In our movies, this is due to either division of this cell's mother, rearrangement or entrance through the field of view's boundary.

- Incomplete cell: cell which intersects the field of view's boundary.

- Integration: progressive insertion of a cell apical area into the visible layer. In our movies, it corresponds to a segmentation error. Also called 'nucleation'. Technically, it is detected as a start of one cell by size increase from zero, without division or fusion, and without entering the field of view.

- Junction: separates two cells.

- Junctional stress: part of the mechanical stress due to cell junction tensions.

- Lagrangian: point of view attached to a fixed group of cells (see 'patch'). Refers to quantities which are a function of time and of these cells. They depend on space only implicitly, through these cells' positions.

- Link: vector $\vec{\ell}$ which starts at a core cell's site, and ends at one of its neighbor's site. Each link is associated to a junction, to which it is roughly perpendicular.

- Loss: disappearance of a link between images $t$ and $t + \delta t$. In our movies, this is due to either division, delamination, rearrangement or exit through the field of view's boundary.

- Mechanical stress: forces internal to the tissue, exerted by a cell patch onto a neighbor cell patch. It is expressed per unit length of patch boundary in 2D, per unit area of patch boundary in 3D.

- Nucleation: see 'integration'.

- Patch: group of cells used for coarse-grained description. It can be small or large. Although in principle it is not strictly necessary, in practice we usually choose a patch completely included at all times in the field of view (and even in the core cells). For Eulerian description, see 'box'. In Lagrangian description, a patch is conserved through time and includes the offspring of the initial cells.

- Rearrangement: change in the list of neighbor cells. This can be a gain or a loss of a junction, and thus of a link, between adjacent cells. A gain followed by a loss within a group of four cells is called a topological process of the first kind (T1).

- Sample's boundary: limit of the recognizable cells. It depends on the imaging configuration, and can be either inside or outside of the field of view's boundary.

- Site: a point defined for each cell. In practice, we choose its center of mass.

- Start: creation of a cell between images $t$ and $t + \delta t$. In our movies, this is due to either division of this cell's mother, or entrance through the field of view's boundary.

