## [Decision Letter]

Thank you for submitting your work entitled "Unified Quantitative Characterization of Epithelial Tissue Development" for peer review at *eLife*. Your submission has been favorably evaluated by Naama Barkai (Senior Editor and Reviewing Editor) and three reviewers.

The reviewers have discussed the reviews with one another and the Reviewing editor has drafted this decision to help you prepare a revised submission.

Summary:

As you can appreciate from the individual reviews, all reviewers greatly appreciated the new formalism you describe for analyzing how different cellular processes to tissue morphogenesis. The application for large-scale analysis of a specific morphogenetic process was also appreciated, although here there was some debate of whether the results are preliminary or already provide deep, or broad enough biological insights.

Essential revisions:

Please try to revise the writing to make it less technical and more accessible to broad biological audience.

The impact of the paper could be greatly increased by predicting and validating the morphogenetic outcome of perturbing a specific cellular process. Please consider if this is within the scope of the present manuscript, and if not, relate to it in the writing. Specific/minor points are included in the reviews below.

*Reviewer #1:* This is a well-written paper presenting a novel method for analyzing the interplay of various mechanisms for tissue remodeling and morphogenesis. The method not only provides a powerful tool for experimentalists to better visualize and understand their data, it should give theorists more microscopic insights to construct models.

I just have the following questions before recommending the paper for publication:

1) If cell rearrangement also happens via higher order intercalations (5-fold vertices or higher), how would this additive model account for the various order of intercalations, are they simply additive? I.e. R=R(4)+R(5)+…?

2) Can the formalism be applied in the presence of cell-substrate interactions in addition to cell-cell interactions, such as when cells are self-propelled?

3) I think the method here requires the tissue to be confluent? Is the texture tensor a good quantity to use when there are holes in the tissue or there are boundaries involved?

*Reviewer #2:* The manuscript "Unified Quantitative Characterization of Epithelial Tissue Development" Yohanns Bellaiche and colleagues details the development of a formalism that describes the contributions of several distinct cellular process to larger-scale tissue morphogenesis. The formalism is validated against in silico datasets before being applied to the large-scale analysis of morphogenetic processes in the developing *Drosophila* notum. Specifically, the authors use the formalism to address the relationship between cell division orientation and mechanical stress in epithelia on an unprecedented scale.

In general, this paper represents a highly significant contribution to the field of epithelial morphogenesis, even though the impact is primarily conceptual. Most importantly, this work demonstrates a robust quantitative method to understand the coordinated behavior of thousands of cells by precisely measuring several key parameters observable in massive timelapse captures of a developing tissue (Division, Rearrangement, Size/shape, Apoptosis, etc.). While several large scale models for epithelial morphogenesis have been described, this work advances the field through development of the novel formalism combined with the scale of analysis in a developing biological tissue.

A secondary impact of the work lies in the dissection of a complex role for cell division in both promoting and opposing tissue elongation, along with evidence that morphogenesis of the pupal notum may involve a more complex interplay between mechanical stress, tissue elongation and cell division than was previously appreciated.

Substantial concerns:

The notion of the formalism itself is a bit obtuse for the average biologist and this should be clarified to make the main concepts more accessible.

The impact of the work would be greatly improved by demonstrating that genetic or biophysical disruption of the tissue (i.e. through gain/loss of function cell clones, mechanical injury, etc.) leads to predictable and reproducible changes in the various cell parameters under consideration. In other words, can the formalism be used to elucidate cell- and tissue-level implications of different genetic or mechanical perturbations? Similarly, can it be applied to other experimental systems/context? As it stands, the work nicely describes wild-type notum morphogenesis but it seems to me the impact of this methodology could be significantly extended.

*Reviewer #3:* The paper by Graner, Bellaiche et al. aims at providing a global computational framework to quantify tissue growth during morphogenesis. This framework includes changes in cell shape and cell rearrangements, as previously published by others, but incorporates other important sources of tissue growth such as cell division and apoptosis. The authors apply this formalism to their highly validated and enormous dataset during pupal metamorphosis. The combination of the complete set of tools to characterize tissue growth with the inference of tissue forces from image segmentation will certainly lead to new discoveries in the field.

1) The paper is mainly technical. As such, it will appeal to a relatively small community. There is no doubt that the method reported here has great potential but the current manuscript does not provide a fundamental advance in our understanding of biology (other than the distinct contributions of cell division to tissue growth). If technical papers like this one fall within the scope of e*Life* then this paper provides an interesting formalism to systematically map tissue dynamics. Otherwise, I think specialized readers would prefer another type of manuscript integrating the main text and the supplement into a single body, and including a more detailed validation.

2) Very important methodological aspects are missing or buried in the formalism of the supplementary materials. The main text should include additional details on how each contribution to the global growth rate is determined. For example, the reader is left wondering how the authors distinguish between cell apoptosis or delamination, or between cell division, inclusion of a new cell, and a cell rearrangement. In general the main text should be much more informative on the methods and, where possible, provide schemes and intuitive pictures to the non-expert reader.

3) The validation of the method (Figure 2) is largely qualitative. The authors should first explain the types of errors the method is sensitive to, and then quantify those errors.

4) The examples on Figure 1 are not very clear. In particular changes in size and shape in panels C, D, E are not visual. Perhaps a more schematic cartoon-like illustration would help.

5) Can the third dimension be really ignored? What is the error in the deformation rates caused by the assumption of a 2D tissue?

6) In the quantification of the force field the authors assume the junctions are the sole structures responsible for stress transmission. However, the cell body is likely to contribute to the generation of active stresses through, for example, changes in cell volume (see Saias et al., Dev Cell 2015) or non-junctional cytoskeletal elements. This should be critically addressed.

7) The finding that cell division can lead to positive or negative growth is intriguing but very preliminary. The authors should at least provide some images of the tissue illustrating the mechanisms underlying the two opposite behaviors. A cartoon would also help.

---

## [Author Response]

*Please try to revise the writing to make it less technical and more accessible to broad biological audience. The impact of the paper could be greatly increased by predicting and validating the morphogenetic outcome of perturbing a specific cellular process. Please consider if this is within the scope of the present manuscript, and if not, relate to it in the writing.*

We thank you and the referees for constructive comments that helped us improve the quality of our manuscript. As requested, we have substantially modified the manuscript to improve its impact and general relevance as follows:

i) For biologists, we have edited the main text and the figures to make the manuscript more accessible to a broad audience: (a) we have removed or replaced mathematical and technical terms; (b) as requested by reviewer 3, we have provided three additional or improved schematics to better explain how tissue growth and morphogenesis is measured (Figure 1; Figure 1—figure supplement 1; Figure 1), how the formalism is implemented in practice (Figure 1 and Video 1) and how the role of each cell process is measured (Figure 4). For physicists, we have significantly improved the appendix by providing a more comprehensive and detailed description of the formalism, its principle and implications.

ii) As suggested by reviewer 2, we have applied the formalism to another epithelial tissue (Figure 5 and Figure 5—figure supplement 1; Video 5). By analyzing a time-lapse movie corresponding to ~2 10^6^ cell contours, we have performed a complete analysis of the wing pupal development during 17 hours. This analysis improves the generality of our works since:

The pupal wing is an important model to study epithelial tissue morphogenesis, this novel dataset and its analysis is therefore of general relevance for the study of tissue morphogenesis.

In our initial version, by analyzing the notum pupal epithelial tissue we described and quantified an unexpected case of interplay between cell division orientation and tissue elongation where cell division is orthogonal to the overall tissue elongation (Figure 4). In the revised version we have demonstrated that this interplay is also present in the central region of the wing (Figure 5). This therefore reinforces the generality of our initial findings.

iii) As requested, we aimed at perturbing a cellular process to illustrate how the formalism can be used to analyze the role of cell processes during epithelial tissue morphogenesis (Figure 6 and Figure 6—figure supplement 1). To this end, we have recorded the morphogenesis of five mutant tissues where the *tribbles* gene (an inhibitor of G2/M transition) is overexpressed to block cell division (*trbl^up^* tissue). This dataset of ~3.7 10^6^ cell contours has been fully segmented and analyzed and has yielded novel results important for the field of tissue development:

In *trbl^up^* tissue, both visual inspection and cell tracking revealed a drastic reduction of the number of cell divisions (4.3% of the number in the wild-type tissue). As expected we found that the convergence-elongation rate of division measured by the formalism nearly vanished.

To go beyond this expected prediction and to analyze in more detail the role of cell division and its interplay with other cell processes, we have then developed a complete set of general methods to compare wild-type and mutant conditions (“Comparisons of wild-type and mutant conditions”). Applying these methods to our experimental dataset demonstrates that both cell proliferation and cell division orientation are critical to promote tissue elongation and furthermore that cell divisions indirectly impact both cell rearrangements and cell shape changes.

To further analyze and better understand these experimental findings, we have performed additional computer simulations. As observed in the experimental dataset, the computer simulations show how cell divisions can promote tissue elongation regardless of their orientation by indirectly influencing both cell rearrangements and cell shape changes during tissue elongation (Figure 6).

Altogether, the above additional results from experiments and computer simulations illustrate that our formalism enables to study quantitatively tissue morphogenesis and to uncover complex interplays between cell divisions, cell rearrangements and cell shape changes by quantifying tissue dynamics in wild-type and mutant tissues and by allowing the comparison between experimental and simulated datasets.

In conclusion we have fully addressed all the reviewers' comments: (i) by editing the manuscript to make it more accessible to biologists and by adding schematics to better explain its technical aspects; (ii) by applying the formalism to another epithelial tissue; and (iii) by analyzing a mutant condition and computer simulations to illustrate the relevance of the formalism in the study of epithelial tissue morphogenesis. We therefore hope the manuscript will now be deemed suitable for publication.

Reviewer #1:

*[…] I just have the following questions before recommending the paper for publication: 1) If cell rearrangement also happens via higher order intercalations (5-fold vertices or higher), how would this additive model account for the various order of intercalations, are they simply additive? I.e. R=R(4)+R(5)+*…*?*

Yes, one of the strong points of the formalism is that it allows to take into account and sum up all cell processes. This includes decomposing the total contribution of rearrangements into its different types.

We have clarified this point by stating in the main text: “Whenever needed, the subdivision of these main cell processes can be further refined, for instance in a mono-layered tissue to distinguish apoptoses from live cell extrusions, or to distinguish simple rearrangements through 4-fold vertices from those with 5-fold vertices or higher.” We have also detailed in the appendix how such a decomposition could be performed.

*2) Can the formalism be applied in the presence of cell-substrate interactions in addition to cell-cell interactions, such as when cells are self-propelled?*

Yes, the formalism uses cell contours and cell lineages to quantify tissue morphogenesis. This approach, which delivers important and quantitative information on tissue development, is purely descriptive and independent of the forces that give rise to this morphogenesis. It can therefore be applied whether in presence or in absence of cell substrate interaction, to describe the respective contribution of each morphogenetic process to tissue deformation.

We now state in the appendix that “our formalism for deformable cellular materials quantifies observed morphogenetic changes independently of the biological or physical origin of the forces that drive these changes, whether internal forces such as active cytoskeletal forces, viscous dissipation or cytoplasm pressure, or external forces such as interaction with another cell layer or solid substrate.”

In addition, the force inference method used to estimate the junctional stress (i.e.the stress transmitted at cell junctions) is independent of other contributions to stress or of external force contributions; it thus remains valid even in the presence of cell-substrate interactions that can cause an additional contribution to the total mechanical interactions.

We now state in the Materials and methods (Force and stress inference): *"*Junctions contribute to transmit stress between cells even if the biological origin of this stress lies in non-junctional cytoskeletal elements and other structures in the body of the cell. Forces created in the cell body or at the cell boundary are transmitted to its neighbours by contact at the cell-cell interface. These forces transmitted by contact can be decomposed in components parallel and perpendicular to the junction, namely tension and pressure, which we both estimate. Changes of cell volume lead to measurable changes in pressure and thus to detectable changes in our stress estimation.*"*

*3) I think the method here requires the tissue to be confluent? Is the texture tensor a good quantity to use when there are holes in the tissue or there are boundaries involved?*

The formalism is constructed rigorously to take into account all links between cells, and all their possible changes, whether these cells are at the tissue boundary or not. It thus applies to tissues with boundaries, even with internal ones like in the case of actual holes in the tissue, or in the case of holes in the skeletonized images due to segmentation problems.

We state in the manuscript appendix that describes in detail the formalism: “Note also that the formalism is meant to extract relevant information from cell-cell links, to apply to a confluent cell assembly. Still, J* systematically takes into account all links between cells at the tissue boundary, and all their possible changes. The formalism is thus designed to rigorously take into account all links between cells, and all their possible changes, whether these cells are at the tissue boundary or not. It applies to tissues with boundaries, even with internal ones like in the case of actual holes in the tissue, or in the case of holes in the skeletonized images due to segmentation problems.*”.*

Reviewer #2:

*[…] Substantial concerns: The notion of the formalism itself is a bit obtuse for the average biologist and this should be clarified to make the main concepts more accessible.*

We have improved the manuscript by making it more accessible to biologists: (i) by removing mathematical and technical terms (ii) by focusing on the four main epithelial processes: cell division, cell rearrangement, cell shape and size change and apoptosis; and (iii) by introducing or improving cartoon-like schemes to better explain the measurements of the formalism (Figure 1; Figure 1—figure supplement 1; Figure 4). Importantly, an appendix now provides in-depth mathematical description of the formalism and its practical implementation.

*The impact of the work would be greatly improved by demonstrating that genetic or biophysical disruption of the tissue (i.e. through gain/loss of function cell clones, mechanical injury, etc.) leads to predictable and reproducible changes in the various cell parameters under consideration. In other words, can the formalism be used to elucidate cell- and tissue-level implications of different genetic or mechanical perturbations? Similarly, can it be applied to other experimental systems/context? As it stands, the work nicely describes wild-type notum morphogenesis but it seems to me the impact of this methodology could be significantly extended.*

We have fully addressed this point by (i) applying the formalism to another epithelial tissue: the *Drosophila* pupal wing (Figure 5; Figure 5—figure supplement 1; Video 5) and (ii) analyzing, using both mutant experimental data and computer simulations, the impact of a drastic reduction of the rate of cell division (Figure 6 and Figure 6—figure supplement 1). We have described in detail the important conclusions we could reach (please see detailed answer in the “Essential revisions” section above) and how these additional results improve the general relevance of our manuscript and provide a comprehensive set of methods to study tissue morphogenesis.

Reviewer #3:

*[…] 1) The paper is mainly technical. As such, it will appeal to a relatively small community. There is no doubt that the method reported here has great potential but the current manuscript does not provide a fundamental advance in our understanding of biology (other than the distinct contributions of cell division to tissue growth). If technical papers like this one fall within the scope of* eLife

*then this paper provides an interesting formalism to systematically map tissue dynamics. Otherwise, I think specialized readers would prefer another type of manuscript integrating the main text and the supplement into a single body, and including a more detailed validation.*

*eLife* guidelines to authors specifically encourage “Tools and Resources” papers. In addition, with respect to the initial version, the revised version has been significantly improved by the incorporation of additional validation, and additional results in the pupal wing and in mutant notum tissues. Finally, we have followed the editors’ recommendation to make the text more accessible to biologists.

*2) Very important methodological aspects are missing or buried in the formalism of the supplementary materials. The main text should include additional details on how each contribution to the global growth rate is determined. For example, the reader is left wondering how the authors distinguish between cell apoptosis or delamination, or between cell division, inclusion of a new cell, and a cell rearrangement. In general the main text should be much more informative on the methods and, where possible, provide schemes and intuitive pictures to the non-expert reader.*

We have addressed this point as follows:

To help the readers and as suggested by the reviewer (point 4), we have added schemes to better illustrate how each cell process is measured by changes in link between cells (Figure 1 and Figure 1—figure supplement 1).

We have added a figure to illustrate that the formalism is applied after cell segmentation and cell tracking and lineage determination (Figure 1); Furthermore, we have described in more detail how upon segmentation of the movie each process is distinguished by cell tracking and cell lineage determination prior to applying the formalism (Materials and methods, “Practical implementation of the formalism”).

We have revised our manuscript to make sure that all methodological and practical aspects are fully described in the text or the Materials and methods, while the appendix contains all technical explanations regarding the theoretical framework. We have followed the overall recommendation to make the main text more accessible to biologists.

Regarding delamination versus apoptosis, we explicitly state that in our movies, our segmentation and tracking methods do not distinguish between delamination associated with apoptosis or cell delamination without apoptosis (live cell extrusion). Distinguishing between the two types of delamination can only be done by introducing an additional marker such as Histone-mRFP to observe nuclear fragmentation. We have described in the appendix how delamination and apoptosis, if they were distinguished by image analysis, could be independently quantified by the formalism, if needed.

Importantly, all cell processes, which can be distinguished by analyzing cell contours, are characterized during the cell tracking and then rigorously quantified separately by our formalism to characterize their contribution to tissue development.

We state in the appendix: “the contribution of delaminations and apoptoses. If needed, this can be subdivided into sub-processes, each associated with a different biological origin, for instance to distinguish simple extrusions from extrusions associated to apoptoses; or apoptoses which occur for cells with 3, 4 or more sides. Their contributions naturally add up, by construction of our formalism.”

3) The validation of the method (Figure

*2) is largely qualitative. The authors should first explain the types of errors the method is sensitive to, and then quantify those errors.*

We have made our validations more quantitative. In the revised version, we have first validated G(tissue deformation) and S(cell shape and size changes) measurements by performing a simulation where a cell patch undergoes a known isotropic dilation followed by a known contraction-elongation along the *x* axis (Figure 2, Video 2). We then measured separately G and S(both dilation and CE rates), compared the measurements to the input deformation and found an error of 0.3%.

Having validated G, we then validated the measurements of D(divisions, Figure 2 and Video 2), R (rearrangements, Figure 2 and Video 2), and A(apoptoses, Figure 2 and Video 2), with simulations involving almost exclusively those cellular processes (since some residual rearrangements and cell shape changes always occur in our simulations due to cell structure disorder). The analysis yields the balances G= D, G= Rand G= A, respectively (up to residual contributions of R and S), thereby validating D, Rand Ameasurements. As now detailed in the appendix, we have explained that the tissue deformation rate is independent of rigid body movements including rotation. This is illustrated and validated by performing additional computer simulations (Figure 1—figure supplement 1 D, E and Video 2).

In the revised version, we have added a complete section in the Materials and methods (Uncertainties) where we now discuss more precisely the errors due to image analysis (segmentation and tracking), which affect the input of our formalism, as well as the approximations used in our description, which affect the output of our formalism.

4) The examples on Figure

*1 are not very clear. In particular changes in size and shape in panels C, D, E are not visual. Perhaps a more schematic cartoon-like illustration would help.*

As requested we have improved this figure by providing schematic cartoon-like illustrations to explicitly describe how changes in link geometry and topology can be used to quantify the four main cell processes (cell division, cell shape change, delamination and cell rearrangements, Figure 1) as well as the other processes (Figure 1—figure supplement 1).

*5) Can the third dimension be really ignored? What is the error in the deformation rates caused by the assumption of a 2D tissue?*

We image the tissue in 3D using the E-Cadherin:GFP marker, which labels the apical *adherens* junctions in *Drosophila* tissue. This permits to measure the tissue curvature of this cuboidal monolayer epithelial tissue. We have better emphasized that the corrections due to tissue curvature are small in the notum and entirely negligible in the wing.

We have mentioned in the Materials and methods (Uncertainties): "The 3D structure of the notum also plays a role because the tissue is curved. It is imaged in 3D using a confocal microscope; the height of its surface is recorded and we can entirely reconstruct its curvature. […] For simplicity, we choose not to perform any correction to our results, knowing that we can perform the correction a posteriori if it appears necessary in the future. All these effects are completely negligible in the wing, which is much flatter than the notum.”

*6) In the quantification of the force field the authors assume the junctions are the sole structures responsible for stress transmission. However, the cell body is likely to contribute to the generation of active stresses through, for example, changes in cell volume (see Saias et al., Dev Cell 2015) or non-junctional cytoskeletal elements. This should be critically addressed.*

In the revised version, we clarify that what we measure is the junctional stress (Materials and methods, Force and stress inference). Junctions contribute to transmit stress from cell to cell even if the biological origin of this stress lies in non-junctional cytoskeletal elements and other structures in the body of the cell. As we explicitly state, forces created in the body of a cell or at its boundary are transmitted to its neighbors through contact forces at the junction, which can be decomposed in components parallel and perpendicular to the junction, namely tension and pressure, which we both estimate (Figure 7—figure supplement 1). Changes of cell volume (such as observed by Saias et al., Dev Cell 2015) lead to measurable changes in pressure and thus to detectable changes in our measurements.

We agree that junctional stress can differ from the total stress in some cases, and that junctions are not the sole structures responsible for stress transmission. We explicitly state that there can be contributions that we do not measure in the total stress (Materials and methods, Force and stress inference): “Note that we cannot measure other contributions to the total stress, for instance through cell membrane parts far from adherens junctions.” Our work focuses on junctional stress because of the growing consensus that junctional stress is an important determinant of tissue morphogenesis as measured by laser ablations of (and outside of) *adherens* junctions (for review Lecuit et al., Annu Rev Cell Dev Biol, 2011), or directly probed by experimental manipulation of *adherens* junctions and compared with simulations (Bambardekar et al., PNAS 2015).

7) The finding that cell division can lead to positive or negative growth is intriguing but very preliminary. The authors should at least provide some images of the tissue illustrating the mechanisms underlying the two opposite behaviors. A cartoon would also help.

In the revised version, we now clarify that division can only contribute positively to growth, in the sense that the isotropic part of the division tensor Dis always positive, thereby always having a positive effect on local tissue growth in term of area increase (Figure 3—figure supplement 1).

In the present article we focus on the anisotropic parts of tensors, which correspond to contraction-elongations that do not change the local tissue area and impact tissue morphogenesis. To address the reviewer comment we have improved our manuscript as follows:

We have rephrased the definition of the projection of each cell process measurement along the total tissue CE rate (G). To avoid any confusion, we now call it “component along G”. We have also included a figure panel to explain more precisely the meaning of a negative versus positive component of each cell process (Figure 4). A negative component of Dsimply results from cell divisions mainly oriented orthogonally to tissue elongation.

We have performed additional simulations of cell divisions oriented orthogonally to the main direction of tissue elongation, illustrating the case of a negative component of cell division along tissue elongation (Figure 6). This illustrates the mechanisms underlying the two opposite cell division behaviors: aligned with versus orthogonal to tissue elongation.

To reinforce the general relevance of our initial findings on cell division, we have firstly performed a complete analysis of wing morphogenesis (Figure 5 and Figure 5—figure supplement 1; Video 5). This analysis shows that divisions are found orthogonal to the main direction of tissue elongation in this tissue too. We have secondly undertaken an in-depth analysis of the role of cell divisions. This was achieved by overexpressing the *tribbles* gene to prevent cell proliferation, analyzing using the formalism five *tribbles* overexpressing tissues (Figure 6 and Figure 6—figure supplement 1), defining a complete set of methods to rigorously compare wild-type and mutant conditions, and using both experimental and simulation data to analyze a complete analysis of cell division contribution to morphogenesis (please see detailed answer in the “Essential revisions” section above). We have therefore validated that the formalism allows comprehensive and novel studies of cell processes during tissue morphogenesis.